# Robust Causal Discovery in Real-World Time Series with Power-Laws

**Matteo Tusoni**[†,‡ 1]   **Giuseppe Masi**[1]   **Andrea Coletta**[‡ 2]   **Aldo Glielmo**[‡ 2]   **Viviana Arrigoni**[1]   **Novella Bartolini**[1]

## Abstract

Exploring causal relationships in stochastic time series is a challenging yet crucial task with a vast range of applications, including finance, economics, neuroscience, and climate science. Many algorithms for Causal Discovery (CD) have been proposed; however, they often exhibit a high sensitivity to noise, resulting in spurious causal inferences in real data. In this paper, we observe that the frequency spectra of many real-world time series follow a power-law distribution, notably due to an inherent self-organizing behavior. Leveraging this insight, we build a robust CD method based on the extraction of power-law spectral features that amplify genuine causal signals. Our method consistently outperforms state-of-the-art alternatives on both synthetic benchmarks and real-world datasets with known causal structures, demonstrating its robustness and practical relevance.

## 1. Introduction

Causal Discovery (CD) from stochastic time series aims to identify causal relationships among time-evolving variables purely from observational data. CD algorithms represent a domain-agnostic alternative to analytical modeling, which can be impractical in many scientific domains characterized by complex dynamics. The resulting causal model is typically represented with a *causal graph*, where nodes are variables, and directed edges reflect asymmetric causal dependencies between them. This methodology has been successfully employed on a vast range of fields, including climate science (Mosedale et al., 2006; Nowack et al., 2020), neuroscience (Bressler & Seth, 2011; Seth et al.,

---

[†] This research was conducted as part of an internship at Banca d'Italia. [‡] The views expressed in this article are those of the authors and do not necessarily represent the views of Banca d'Italia or the Eurosystem. [1]Department of Computer Science, Sapienza University of Rome, Rome, Italy [2]Banca d'Italia, Rome, Italy. Correspondence to: Matteo Tusoni <tusoni@di.uniroma1.it>.

2015), finance (Billio et al., 2012; Diebold & Yılmaz, 2014), and, more recently, generative AI (Kocaoglu et al., 2018; Yang et al., 2021; Jiao et al., 2024). Nevertheless, inferring causal relationships in time series is particularly challenging due to factors such as noise and non-stationarity (i.e., time-varying dynamics), which can obscure the underlying causal structure and reduce the robustness of causal discovery algorithms. Classical CD methods, most notably Granger Causality and its extensions, rely on restrictive assumptions such as noise stationarity and the existence of a single characteristic scale to define vector autoregressive (VAR) models appropriately. Unfortunately, these assumptions are frequently violated, as real-world systems are typically non-equilibrium, history-dependent, and often display scale-free temporal correlations and power-law frequency spectra (Bak, 2013). In such contexts, conventional CD algorithms can easily incur errors, and detect spurious relationships or fail to detect true interactions. To address these shortcomings, we introduce **PLaCy** (**P**ower-**La**w **C**ausal discover**y**), which is specifically designed to leverage the scale-free properties commonly observed in real-world time series. Instead of comparing variables at individual time points, it fits a power-law model to the frequency spectrum of each process and tracks the evolution of the fitted spectral exponents and amplitudes. In this way, **PLaCy** isolates structural causal changes that propagate from one variable to another by filtering out non-stationary and nonlinear external influences, bearing the absence of a characteristic scale. Classical Granger-type hypothesis tests are then applied to the trajectories of power-law spectral exponents and amplitudes, rather than to the raw signals, preserving the statistical power of established testing theory. By running extensive experiments on synthetic benchmarks with controlled nonlinear and non-stationary noise, or scale-free characteristics, as well as on two real-world data sets, we demonstrate that **PLaCy** outperforms state-of-the-art CD methods, particularly in regimes where the non-equilibrium, nonlinear, or scale-free properties of the time series are more pronounced.

The main contribution of this paper is the following:

- We propose **PLaCy**, a novel framework that leverages spectral trends for robust causal discovery in time-series with power-law frequency distributions.

- We theoretically demonstrate that the frequency-domain transformation used in **PLaCy** preserves the underlying causal graph structure, guaranteeing results consistent with the time-domain graph.

- We empirically show that **PLaCy** provides more robust and accurate estimations, validated through extensive experiments on both synthetic and real-world datasets.

## 2. Preliminaries

### 2.1. Causal Discovery

Causal Discovery is the task of identifying the underlying structure of cause-and-effect relationships among the components of a multivariate system. Given a collection of observed multivariate time series, the goal is to infer a directed graph that encodes which variables influence others in a causal sense. Formally, given a time-series $\mathbf{x} \in \mathbb{R}^{L \times d}$, the task is to determine a directed graph $G = (V, E)$, where $V = \{1, 2, \ldots, d\}$ represents the variables of the system, and $E \subseteq V \times V$ is the set of directed edges. A directed edge $(i, j)$ exists if and only if $\mathbf{x}_i$ is inferred to be a cause of $\mathbf{x}_j$. The goal of our approach is to derive the causal graph representing the causal relationships among the variables.

### 2.2. Granger Causality

Granger causality holds when past values of one time series provide statistically significant predictive information for another. In particular, we say that $\mathbf{x}_i$ *Granger-causes* $\mathbf{x}_j$ if the past values of $\mathbf{x}_i$ are useful to predict $\mathbf{x}_j$, given the past of all other time series. In time series data, Granger causality is typically studied using a multivariate vector autoregressive model (VAR) (Lütkepohl, 2005):

$$\mathbf{x}(t) = \sum\nolimits_{\tau=1}^{T} \mathbf{A}_\tau \mathbf{x}(t - \tau) + \boldsymbol{\varepsilon}_t,$$

where $\mathbf{x}(t)$ is the multivariate time-series at time $t$, with each component defined as a linear combination of the past $T$ values of all variables. Granger causal analysis involves fitting a VAR model and testing the statistical significance of the autoregressive coefficient matrices $\mathbf{A}_\tau$, typically using a Wald test (Fahrmeir et al., 2013). This requires comparing two models: the unrestricted model, which includes lagged terms of both $\mathbf{x}_i$ and $\mathbf{x}_j$, and the restricted model, which excludes the lagged terms of $\mathbf{x}_i$ from the prediction of $\mathbf{x}_j$. The null hypothesis states that all coefficients related to the lagged $\mathbf{x}_i$ terms are zero. Failing to reject this null hypothesis implies that $\mathbf{x}_i$ does not Granger-cause $\mathbf{x}_j$. Notice that the Granger causality definition does not explicitly account for the time elapsed between cause and effect, since it jointly tests all specified lags together. Similarly, in our work, we focus on identifying the existence of causal relationships, regardless of the specific time lag between cause and effect.

### 2.3. Power-laws in the real-world

Over the past six decades, extensive empirical evidence has shown that power-law spectra of the form $S(f) \propto f^{-2\lambda}$, with $\lambda > 0$, are ubiquitous in real-world time series. Classic examples can be found in finance (Bonanno et al., 2000; Di Matteo et al., 2005), climate science (Huybers & Curry, 2006; Fredriksen & Rypdal, 2016) or neuroscience (Palva et al., 2013; He, 2014; Gyurkovics et al., 2022; Medel et al., 2023). Power-law spectra frequently arise in systems composed of many interacting units, such as traders in a market or nodes in communication networks, that *self-organize* into structured behavior without any external regulator/coordinator (Bak et al., 1987; Bak & Chen, 1991). Specifically, self-organizing systems often exhibit scale invariance (Proekt et al., 2012), precisely due to the absence of any external coordinator enforcing a characteristic scale. A stochastic process $\{x(t)\}$, is *scale invariant* if $\forall a \in \mathbb{R}^+$, the rescaled process $\{x(at)\}$ is statistically equivalent to $\{a^H x(t)\}$, for some $H \in \mathbb{R}^+$. This property implies that any magnified fragment of a scale-invariant stochastic process looks identical to the original series and, for this reason, scale invariance is sometimes referred to as self-similarity and is very related to the geometric concept of a fractal (Bak, 2013). It is also known that, under very loose assumptions, scale-invariant stochastic processes are also *scale-free*, meaning that they exhibit power-law correlations, and power-law distributed frequency spectra with exponent $\lambda = H - 1/2$ (Flandrin, 1989). Given the ubiquity of power-law distributed frequency spectra in the real-world, this structural regularity can be leveraged to improve the extraction of causal signals from time series, reducing spurious temporal dependencies.

## 3. Proposed Methodology

A well-established approach in signal processing involves analyzing the frequency content of a signal via its spectral representation. To this end, we employ the Discrete Fourier Transform (DFT). Given a real-valued time series $x(t)$ of length $L$, the DFT is defined as:

$$\phi(k) = \sum\nolimits_{t=0}^{L-1} x(t) \, e^{-i2\pi \frac{k}{L} t}, \quad k \in \{0, \ldots, L-1\}, \quad (1)$$

where $\phi(k) \in \mathbb{C}$ denotes the complex-valued coefficient corresponding to the $k$-th discrete frequency. The associated normalized frequency is given by $f_k = \frac{k}{L}$. The magnitude of each Fourier coefficient quantifies the contribution of the corresponding frequency component to the overall signal. We therefore denote the *spectral amplitude* as $A(f_k) = |\phi(k)|$.

As discussed previously, many natural and social systems exhibit long-range dependencies and scale-free behavior in their frequency content, often associated with self-organized phenomena. A defining characteristic of these systems is the

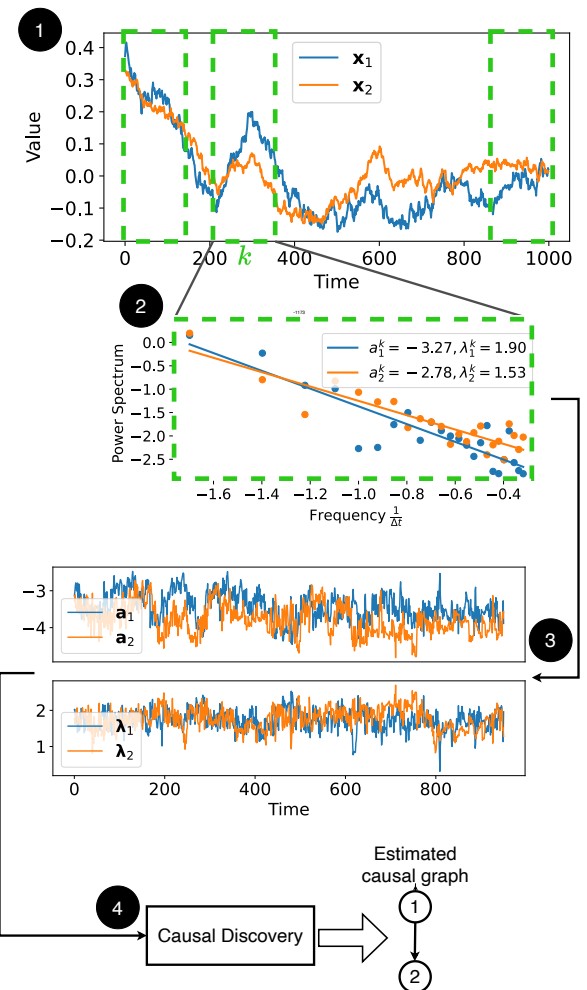

*Figure 1.* **Schematic illustration of the proposed methodology**. The original time series, here $x_1$ and $x_2$, are segmented into overlapping windows (**step ❶**). Then, for each window $k$, the amplitudes $(a_1^k, a_2^k)$ and the exponents $(\lambda_1^k, \lambda_2^k)$ of the power-law distributed spectra are computed (**step ❷**). These give rise to new, multi-dimensional, time series: $(\mathbf{a}_1, \boldsymbol{\lambda}_1)$ for $x_1$ and $(\mathbf{a}_2, \boldsymbol{\lambda}_2)$ for $x_2$ respectively (**step ❸**). Finally, multivariate Granger causality tests are performed on these new series, and the causal graph is constructed (**step ❹**).

power-law decay of their power spectral amplitude, typically modeled as $A(f) = e^a \cdot f^{-\lambda}$, where $e^a$ is a scaling constant and $\lambda > 0$ is the *spectral exponent*. The exponent $\lambda$ is tightly linked to structural features of the process, such as its autocorrelation. Importantly, the spectral parameters $a$ and $\lambda$ may vary over time due to exogenous perturbations or endogenous interactions. These variations provide an opportunity to study causal structures through their temporal dynamics. Instead of analyzing the raw time series directly, we propose to monitor the evolution of $(\mathbf{a}, \boldsymbol{\lambda})$ as informative summaries of the underlying processes. To achieve this, we segment each time series into overlapping windows (**step ❶** in Figure 1) and compute the local spectral parameters

within each window. This is done by estimating the slope and intercept of the spectrum in log-log space:

$$\log A(f) = a - \lambda \log f. \tag{2}$$

The linear form permits efficient estimation via ordinary least squares, yielding one value of $a$ and $\lambda$ per window (**step ❷** in Figure 1). Repeating this procedure across the entire series results in two new time series per original signal: $\mathbf{a}$ and $\boldsymbol{\lambda}$ (**step ❸** in Figure 1). To capture the spectral behavior exhibited within each analysis window, we apply overlapping windows. This design is critical to preserve the detection of short-lived or temporally localized causal effects. To maximize sensitivity, the stride between consecutive windows can be fixed at 1, so that each new window shifts by a single time step. This dense sampling guarantees that even subtle or rapid changes in the spectral parameters $(a, \lambda)$ are preserved in the constructed feature time series. The window length is selected adaptively to balance two competing requirements: it must be short enough to capture temporal variations in the spectral parameters, yet long enough to ensure a reliable estimation of the power-law behavior. To meet this trade-off, we evaluate the $p$-value of a Wald test on the linear fit in log-log space for each candidate window size, and select the shortest window for which the fit achieves a statistical significance threshold of $p = 0.05$. Further details of this procedure are provided in the Appendix. Once the feature series $(\mathbf{a}, \boldsymbol{\lambda})$ are built for a couple of original signals, we perform multivariate Granger causality tests, as described in Section 2.2 (**step ❹** in Figure 1). Since the causal information is primarily encoded in the $\lambda$ parameter, the Granger test is applied to assess whether $(\boldsymbol{\lambda}_i, \mathbf{a}_i)$ of the candidate causing series $x_i$ provide statistically significant information about the dynamics of $\boldsymbol{\lambda}_j$ in the target series $x_j$ (see Appendix for further details). In the end, a causal edge is retained in the resulting graph if the corresponding $p$-value falls below the fixed threshold of 0.05. This procedure is repeated across all variable pairs to reconstruct the full causal graph, as detailed in Algorithm 1.

### 3.1. Invariance of the Causal Graph under Spectral Feature Mapping

Unlike conventional Granger methods, which analyze lagged relationships in the original signal space, our approach infers causality from the coordinated evolution of spectral properties.

Moreover, the spectral fitting acts as a natural denoising step, improving robustness to non-Gaussian fluctuations and high-frequency noise. In the following Theorem 3.1, we discuss the correctness of this approach by showing that the causal graph of a stochastic process is invariant under the spectral transformation applied in Algorithm 1, which preserves the causal semantics of the original process.

**Algorithm 1 PLACY**

**Input:** Time series $\mathbf{x} = (\mathbf{x}_1, \ldots, \mathbf{x}_d)$ of length $L$; stride $s$; window size $l$.

**Output:** Causal Graph $\mathcal{G}$.

Divide each $\mathbf{x}_i$ into $\lfloor \frac{L-l}{s} \rfloor + 1$ sliding windows, namely $\mathbf{w}_i^k$, of size $l$ with stride $s$.

**for** each $i \in \{1, \ldots, d\}$ **do**
  **for** each $k \in \{0, \ldots, \lfloor \frac{L-l}{s} \rfloor\}$ **do**
    Apply the DFT (Equation (1)) to $\mathbf{w}_i^k$ to get $\phi_i^k$.
    Obtain $(a_i^k, \lambda_i^k)$ by using the fit in Equation (2) on $\phi_i^k$.
  **end**
  Concatenate $(a_i^k, \lambda_i^k)$ over $k$ to obtain time series $(\mathbf{a}_i, \boldsymbol{\lambda}_i)$.
**end**
**for** each $i, j \in \{1, \ldots, d\}$ such that $i \neq j$ **do**
  $\mathcal{G}_{i,j} \leftarrow$ Granger Causality test 2.2 with $(\mathbf{a}_i, \boldsymbol{\lambda}_i)$ as causing series and $\boldsymbol{\lambda}_j$ as caused series.
**end**
Optionally adjust the resulting $p$-values using FDR control.[1]
**return** $\mathcal{G}$.

**Theorem 3.1** (Preservation of Linear Causal Graphs under Spectral Transformations). *Let $x$ be a multivariate time series generated by a linear structural causal process with ground-truth causal graph $G^*$. Let $T$ be the spectral transformation in Algorithm 1, which extracts for each component $x_i$ a sequence of time-evolving spectral features $(a_i, \lambda_i)$.*

*Suppose that:*

- *within each analysis window, each component admits a local power-law spectral approximation of the form $A(f) = e^a \cdot f^{-\lambda}$;*

- *the process is weakly stationary within each analysis window;*

- *the underlying causal mechanisms are linear;*

- *the noise is additive and does not dominate the signal;*

- *causal dependencies are identifiable from amplitude-spectral dynamics;*

- *the transformed feature process satisfies the standard assumptions required for valid VAR-based Granger causal inference;*

---

[1]For the main experimental results presented in this paper, standard FDR (Benjamini & Hochberg, 1995) correction is not applied, in order to ensure comparability with the other algorithms and because our analysis focuses on individual link performance. Results with FDR correction are reported in Appendix E.8.

*Then causal discovery performed on the transformed feature sequence $(a, \lambda)$ recovers the same causal graph $G^*$ as causal discovery performed in the original time domain.*

*Proof sketch.* The proof proceeds in two steps. First, we show that, under the stated assumptions, the feature series $(\mathbf{a}, \boldsymbol{\lambda})$ satisfies the classical conditions required for valid inference in vector autoregressive (VAR) models, as established in (Lütkepohl, 2005). Second, we show that, for amplitude-expressive causal mechanisms, the mapping $\mathcal{T}$ preserves the causal dependencies encoded in the ground-truth graph $\mathcal{G}^*$. Combining these two results, we conclude that applying Granger causality analysis to the transformed sequence $(\mathbf{a}, \boldsymbol{\lambda})$ recovers the same causal graph $\mathcal{G}^*$ as time-domain analysis. The complete formal proof of Theorem 3.1 is provided in the Appendix. $\square$

## 4. Related Work

Causal discovery from observational time series has been extensively studied (Gong et al., 2024; Assaad et al., 2022), leading to a broad spectrum of methods, from classical statistical tests to more advanced machine learning and spectral approaches. We review many of them here, highlighting in **bold** the ones we compare against in this work. Moving beyond **Granger Causality** described in Section 2.2, constraint-based methods have been developed and adapted for the temporal domain. Notably, the PC (Peter–Clark) algorithm (Spirtes et al., 2000) (and its extension, FCI (Strobl et al., 2018)) serves as the foundation for several approaches. These utilize conditional independence tests to infer graphical causal structures while accounting for temporal ordering. Building on these, the **PCMCI** algorithm (Runge et al., 2019) enhances causal discovery in time series by combining the PC methodology with the Momentary Conditional Independence (MCI) test, which rigorously controls for autocorrelation and indirect associations. This algorithm was recently extended with **PCMCI$_\Omega$** (Gao et al., 2023) to the case of semi-stationary structural causal models. Optimization-based and deep-learning approaches have further broadened the field. **DYNOTEARS** (Pamfil et al., 2020) casts causal discovery as a continuous optimization problem subject to acyclicity constraints, preserving efficiency in handling high-dimensional data. **Rhino** (Gong et al., 2023) represents an innovative deep learning-based approach where the CD task is addressed in scenarios where the noise distributions may depend on historical information. On the one hand, the use of neural networks allows Rhino to handle history-dependent and non-stationary noise; on the other hand, this advantage comes at the cost of significant computational overhead during training. Other techniques, such as Convergent Cross Mapping (CCM) (Sugihara et al., 2012), although robust in theory, exhibit significant performance degradation in noisy settings.

Recent efforts have been made to enhance the noise-resilience of existing algorithms. **CCM-Filtering** (**CCM**) (Zhang et al., 2024) improves the performance of CCM by simply pre-processing the time series with an averaging filter. **RCV-VarLiNGAM** (**RCV**) (Yu et al., 2024) integrates the $K$-fold cross-validation technique with the VarLiNGAM method (Hyvärinen et al., 2010), addressing the challenges of a lack of noise robustness encountered in the standard method.

Previous works have also studied causal discovery in the frequency domain. **Geweke**'s seminal work (Geweke, 1982) extended Granger causality by decomposing directional influence across frequencies, revealing dynamic interdependencies often hidden in time-domain analyses. Subsequent studies applied this methodology to diverse domains, including economic cycles and oscillatory phenomena (Breitung & Candelon, 2006), network and finance (Wang et al., 2021), commodity markets (Ashley & Verbrugge, 2008), and market volatility (Maitra & Dash, 2017). Among frequency-domain approaches, the **Directed Transfer Function (DTF)** (Kamiński & Blinowska, 1991; Korzeniewska et al., 2003) quantifies directional interactions as a function of frequency through the multivariate VAR transfer function. A **nonparametric spectral Granger (GewekeNP)** variant computes Geweke's frequency-domain causality measure directly from the empirically estimated power spectra, instead of the canonical VAR-based derivation (Welch, 1967; Theiler et al., 1992). We also implement the **BCGeweke** system used by Wang et al. (Wang et al., 2021), which combines Geweke's frequency decomposition with the Breitung–Candelon (Breitung & Candelon, 2006) band-constrained statistical testing framework to detect directional dependencies within specific frequency bands.

Despite these advancements, many current methods remain vulnerable to noise, spurious dependencies, and deviations from Gaussianity. Motivated by these limitations, this paper proposes a novel frequency-domain strategy designed explicitly for robust causal discovery within stochastic power-law processes. Our approach inherently leverages the frequency-dependent structure of power-law processes, enhancing resilience to nonlinear, complex, noisy signals.

# 5. Experiments

Unless otherwise stated, our method employs a sliding window of size $l = 50$, selected through the $p$-value procedure outlined in Section 3, a stride $s = 1$ (refer to Algorithm 1), and 10 lagged values. Quantitative results are averaged over 100 runs for all methods, except for Rhino, which is averaged over only 10 random seeds due to the high computational cost of neural network training.

We consider both synthetic and real-world datasets to rig-orously evaluate our approach in terms of its robustness to noise and spurious associations. In particular, we create four synthetic datasets with increasing complexity, and we consider two real-world benchmark datasets with known causal graphs. Some prior datasets were excluded due to the absence of ground-truth causal graph or insufficient time series length for spectral estimation (see Section 6).

The code for all experiments is publicly available [2].

**Metrics**  We evaluate the performance of the algorithms based on their ability to accurately identify causal relationships among variables. By comparing the edges of the predicted causal graph with those of the ground-truth graph, we compute the following metrics: *F1-score* (F1) that measures the performance of algorithms in correctly identifying causal relationships; and *True Negative Rate* (TNR) to evaluate the robustness of the algorithms to noise and spurious associations, by measuring its ability to correctly identify the absence of causal links. The latter metric is particularly insightful, as it evaluates a method's ability to exclude erroneous relationships in the generated causal graphs, an aspect not directly captured by the F1-score. Notice that our main experiments report uncorrected pairwise significance tests to maintain comparability with prior work. However, False Discovery Rate (FDR) (Benjamini & Hochberg, 1995) correction can be readily applied; supplementary results in Appendix E.8 confirm that both method rankings and relative performance remain unchanged under this correction.

## 5.1. Synthetic Scenarios

**Data Generation**  To generate complex benchmark datasets, we use the well-known Ornstein-Uhlenbeck (OU) processes, originally introduced in statistical physics to describe the velocity of a Brownian particle under friction (Uhlenbeck & Ornstein, 1930). These stochastic processes are widely used to model systems exhibiting mean-reverting behavior and power-law spectral characteristics. For example, they have been applied to capture the complexity of financial data (Maller et al., 2009). In the frequency domain, OU processes exhibit a characteristic power-law decay, reflecting the behavior of various natural systems that our approach seeks to address. We simulate the baseline dynamics of each time series using a generalized OU process, defined as follows:

$$x(t+\Delta t) = x(t) + \frac{\Delta t}{\tau_c}\big(\mu - x(t)\big) \tag{3}$$
$$+ \Big(\sigma_b \epsilon_b(t) + \sigma_g^a \epsilon_g^a(t) + \sigma_g^m \epsilon_g^m(t) \cdot x(t)\Big)\sqrt{\Delta t},$$

where $\Delta t = 0.01$ is the time step, $\tau_c = 0.5$ denotes the timescale of mean reversion, and $\mu = 1$ is the long-term

---

[2] https://github.com/matteotusoni/PLACy

mean. $\forall t$, $\epsilon_b(t)$, $\epsilon_g^a(t)$, and $\epsilon_g^m(t)$ represent the noise terms modeled as independent stochastic variables: the first is Brownian noise, while the latter two are standard Gaussian white noise processes. Specifically, $\epsilon_g^a(t)$ is an additive noise component, whereas $\epsilon_g^m(t)$ acts as a multiplicative noise term that induces non-stationarity as its impact scales with the process value. The parameters $\sigma_b$, $\sigma_g^a$, and $\sigma_g^m$ represent different sources of noise volatility. This formulation enables the system to capture both additive and multiplicative stochastic effects, which are common in real-world dynamic systems. To represent causal relationships between different time series, we construct a Directed Acyclic Graph (DAG) $\mathcal{G}$, represented by an upper triangular adjacency matrix ($M \in \{0,1\}^{N \times N}$), which enforces a unidirectional flow of causality and prevents cycles. Each entry in the matrix is randomly set to 1 with a probability of 0.3, indicating a causal influence from one series to another.

Finally, causal dependencies are introduced by applying the generated ground truth causal matrix to the time series. In particular, if $M_{i,j} = 1$, indicating that series $i$ influences series $j$, then

$$\forall t, \; x_j(t) \leftarrow x_j(t) + C \cdot x_i(t - \tau),$$

where $C$ represents the *causal strength*, and $\tau = 5$ is the number of time-steps between the cause and the effect. This ensures that the current value of the influenced series incorporates a lagged contribution from the influencing series. Finally, we scale all time series to their original range to remove any unintended amplifications or distortions during the causal injection.

**Datasets** Following the generation process described in the previous paragraph, we define four representative scenarios to evaluate the robustness of our method under different dynamic conditions according to Equation (3): **(1)** $\text{OU}(\sigma_g^m = 0)$ represents an OU process with no noise component proportional to the process itself; **(2)** $\text{OU}(\sigma_g^m > 0)$ includes a Gaussian noise term that is proportional to the current value of the process. Both processes are initialized in equilibrium conditions, with the first time step $t = 0$ set to 1. Non-equilibrium and phase-transitioning systems display complex behaviors, as observed in several domains (e.g., in financial markets (Ang & Timmermann, 2012)). Therefore, we introduce a transition phase by initializing the process at 100. By extending the previous two scenarios, we obtain **(3)** $\widehat{\text{OU}}(\sigma_g^m = 0)$ and **(4)** $\widehat{\text{OU}}(\sigma_g^m > 0)$. For each dataset, we generate different scenarios with $N = 5$ and $N = 10$ variables (i.e., time series), each of length $L = 5000$ time-steps.

**Results** Figure 2 reports the F1-score and the TNR obtained on synthetic datasets with $N = 5$ variables, causal strength $C = 0.5$ and $\sigma_g^a = 0$.

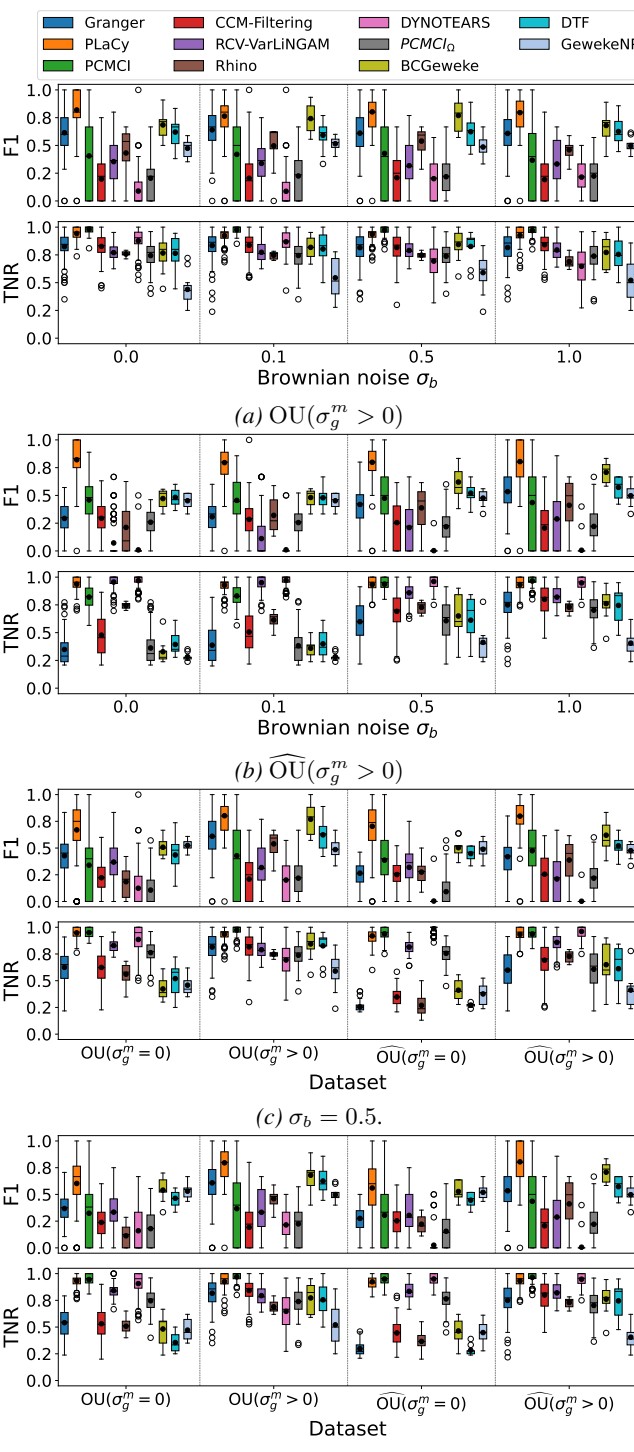

*Figure 2.* Results on synthetic datasets, with $N = 5$, $C = 0.5$, $\sigma_g^a = 1.0$.

A complete overview of the results can be found in Table 1 for the case of $N = 5$ and $\sigma_g^a = 1$, averaging over $\sigma_b \in \{0, 0.1, 0.5, 1\}$. Additional results, including scenarios with $N = 10$ and $\sigma_g^a = 0.5$, can be found in the Appendix. Results obtained with the basic **Geweke** method

have been excluded from Figure 2, Tables 2 and 5 due to its consistently poor performance. In particular, the *True Negative Rate* (TNR) across the same experimental settings is uniformly zero, as the method fails to reject any spurious connections.

**Key takeaway:** Our method consistently outperforms existing approaches across all scenarios, demonstrating strong robustness to both structural variations and noise. In particular, Figures 2a and 2b show that our approach, **PLaCy**, achieves overall the best performance in terms of F1-score for all the noise settings $\sigma_b$, for $\mathrm{OU}(\sigma_g^m > 0)$ and $\widehat{\mathrm{OU}}(\sigma_g^m > 0)$. In fact, the presence of multiplicative Gaussian noise introduces non-stationarity, to which other methods are highly sensitive. In contrast, analyzing causal dynamics via spectral parameters enables **PLaCy** to filter variability from multiplicative noise, improving robustness. Our approach outperforms the other methods even in the absence of multiplicative noise, for $\sigma_b = 0.5$ and $\sigma_b = 1$ (Figures 2c and 2d): while other methods heavily rely on stationary assumptions, **PLaCy** can capture structural shifts in spectral parameters, identifying genuine causal relationships without confusing transient behaviors for structural causal patterns.

In all the experiments in Figure 2, the TNR of **PLaCy** is always very high, sometimes moderately outperformed by **PCMCI**. This is due to the fact that **PCMCI** is designed to explicitly control false positives, with a conservative approach to edge selection (MCI phase). **PLaCy**, however, shows a more permissive behavior in edge inclusion. This causes occasional acceptance of marginal causal association, but it is also the key to capturing genuine causal relations, as confirmed by the higher F1-score. Figure 2b considers the impact of both multiplicative noise and non-equilibrium initialization, two aspects that constitute a significant challenge for traditional CD approaches. In this setting, **PLaCy** significantly outperforms the other methods, with the highest F1-score and TNR close to 1, thanks to its capability to distinguish meaningful causal perturbations in spectral trends from spurious correlations possibly due to transient non-stationary dynamics.

**PCMCI**'s lack of robustness in the non-Gaussian and non-stationary noise scenario is evident from its low F1-score. The same limitations apply to its generalized version, **PCMCI$_\Omega$**, designed to handle semi-stationary causal relationships.

Regarding the other methods, although **Granger causality** is not designed for nonstationary data or non-Gaussian noise, it often retains moderate effectiveness in inferring causal relationships in such settings. Even if **CCM-Filtering** is designed to improve the robustness of the original CCM by reducing high-frequency noise and preserving lower-frequency signals, this solution does not generalize to stochastic processes, where delay embeddings fail to capture a deterministic manifold. **RCV-VarLiNGAM** does not show good results in a non-stationary noise environment, as expected by its assumption of stationarity. Although designed to uncover causal relationships via exploitation of non-Gaussian noise, the improvement with respect to other models remains limited. **DYNOTEARS** is designed to handle additive noise, but in our experiments, it struggles under multiplicative perturbations and strong non-stationarity. As a result, the method tends to miss true causal links, despite maintaining a high TNR. **Rhino** achieves moderate performance as proved by the experiments in Figures 2a and 2b, validating the authors' claim that the method is robust to increasing non-Gaussian noise. However, its computational complexity does not justify the lower performance compared to our approach, which also preserves sample efficiency. Finally, the three frequency-domain algorithms **BCGeweke**, **DTF**, and **GewekeNP**, achieve good F1 scores throughout the experimental campaign, suggesting that exploiting spectral properties provides a valuable avenue for detecting causal dependencies. Nevertheless, their high F1 score comes at the expense of a low TNR. Remarkably, PLaCy not only surpasses these methods in terms of F1 accuracy, but also mitigates their TNR weakness, achieving a more balanced performance.

### 5.2. Real Data Scenarios

As real-world scenarios, we consider two datasets with known causal graphs:

• **Rivers** dataset[3] (Ahmad et al., 2022) contains $N = 6$ time series from three hydrological stations located in southern Germany, namely Dillingen, Kempten, and Lenggries. At each location, river level and local precipitation are recorded over several thousand time steps. Since the Iller feeds into the Danube, increases in its discharge are expected to impact the Danube with a one-day lag.

• **AirQuality** (**AQI**) dataset[4] contains hourly PM2.5 pollution measurements collected over one year from $N = 36$ monitoring stations across various Chinese cities. Following the benchmark protocol of CausalTime (Cheng et al., 2024), we use the provided reference causal graph, whose causal matrix is constructed from pairwise sensor distances.

We extract sub-samples of length 500 from the original time series.

**Results**   Table 2 reports the results for both datasets. Our method achieves overall competitive or the best performance in terms of F1-score and TNR for both datasets.

---

[3]Bavarian Environmental Agency data provider: https://www.gkd.bayern.de.

[4]Microsoft data provider: https://www.microsoft.com/en-us/research/project/urban-computing.

*Table 1.* F1 and TNR Score - $N = 5$, $\sigma_g^a = 1.0$

| Dataset | C | Score | Granger | PLACy | PCMCI | RCV-VarLiNGAM | DYNOTEARS | PCMCI$_\Omega$ | BCGeweke | DTF | GewekeNP |
|---|---|---|---|---|---|---|---|---|---|---|---|
| OU($\sigma_g^m = 0$) | 0.2 | F1 | 0.58±0.26 | 0.14±0.22 | 0.15±0.24 | 0.40±0.19 | 0.17±0.18 | 0.07±0.13 | **0.61**±0.18 | 0.35±0.18 | 0.59±0.06 |
| | | TNR | 0.75±0.19 | 0.96±0.04 | **0.97**±**0.04** | 0.76±0.08 | 0.69±0.24 | 0.78±0.09 | 0.57±0.24 | 0.70±0.25 | 0.60±0.10 |
| | 0.5 | F1 | 0.56±0.25 | **0.73**±0.24 | 0.20±0.27 | 0.39±0.19 | 0.08±0.18 | 0.11±0.14 | 0.58±0.15 | 0.52±0.18 | 0.52±0.07 |
| | | TNR | 0.73±0.20 | 0.95±0.05 | **0.97**±**0.04** | 0.79±0.08 | 0.93±0.10 | 0.76±0.11 | 0.54±0.23 | 0.67±0.21 | 0.46±0.13 |
| | 1.0 | F1 | 0.53±0.25 | **0.77**±0.17 | 0.28±0.29 | 0.32±0.21 | 0.08±0.17 | 0.13±0.15 | 0.56±0.15 | 0.54±0.13 | 0.52±0.08 |
| | | TNR | 0.70±0.22 | 0.91±0.07 | **0.96**±**0.04** | 0.81±0.10 | 0.91±0.11 | 0.75±0.12 | 0.51±0.23 | 0.55±0.20 | 0.47±0.13 |
| $\widetilde{\text{OU}}(\sigma_g^m = 0)$ | 0.2 | F1 | 0.47±0.28 | 0.24±0.25 | 0.16±0.25 | 0.41±0.19 | 0.02±0.10 | 0.08±0.13 | **0.58**±0.18 | 0.44±0.07 | 0.49±0.10 |
| | | TNR | 0.55±0.30 | 0.93±0.06 | 0.96±0.05 | 0.76±0.07 | **0.98**±**0.03** | 0.78±0.09 | 0.52±0.25 | 0.27±0.03 | 0.38±0.16 |
| | 0.5 | F1 | 0.45±0.27 | **0.69**±0.22 | 0.21±0.26 | 0.37±0.20 | 0.01±0.06 | 0.10±0.14 | 0.55±0.16 | 0.44±0.07 | 0.46±0.07 |
| | | TNR | 0.54±0.29 | 0.92±0.06 | 0.97±0.04 | 0.79±0.08 | **0.98**±**0.04** | 0.76±0.10 | 0.48±0.23 | 0.27±0.03 | 0.32±0.10 |
| | 1.0 | F1 | 0.45±0.26 | **0.70**±0.17 | 0.29±0.30 | 0.32±0.19 | 0.01±0.06 | 0.13±0.15 | 0.54±0.16 | 0.43±0.09 | 0.46±0.09 |
| | | TNR | 0.54±0.29 | 0.88±0.09 | 0.96±0.05 | 0.81±0.09 | **0.98**±**0.05** | 0.75±0.12 | 0.48±0.22 | 0.26±0.02 | 0.33±0.12 |
| OU($\sigma_g^m > 0$) | 0.2 | F1 | 0.63±0.21 | 0.72±0.24 | 0.23±0.29 | 0.39±0.18 | 0.22±0.14 | 0.16±0.16 | **0.79**±0.11 | 0.50±0.16 | 0.51±0.08 |
| | | TNR | 0.87±0.09 | 0.96±0.05 | **0.98**±**0.03** | 0.76±0.08 | 0.52±0.20 | 0.76±0.11 | 0.88±0.09 | 0.91±0.07 | 0.64±0.12 |
| | 0.5 | F1 | 0.62±0.20 | **0.80**±0.17 | 0.40±0.33 | 0.34±0.20 | 0.15±0.17 | 0.22±0.16 | 0.74±0.11 | 0.64±0.14 | 0.50±0.07 |
| | | TNR | 0.82±0.13 | 0.93±0.06 | **0.97**±**0.04** | 0.78±0.08 | 0.77±0.17 | 0.74±0.12 | 0.79±0.13 | 0.79±0.12 | 0.48±0.17 |
| | 1.0 | F1 | 0.57±0.18 | **0.77**±0.18 | 0.60±0.30 | 0.28±0.21 | 0.17±0.18 | 0.23±0.18 | 0.69±0.14 | 0.59±0.12 | 0.51±0.08 |
| | | TNR | 0.78±0.16 | 0.91±0.09 | **0.97**±**0.05** | 0.81±0.09 | 0.75±0.16 | 0.73±0.12 | 0.73±0.17 | 0.69±0.18 | 0.49±0.15 |
| $\widetilde{\text{OU}}(\sigma_g^m > 0)$ | 0.2 | F1 | 0.39±0.20 | **0.75**±0.21 | 0.32±0.27 | 0.22±0.23 | 0.02±0.10 | 0.22±0.14 | 0.61±0.17 | 0.46±0.13 | 0.46±0.08 |
| | | TNR | 0.55±0.24 | 0.95±0.05 | 0.89±0.10 | 0.89±0.10 | **0.96**±**0.05** | 0.53±0.20 | 0.58±0.22 | 0.61±0.22 | 0.40±0.17 |
| | 0.5 | F1 | 0.39±0.18 | **0.80**±0.17 | 0.46±0.26 | 0.17±0.21 | 0.01±0.05 | 0.24±0.14 | 0.57±0.13 | 0.52±0.11 | 0.46±0.08 |
| | | TNR | 0.52±0.23 | 0.93±0.06 | 0.89±0.10 | 0.90±0.09 | **0.96**±**0.05** | 0.51±0.21 | 0.54±0.18 | 0.54±0.20 | 0.35±0.12 |
| | 1.0 | F1 | 0.39±0.18 | **0.78**±0.17 | 0.55±0.25 | 0.15±0.20 | 0.01±0.07 | 0.26±0.14 | 0.54±0.13 | 0.52±0.10 | 0.46±0.08 |
| | | TNR | 0.51±0.24 | 0.91±0.08 | 0.89±0.10 | 0.91±0.09 | **0.96**±**0.06** | 0.51±0.20 | 0.51±0.18 | 0.48±0.18 | 0.35±0.13 |

*Table 2.* Performance on real-world datasets.

| Algorithm | Rivers | | AirQuality | |
|---|---|---|---|---|
| | F1 | TNR | F1 | TNR |
| **Granger** | 0.47±0.07 | 0.64±0.09 | 0.41±0.02 | 0.22±0.05 |
| **PLaCy** | **0.51**±0.10 | **0.75**±0.13 | **0.45**±0.04 | 0.66±0.07 |
| **PCMCI** | 0.47±0.07 | 0.74±0.05 | 0.25±0.03 | **0.95**±0.02 |
| **CCM** | 0.28±0.01 | 0.19±0.00 | 0.40±0.00 | 0.04±0.00 |
| **RCV** | 0.16±0.12 | 0.51±0.12 | — | — |
| **Rhino** | 0.29±0.03 | 0.35±0.05 | 0.44±0.01 | 0.23±0.04 |
| **DYNO.** | 0.12±0.07 | 0.53±0.06 | 0.37±0.08 | 0.92±0.03 |
| **PCMCI$_\Omega$** | 0.10±0.09 | 0.57±0.07 | 0.36±0.04 | 0.69±0.07 |
| **BCGeweke** | 0.41±0.06 | 0.61±0.08 | 0.39±0.02 | 0.16±0.07 |
| **DTF** | 0.36±0.05 | 0.42±0.12 | 0.43±0.02 | 0.36±0.09 |
| **GewekeNP** | 0.26±0.05 | 0.31±0.06 | 0.40±0.02 | 0.13±0.07 |

**Key takeaway:** The Rivers dataset poses an additional challenge due to its heterogeneous dynamics and the presence of exogenous factors such as seasonal precipitation. Because precipitation series lack clear power-law behavior, this dataset highlights PLaCy's ability to generalize beyond its core assumptions. Indeed, our method achieves robust performance and surpasses the other methods in detecting the causal effects of precipitation and rivers' flow in both F1 and TNR. On the other hand, the AirQuality dataset includes missing values, which were filled using linear interpolation. Missing data, common in environmental datasets, challenges most CD methods. **PLaCy** maintains competitive performance despite these imperfections, highlighting its robustness to real-world data issues thanks to the advantages of performing causal discovery in the frequency domain, which is inherently more resilient to missing data and noise. **RCV-VarLiNGAM**, failed to converge. In our tests, we observe that this behavior is due to the impossibility of running the Cholesky decomposition on the residual covariance matrix, which results in being non–positive definite as a consequence of missing data.

Despite the original CCM promises to be robust in correctly excluding causal links between non-coupled variables in the presence of external forcing, the performance of **CCM-Filtering**, and in particular the achieved TNR, degrades on more complex data, influenced by unobserved exogenous factors. The same considerations mentioned about the TNR attained by **PCMCI** on the synthetic datasets (Section 5.1) hold on AirQuality, where the high TNR is achieved at the expense of the lowest F1.

Interestingly, on the AirQuality dataset, **BCGeweke** exhibits a comparatively low TNR. This limitation likely stems from the algorithm's design: it analyzes distinct sub-bands of the spectral domain and infers causality whenever at least one sub-band yields a positive result in the Breitung–Candelon (BC) causal test. In this case study, the method likely produced false positives within one of these sub-bands, leading to an overall degradation in TNR.

## 6. Limitations

Despite the strong performance demonstrated in the experimental campaign described in Section 5, **PLaCy** presents some limitations that should be acknowledged. First, we recall that **PLaCy** is not intended as a universal causal discovery method, but rather as a specialized approach that is effective for systems exhibiting scale-free spectral structure. Moreover, our method struggles to correctly identify causal relationships in the presence of slowly varying spectra. Some strategies may improve the algorithm's performance in such scenarios, such as increasing the number of lags considered in the causal analysis or extending the length of the analysis window (see Appendix). Nevertheless, in these cases, time-domain analyses like Granger causality may be more appropriate. While this represents a limitation, a reasonable choice of the causal analysis method can be guided by a preliminary spectral inspection. Finally, the method is not well-suited for very short time series, as it relies on local spectral estimation, which requires a minimum sequence length to produce stable features.

## 7. Conclusions

This study introduces **PLaCy**, a novel algorithm for causal discovery in stochastic time series, leveraging power spectrum analysis to identify underlying causal structures. An extensive experimental campaign on both synthetic and real-world datasets reveals the effectiveness of this methodology in comparison to state-of-the-art methods. Our findings underscore the advantages of frequency-domain analysis for causal discovery, highlighting its potential to avoid detecting spurious associations as causal relationships while maintaining high F1 scores. Future work will focus on extending the idea of exploiting the power spectrum to enhance non-VAR causal discovery methods. A deeper investigation should be conducted on the summary statistical parameters of the spectral density. Furthermore, an additional study must be conducted to address the possible presence of latent confounders.

## Impact Statement

This paper presents work whose goal is to advance the field of machine learning. There are many potential societal consequences of our work, none of which we feel must be specifically highlighted here.

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

## A. Robust Causal Discovery in Real-World Time Series with Power-Law: Supplementary Materials

## B. Theoretical Analysis

The proof of Theorem 3.1 is articulated in two main steps:

**(A)** We first show that the feature sequence $(\mathbf{a}, \boldsymbol{\lambda})$ satisfies the classical assumptions required for valid inference using vector autoregressive (VAR) models, as established in (Lütkepohl, 2005).

**(B)** We then demonstrate that the transformation $\mathcal{T}$, described in Algorithm 1, preserves the structure of the ground-truth causal graph $\mathcal{G}^*$, meaning that no spurious edges are introduced and no genuine dependencies are lost.

By combining these two results, we conclude that applying Granger causality analysis to the transformed sequence $(\mathbf{a}, \boldsymbol{\lambda})$ enables consistent recovery of the original causal graph $\mathcal{G}^*$.

In order to validate step **(A)**, we decompose it into the following supporting claims:

**(A1) Weak Stationarity and Noise Assumptions:** The sequences $(\mathbf{a}, \boldsymbol{\lambda})$ are approximately weakly stationary and with an asymptotically Gaussian noise.

**(A2) Preservation of Linear Dependence under Spectral Transformation:** The transformation $\mathcal{T}$, described in Algorithm 1, preserves linear relationships between corresponding time series. That is, if two time series are linearly dependent (e.g., one is a scalar multiple of the other), then their corresponding spectral features remain linearly dependent. Conversely, if the time series are not collinear in the time domain, their spectral representations will also be linearly independent.

Step **(B)** relies on just one structural condition:

**(B1) Spectral Causality Preservation:** Causal relationships in the time domain that manifest through changes in spectral amplitude induce dependencies among the corresponding spectral features.

Step **(A1)** establish the key requirements needed to apply a VAR analysis for studying Granger causality.

Theorem B.1 discusses the assumptions of weak stationarity and Gaussianity of the noise. It shows that the procedure $\mathcal{T}$, described in Algorithm 1, transforms any colored noise affecting $\mathbf{x}$ into an asymptotically Gaussian noise on $(\mathbf{a}, \boldsymbol{\lambda})$. This proves step **(A1)**.

Theorem B.2 addresses the component **(A2)**, proving the preservation of linear dependencies under the transformation $\mathcal{T}$.

Finally, Theorem B.3 concludes the proof showing that causal relationships are preserved in the transformed time series $(\mathbf{a}, \boldsymbol{\lambda})$ **(B1)**.

**(A1) Weak Stationarity and Noise Assumptions**

**Theorem B.1** (Asymptotic Gaussianity and Stationarity under Colored Noise). *Let* $\mathbf{x} = \{x(t)\}$ *be a weakly stationary process with power spectral density* $S(f_k) \propto 1/f_k^\alpha$ *for* $\alpha \geq 0$, *where* $f_k$ *denotes the $k$-th frequency component. Let* $\phi(k)$ *be the DFT over a window of size $l$, and define* $(\mathbf{a}, \boldsymbol{\lambda})$ *as in Algorithm 1. Then, for sufficiently large $l$, the sequences* $(\mathbf{a}, \boldsymbol{\lambda})$ *are approximately Gaussian and weakly stationary.*

*Proof.* As shown in (Brillinger, 1981), if $\mathbf{x}$ is a weakly stationary process with decaying autocorrelation, then the Discrete Fourier Transform (DFT) coefficients computed over a window of size $l$ converge in distribution to complex Gaussian variables with variance given by the power spectral density $S(f_k)$. That is,

$$\phi_k \xrightarrow{d} \mathcal{CN}(0, S(f_k)).$$

The spectral amplitudes $A_k = |\phi_k|$ follow a Rayleigh distribution with scale parameter determined by $S(f_k)$. Let $(\mathbf{a}, \boldsymbol{\lambda})$ be computed as in Algorithm 1, by linear regression on the pairs $(\log f_k, \log A_k)$ within each window. Although the $\log A_k$ are

not Gaussian, the regression combines information from many such values. Since the estimators are linear combinations of independent (or weakly dependent) inputs with finite variance, they are approximately Gaussian by the Central Limit Theorem.

Therefore, for sufficiently large window length $l$, the estimated features $(\mathbf{a}, \boldsymbol{\lambda})$ can be treated as approximately Gaussian variables.

Regarding the i.i.d. assumption of the noise, this trivially holds in the case of non-overlapping windows. However, when the windows overlap, the input data for consecutive Fourier transforms share common segments, which induces statistical dependence between the resulting spectral estimates. As a consequence, the noise affecting the estimated parameters $(\mathbf{a}, \boldsymbol{\lambda})$ is not independent across windows, but exhibits weak temporal correlation. $\square$

### (A2) Preservation of Linear Dependence under Spectral Transformation

**Theorem B.2** (Preservation of Linear Dependence under Spectral Transformation). *Let $\mathbf{x}_i$ and $\mathbf{x}_j$ be two power-law time series and let $\mathcal{T}$ denote the transformation described in Algorithm 1.*

*Then, if $\mathbf{x}_j = \alpha \cdot \mathbf{x}_i + \beta$ for some constants $\alpha, \beta \in \mathbb{R}$ and all t, then, for $\alpha \neq 0$,*

$$\boldsymbol{\lambda}_j = \boldsymbol{\lambda}_i \qquad and \qquad \boldsymbol{a}_j = \boldsymbol{a}_i + \log |\alpha|.$$

*Therefore, the spectral transformation $\mathcal{T}$ preserves linear dependence between time series.*

*Proof.* Suppose $\mathbf{x}_j = \alpha \cdot \mathbf{x}_i + \beta$. By linearity of the Fourier transform, for each frequency $f$ we have

$$\mathcal{F}[\mathbf{x}_j](f) = \mathcal{F}[\alpha \mathbf{x}_i + \beta](f) = \alpha \mathcal{F}[\mathbf{x}_i](f) + \mathcal{F}[\beta](f).$$

Since $\beta$ is constant in time, its Fourier transform is concentrated only at zero frequency, i.e.,

$$\mathcal{F}[\beta](f) \propto \delta(f).$$

However, in the PLaCy fitting procedure, the zero-frequency component is discarded and the power-law fit is performed only over frequencies $f > 0$. Hence, for all frequencies used in the fit,

$$\mathcal{F}[\mathbf{x}_j](f) = \alpha \mathcal{F}[\mathbf{x}_i](f).$$

Therefore, for $\alpha \neq 0$,

$$A_j(f) = |\mathcal{F}[\mathbf{x}_j](f)| = |\alpha \mathcal{F}[\mathbf{x}_i](f)| = |\alpha| A_i(f).$$

Since

$$A_i(f) = e^{a_i} f^{-\lambda_i},$$

it follows that

$$A_j(f) = |\alpha| e^{a_i} f^{-\lambda_i} = e^{a_i + \log |\alpha|} f^{-\lambda_i}.$$

Taking logarithms gives

$$\log A_j(f) = \log A_i(f) + \log |\alpha|,$$

and therefore

$$-\boldsymbol{\lambda}_j \log f + \boldsymbol{a}_j = -\boldsymbol{\lambda}_i \log f + \boldsymbol{a}_i + \log |\alpha|.$$

Hence,

$$\boldsymbol{\lambda}_j = \boldsymbol{\lambda}_i, \qquad \boldsymbol{a}_j = \boldsymbol{a}_i + \log |\alpha|.$$

If $\alpha = 0$, then $\mathbf{x}_j = \beta$ is a constant signal. Its Fourier transform is concentrated only at $f = 0$, which is discarded before the power-law fitting procedure. Therefore, this degenerate case carries no nonzero-frequency spectral structure and does not affect the analysis.

Hence, $\mathcal{T}$ preserves linear dependence between the transformed spectral features. $\square$

**(B1) Spectral Causality Preservation**

**Theorem B.3** (Preservation of Linear Causal Structure under Spectral Transformation). *Let $\mathbf{x}_i$ and $\mathbf{x}_j$ be two power-law time series such that*

$$\mathbf{x}_i = g(\mathbf{x}_j) + \boldsymbol{\eta}, \tag{4}$$

*where $g$ is a linear function and $\boldsymbol{\eta} = \{\eta(t)\}$ is additive noise independent of $\mathbf{x}_j$. Assume further that, over the nonzero frequencies used in the power-law fit, the noise contribution is dominated by the causal signal, i.e.,*

$$|\mathcal{F}[\boldsymbol{\eta}](f)| \ll |\mathcal{F}[g(\mathbf{x}_j)](f)|, \qquad f > 0.$$

*Then, the spectral parameters $(\mathbf{a}_i, \boldsymbol{\lambda}_i)$ defined in Algorithm 1 retain information about the causal influence from $\mathbf{x}_j$ to $\mathbf{x}_i$.*

*Proof.* Applying the Fourier transform to both sides of Equation (4) and exploiting the linearity of the Fourier transform, we get

$$\mathcal{F}[\mathbf{x}_i](f) = \mathcal{F}[g(\mathbf{x}_j)](f) + \mathcal{F}[\boldsymbol{\eta}](f).$$

Taking the modulus does not, in general, distribute over sums. However, by the assumed signal-dominance condition, for the nonzero frequencies used in the fit we have

$$A_i(f) = |\mathcal{F}[g(\mathbf{x}_j)](f) + \mathcal{F}[\boldsymbol{\eta}](f)| \approx |\mathcal{F}[g(\mathbf{x}_j)](f)|.$$

Since $g$ is linear, the causal component preserves the power-law structure of the parent process up to the spectral effect induced by $g$. Therefore, if

$$A_j(f) \propto f^{-\boldsymbol{\lambda}_j},$$

then the causal component in $A_i(f)$ also carries a power-law contribution induced by $\mathbf{x}_j$. Consequently, the log-log fit of $A_i(f)$ produces spectral parameters $(\mathbf{a}_i, \boldsymbol{\lambda}_i)$ that retain information about the causal transformation from $\mathbf{x}_j$ to $\mathbf{x}_i$.

$\square$

Nonlinear causal relationship $g$. For nonlinear $g$, $\mathcal{F}[g(\mathbf{x}_j)] \neq g(\mathcal{F}[\mathbf{x}_j])$ in general. However, due to the orthogonality and completeness of the Fourier basis, the operation $g$ might induce structured interactions among frequencies. This might distort without destroying the underlying spectral shape. In practice, we found experimentally that even under nonlinear transformations the global decay behavior captured by the spectral parameters $\boldsymbol{\lambda}$ typically remains a meaningful summary of the causal influence, as suggested by the experiments of Appendix E.5.

## C. Mathematical computation of $\lambda$ for our Synthetic Datasets

In this section, we provide a mathematical derivation of the spectral parameter $\lambda$ for the linear additive system used to generate the synthetic datasets in Section 5.1.

Let $\mathbf{x}_1$, $\mathbf{x}_2$, and $\mathbf{x}_3$ be three time series whose spectral amplitudes follow a power-law profile. In the case of an additive interaction in the time domain, the resulting spectral amplitude of $\mathbf{x}_1$ can be expressed—under idealized linearity and independence assumptions—as:

$$\forall f \quad A_1(f) = A_2(f) + c \cdot A_3(f)$$

where $c$ is a scalar coefficient regulating the strength of the contribution from $\mathbf{x}_3$ to $\mathbf{x}_1$, and $f$ the value of a frequency of the spectrum.

$$A_1(f) = e^{\mathbf{a}_1} \cdot f^{-\boldsymbol{\lambda}_1}, \quad A_2(f) = e^{\mathbf{a}_2} \cdot f^{-\boldsymbol{\lambda}_2}, \quad A_3(f) = e^{\mathbf{a}_3} \cdot f^{-\boldsymbol{\lambda}_3}$$

Substituting into the first equation gives:

$$e^{\mathbf{a}_1} \cdot f^{-\boldsymbol{\lambda}_1} = e^{\mathbf{a}_2} \cdot f^{-\boldsymbol{\lambda}_2} + c \cdot e^{\mathbf{a}_3} \cdot f^{-\boldsymbol{\lambda}_3} \tag{5}$$

**Case 1:** Assume the following:

$$\frac{c \cdot e^{\mathbf{a_3}}}{e^{\mathbf{a_2}}} \cdot f^{\boldsymbol{\lambda_2} - \boldsymbol{\lambda_3}} \gg 1. \tag{6}$$

Notice that we can write the right-hand side of Equation (5) as $e^{\mathbf{a_2}} \cdot f^{-\boldsymbol{\lambda_2}} \left[ 1 + \frac{c \cdot e^{\mathbf{a_3}}}{e^{\mathbf{a_2}}} \cdot f^{\boldsymbol{\lambda_2} - \boldsymbol{\lambda_3}} \right]$. By applying this substitution, taking the natural logarithm in Equation (5), and using logarithmic properties, we get:

$$\log \left( e^{\mathbf{a_1}} \cdot f^{-\boldsymbol{\lambda_1}} \right) = \log \left( e^{\mathbf{a_2}} \cdot f^{-\boldsymbol{\lambda_2}} \left[ 1 + \frac{c \cdot e^{\mathbf{a_3}}}{e^{\mathbf{a_2}}} \cdot f^{\boldsymbol{\lambda_2} - \boldsymbol{\lambda_3}} \right] \right) \tag{7}$$

$$= \mathbf{a_2} - \boldsymbol{\lambda_2} \log f + \log \left( 1 + \frac{c \cdot e^{\mathbf{a_3}}}{e^{\mathbf{a_2}}} \cdot f^{\boldsymbol{\lambda_2} - \boldsymbol{\lambda_3}} \right) \tag{8}$$

Given the assumption in Equation (6), the logarithmic term in the previous equation can be approximated as:

$$\log \left( 1 + \frac{c \cdot e^{\mathbf{a_3}}}{e^{\mathbf{a_2}}} \cdot f^{\boldsymbol{\lambda_2} - \boldsymbol{\lambda_3}} \right) \approx \log \left( \frac{c \cdot e^{\mathbf{a_3}}}{e^{\mathbf{a_2}}} \cdot f^{\boldsymbol{\lambda_2} - \boldsymbol{\lambda_3}} \right)$$

By substituting this expression in Equation (7), we obtain the following equation:

$$\mathbf{a_1} - \boldsymbol{\lambda_1} \log f \approx \mathbf{a_2} - \boldsymbol{\lambda_2} \log f + \log \left( \frac{c \cdot e^{\mathbf{a_3}}}{e^{\mathbf{a_2}}} \right) + (\boldsymbol{\lambda_2} - \boldsymbol{\lambda_3}) \log f$$

Which can be simplified as follows:

$$\mathbf{a_1} - \boldsymbol{\lambda_1} \log f \approx \log c + \mathbf{a_3} - \boldsymbol{\lambda_3} \log f$$

Finally, by solving for $\boldsymbol{\lambda_1}$, we get the following expression for $\boldsymbol{\lambda_1}$:

$$\boldsymbol{\lambda_1} \approx \boldsymbol{\lambda_3} + \frac{\log c + \mathbf{a_3} - \mathbf{a_1}}{\log f}$$

This shows that the spectral decay parameter $\boldsymbol{\lambda_1}$ approximates $\boldsymbol{\lambda_3}$ with a correction term depending on the scaling factor $c$, the spectral amplitude, and frequency.

**Case 2:** Now consider the case where $\frac{c \cdot e^{\mathbf{a_3}}}{e^{\mathbf{a_2}}} \cdot f^{\boldsymbol{\lambda_2} - \boldsymbol{\lambda_3}} \ll 1$, such as when $c \to 0$. The logarithmic term is approximated using the first-order Taylor expansion:

$$\log \left( 1 + \frac{c \cdot e^{\mathbf{a_3}}}{e^{\mathbf{a_2}}} \cdot f^{\boldsymbol{\lambda_2} - \boldsymbol{\lambda_3}} \right) \approx \frac{c \cdot e^{\mathbf{a_3}}}{e^{\mathbf{a_2}}} \cdot f^{\boldsymbol{\lambda_2} - \boldsymbol{\lambda_3}}$$

Substituting into the main expression:

$$\mathbf{a_1} - \boldsymbol{\lambda_1} \log f \approx \mathbf{a_2} - \boldsymbol{\lambda_2} \log f + \frac{c \cdot e^{\mathbf{a_3}}}{e^{\mathbf{a_2}}} \cdot f^{\boldsymbol{\lambda_2} - \boldsymbol{\lambda_3}}$$

Solving for $\boldsymbol{\lambda_1}$:

$$\boldsymbol{\lambda_1} \approx \boldsymbol{\lambda_2} + \frac{\mathbf{a_2} - \mathbf{a_1}}{\log f} + c \cdot e^{\mathbf{a_3} - \mathbf{a_2}} \cdot \frac{f^{\boldsymbol{\lambda_2} - \boldsymbol{\lambda_3}}}{\log f}$$

As $c \to 0$, the correction vanishes, and $\boldsymbol{\lambda_1} \to \boldsymbol{\lambda_2}$, showing that the spectrum of the target converges to that of the source.

## D. Discussion on the VAR Inputs

From the expressions derived in Section C, it becomes evident that in the linear case one can identify certain quantities that are valuable inputs to a VAR model. As an example, given the two time series $\mathbf{x}_1$ and $\mathbf{x}_3$ of Section C, the relevant quantities are:

- $\mathbf{a}_1$: spectral intercept of the caused time series

- $\mathbf{a}_3$: spectral intercept of the causing time series

- $\boldsymbol{\lambda}_1$: spectral slope of the caused time series

- $\boldsymbol{\lambda}_3$: spectral slope of the causing time series

In this analysis, we neglect higher-order terms (e.g., exponentials in $\mathbf{a}$), as they empirically fail to improve system performance and are not present in the asymptotic regime discussed in Section C.

Each time series thus provides two spectral parameters: $\boldsymbol{\lambda}$ and $\mathbf{a}$. Among these, the $\boldsymbol{\lambda}$ parameters are identified as the main carriers of causal information. The simplest model we can consider involves assessing Granger causality between the $\boldsymbol{\lambda}$ sequences, i.e., testing for $\boldsymbol{\lambda}_3 \rightarrow \boldsymbol{\lambda}_1$ relationships.

To enhance the system's robustness in the presence of stationary processes, including the $\mathbf{a}$ parameter as a covariate in the VAR model has shown some performance improvements. However, in non-stationary settings, adding the $\mathbf{a}$ provides no additional causal information beyond $\boldsymbol{\lambda}$.

Finally, using the $\mathbf{a}$ parameter of the caused series either as an input or as a target in the VAR model has proven ineffective: while it does not meaningfully improve performance in the stationary case, it introduces false positives in non-stationary regimes. Therefore, its inclusion is not theoretically nor empirically justified.

### D.1. Study of $p$-values for Setting Window Length and Stride

In the proposed framework, both the stride and the sliding window length are configurable parameters that can be tuned by the user. The stride primarily serves to increase the number of available data points for analysis and plays a crucial role in the system's ability to detect short-range causal dependencies. If the true causal lag is shorter than the selected stride, the system may fail to capture such dynamics. Therefore, it is generally recommended to set the stride to 1. The sliding window length is a more delicate parameter, as it is influenced by multiple factors. A window that is too short may yield unreliable estimates due to limited data, whereas an excessively long window may blur or attenuate the causal signal, thus reducing detection sensitivity. To mitigate this trade-off, we introduce a data-driven procedure to estimate a suitable window length. This involves a preliminary evaluation of the fitting procedure across various window sizes, based on the distribution of $p$-values over the entire spectrum. The smallest window size that satisfies a $p$-value of $0.05$ is then selected for subsequent causal inference.

A reasonable requirement is also to fix a lower bound of this experimental criterion to a window length of a minimum of 50 datapoints. A smaller window may still satisfy the $p$-value requirement, but it will also make the fitted $\boldsymbol{\lambda}$ parameter too sensitive to endogenous variation of the autocorrelation caused by the phenomenon.

A study in function of stride and window length is shown in Figure 3.

### D.2. Slow-Varying Spectrum Systems

There are cases in which the system is sampled at such a high rate that it exhibits minimal local spectral variation. This typically occurs when the white noise component is small compared to other system dynamics. In these scenarios, spectral analysis becomes challenging, as we rely on observing how the spectrum evolves over time to infer causal relationships.

To address this issue, one should increase the window length and, if possible, reduce the stride of the sliding window. This allows the VAR model in the frequency domain to capture meaningful variations in the time series and better estimate potential causal links.

In these situations, time-domain methods like Granger causality may be more appropriate. While this introduces a limitation, the selection of a suitable causal inference technique can be effectively informed by an initial spectral analysis.

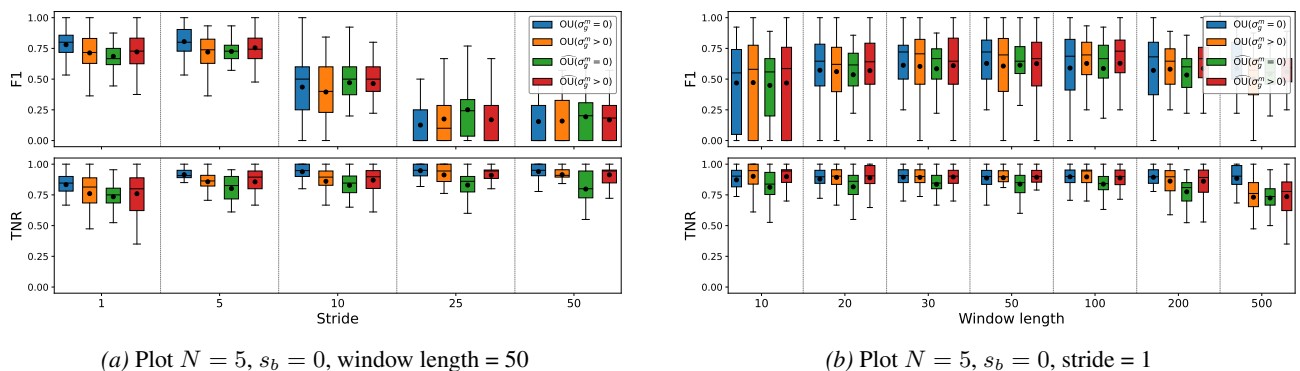

*(a)* Plot $N = 5$, $s_b = 0$, window length = 50      *(b)* Plot $N = 5$, $s_b = 0$, stride = 1

*Figure 3.* Stride and window length analysis.

# E. Further Experimental Details and Results

## E.1. OU Processes

Figure 4 shows two examples of the simulated OU processes, and the related causal graphs. In particular, an example of $\mathrm{OU}(\sigma_g^m = 0)$ is shown in Figure 4a and an example of $\mathrm{OU}(\sigma_g^m > 0)$ is shown in Figure 4b.

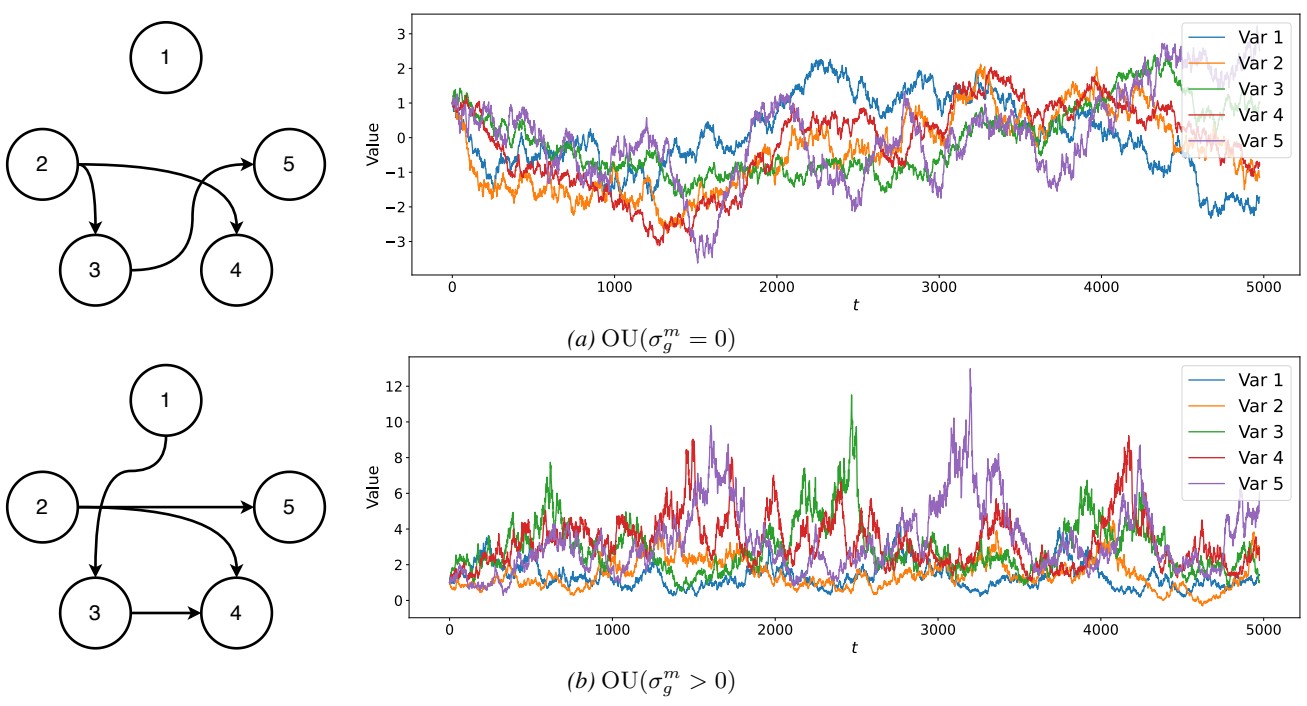

*(a)* $\mathrm{OU}(\sigma_g^m = 0)$

*(b)* $\mathrm{OU}(\sigma_g^m > 0)$

*Figure 4.* Generated synthetic processes.

## E.2. Additional Results

Tables 3 to 10 present all the experimental results for different configurations. We analyze the impact of varying the number of variables ($N \in \{5, 10\}$) and the scaling factor $\sigma_g^a \in \{0.5, 1\}$ on the methods' performance. In bold, we highlight the results of the best-performing algorithm in each scenario for a given estimator. Since some algorithms mainly detect causal relationships, their $TNR$ values tend to be high. In the $TNR$ tables, we also highlighted in green the best $TNR$ experiments that achieved an F1 score of at least 0.5. These experiments clearly demonstrate that PLaCy is the most reliable algorithm,

achieving the highest F1 score in most scenarios and a competitive or even superior TNR among all methods analyzed. Please note that we omit the results of RHINO and CCM in the tables: as explained in Section E.10, their prohibitive execution times made it impossible to run a sufficient number of experiments. In fact, Rhino lacks cross-configuration generalization along sequence length, noise level, and causal strength, necessitating $192^5$ distinct trainings to cover our experimental campaign. Combined with markedly greater runtime (see Section E.10) and lower F1 and TNR in the results of Section 5, this yields an unfavorable accuracy–efficiency trade-off. For these reasons, we omit it from the additional experiments reported in this section. A similar reasoning applies to CCM, for which, using the mean execution time reported in Section E.10, the total computation would amount to approximately one month in total.

Partial results on these two methods did not considerably differ from those reported in Section 5.

---

[5] Our tested configurations include 4 OU processes, 4 values for $\sigma_b$, 2 values for $\sigma_g$, 3 values for $C$, and 2 for $N$.

*Table 3.* F1 Score - $N = 5$, $\sigma_g^a = 0.5$

| Dataset | C | $\sigma_b$ | Granger | PLACy | PCMCI | RCV-VarLiNGAM | DYNOTEARS | PCMCI$\Omega$2 | BCGeweke | DTF | GewekeNP |
|---|---|---|---|---|---|---|---|---|---|---|---|
| $O_\cap(\sigma_m^\beta = 0)$ | 0.2 | 0.0 | **0.85**±0.19 | 0.12±0.19 | 0.06±0.15 | 0.44±0.17 | 0.02±0.11 | 0.06±0.12 | 0.84±0.08 | 0.44±0.09 | 0.61±0.05 |
| | | 0.1 | 0.44±0.17 | 0.15±0.22 | 0.09±0.19 | 0.44±0.17 | 0.12±0.21 | 0.08±0.12 | 0.57±0.17 | 0.27±0.21 | **0.60**±0.05 |
| | | 0.5 | 0.40±0.16 | 0.14±0.19 | 0.24±0.27 | 0.36±0.23 | 0.23±0.13 | 0.10±0.14 | 0.47±0.08 | 0.46±0.09 | **0.60**±0.07 |
| | | 1.0 | 0.49±0.18 | 0.10±0.18 | 0.11±0.20 | 0.34±0.24 | 0.21±0.13 | 0.14±0.15 | 0.55±0.10 | 0.44±0.08 | **0.60**±0.06 |
| | 0.5 | 0.0 | 0.82±0.18 | **0.84**±0.17 | 0.11±0.21 | 0.44±0.18 | 0.01±0.10 | 0.06±0.12 | 0.77±0.06 | 0.73±0.08 | 0.52±0.06 |
| | | 0.1 | 0.43±0.17 | **0.77**±0.23 | 0.13±0.21 | 0.41±0.18 | 0.02±0.10 | 0.07±0.11 | 0.51±0.09 | 0.42±0.23 | 0.52±0.07 |
| | | 0.5 | 0.38±0.14 | **0.60**±0.26 | 0.32±0.27 | 0.32±0.24 | 0.10±0.18 | 0.15±0.16 | 0.47±0.07 | 0.48±0.09 | 0.52±0.05 |
| | | 1.0 | 0.44±0.14 | 0.41±0.27 | 0.21±0.27 | 0.32±0.23 | 0.11±0.18 | 0.24±0.17 | 0.48±0.09 | 0.46±0.08 | **0.53**±0.06 |
| | 1.0 | 0.0 | 0.78±0.19 | **0.78**±0.16 | 0.20±0.28 | 0.40±0.17 | 0.01±0.10 | 0.07±0.13 | 0.76±0.10 | 0.64±0.10 | 0.53±0.08 |
| | | 0.1 | 0.44±0.18 | **0.77**±0.16 | 0.20±0.26 | 0.37±0.18 | 0.04±0.11 | 0.07±0.11 | 0.49±0.10 | 0.45±0.19 | 0.52±0.07 |
| | | 0.5 | 0.35±0.12 | **0.74**±0.28 | 0.42±0.28 | 0.22±0.21 | 0.17±0.22 | 0.24±0.11 | 0.45±0.09 | 0.44±0.10 | 0.53±0.07 |
| | | 1.0 | 0.41±0.13 | **0.65**±0.21 | 0.31±0.28 | 0.18±0.22 | 0.18±0.22 | 0.31±0.16 | 0.48±0.09 | 0.43±0.09 | 0.54±0.07 |
| $O_\cup(\sigma_m^\beta = 0)$ | 0.2 | 0.0 | **0.85**±0.19 | 0.20±0.24 | 0.06±0.18 | 0.44±0.17 | 0.00±0.00 | 0.06±0.12 | 0.83±0.11 | 0.44±0.07 | 0.44±0.07 |
| | | 0.1 | 0.33±0.13 | 0.29±0.23 | 0.16±0.24 | 0.42±0.17 | 0.00±0.00 | 0.05±0.11 | **0.46**±0.07 | 0.43±0.07 | 0.44±0.07 |
| | | 0.5 | 0.27±0.11 | 0.17±0.22 | 0.29±0.30 | 0.38±0.18 | 0.03±0.12 | 0.12±0.14 | 0.47±0.08 | 0.44±0.07 | **0.49**±0.05 |
| | | 1.0 | 0.36±0.14 | 0.20±0.23 | 0.15±0.23 | 0.33±0.19 | 0.06±0.15 | 0.14±0.15 | 0.52±0.09 | 0.44±0.07 | **0.57**±0.07 |
| | 0.5 | 0.0 | **0.81**±0.19 | 0.74±0.17 | 0.09±0.18 | 0.44±0.17 | 0.00±0.00 | 0.08±0.13 | 0.78±0.10 | 0.44±0.07 | 0.44±0.07 |
| | | 0.1 | 0.32±0.12 | **0.61**±0.20 | 0.17±0.23 | 0.42±0.18 | 0.00±0.00 | 0.09±0.13 | 0.47±0.08 | 0.44±0.07 | 0.44±0.07 |
| | | 0.5 | 0.27±0.11 | **0.57**±0.11 | 0.36±0.28 | 0.34±0.22 | 0.01±0.05 | 0.16±0.15 | 0.47±0.07 | 0.44±0.07 | 0.47±0.10 |
| | | 1.0 | 0.35±0.12 | 0.46±0.28 | 0.15±0.24 | 0.30±0.21 | 0.03±0.10 | 0.27±0.18 | 0.50±0.07 | 0.44±0.07 | **0.51**±0.06 |
| | 1.0 | 0.0 | **0.78**±0.19 | 0.71±0.17 | 0.18±0.25 | 0.38±0.17 | 0.00±0.00 | 0.09±0.13 | 0.75±0.09 | 0.43±0.09 | 0.43±0.09 |
| | | 0.1 | 0.32±0.12 | **0.58**±0.16 | 0.19±0.26 | 0.35±0.17 | 0.00±0.00 | 0.07±0.12 | 0.47±0.09 | 0.43±0.09 | 0.43±0.09 |
| | | 0.5 | 0.27±0.11 | **0.65**±0.11 | 0.37±0.28 | 0.34±0.22 | 0.01±0.04 | 0.22±0.17 | 0.44±0.09 | 0.43±0.09 | 0.46±0.10 |
| | | 1.0 | 0.33±0.12 | **0.60**±0.18 | 0.28±0.29 | 0.24±0.23 | 0.03±0.13 | 0.34±0.17 | 0.49±0.07 | 0.43±0.09 | 0.51±0.08 |
| $O_\cap(\sigma_m^\beta > 0)$ | 0.2 | 0.0 | 0.74±0.20 | 0.45±0.31 | 0.14±0.24 | 0.45±0.17 | 0.19±0.16 | 0.14±0.15 | **0.88**±0.07 | 0.49±0.10 | 0.58±0.05 |
| | | 0.1 | 0.68±0.20 | 0.57±0.29 | 0.11±0.22 | 0.41±0.17 | 0.20±0.19 | 0.11±0.14 | **0.78**±0.08 | 0.33±0.07 | 0.54±0.06 |
| | | 0.5 | 0.69±0.22 | 0.60±0.29 | 0.11±0.21 | 0.40±0.17 | 0.22±0.13 | 0.14±0.16 | **0.76**±0.10 | 0.23±0.21 | 0.54±0.07 |
| | | 1.0 | 0.69±0.22 | 0.70±0.20 | 0.08±0.19 | 0.43±0.12 | 0.24±0.12 | 0.13±0.14 | **0.76**±0.10 | 0.42±0.16 | 0.50±0.06 |
| | 0.5 | 0.0 | 0.70±0.19 | 0.81±0.14 | 0.24±0.31 | 0.42±0.17 | 0.06±0.16 | 0.16±0.16 | **0.82**±0.06 | 0.66±0.15 | 0.52±0.07 |
| | | 0.1 | 0.66±0.18 | 0.81±0.19 | 0.26±0.29 | 0.40±0.17 | 0.04±0.13 | 0.15±0.16 | **0.83**±0.07 | 0.65±0.09 | 0.51±0.06 |
| | | 0.5 | 0.64±0.19 | 0.77±0.20 | 0.20±0.26 | 0.37±0.19 | 0.13±0.16 | 0.17±0.17 | **0.80**±0.12 | 0.53±0.24 | 0.56±0.08 |
| | | 1.0 | 0.65±0.19 | 0.80±0.19 | 0.22±0.29 | 0.39±0.19 | 0.14±0.15 | 0.18±0.16 | **0.86**±0.08 | 0.61±0.15 | 0.52±0.08 |
| | 1.0 | 0.0 | 0.66±0.19 | **0.78**±0.15 | 0.43±0.34 | 0.39±0.18 | 0.07±0.15 | 0.20±0.17 | 0.75±0.12 | 0.63±0.09 | 0.53±0.07 |
| | | 0.1 | 0.63±0.19 | **0.77**±0.14 | 0.44±0.34 | 0.35±0.17 | 0.03±0.11 | 0.21±0.19 | 0.74±0.12 | 0.62±0.10 | 0.52±0.06 |
| | | 0.5 | 0.64±0.18 | **0.78**±0.17 | 0.38±0.34 | 0.35±0.20 | 0.13±0.18 | 0.21±0.18 | 0.74±0.13 | 0.61±0.11 | 0.52±0.09 |
| | | 1.0 | 0.62±0.20 | **0.74**±0.17 | 0.33±0.34 | 0.33±0.21 | 0.15±0.16 | 0.22±0.16 | 0.69±0.12 | 0.53±0.15 | 0.54±0.07 |
| $O_\cup(\sigma_m^\beta > 0)$ | 0.2 | 0.0 | 0.27±0.11 | **0.55**±0.31 | 0.35±0.21 | 0.09±0.18 | 0.04±0.14 | 0.26±0.11 | 0.48±0.08 | 0.44±0.07 | 0.44±0.07 |
| | | 0.1 | 0.27±0.11 | **0.64**±0.28 | 0.36±0.24 | 0.11±0.21 | 0.04±0.14 | 0.26±0.11 | 0.47±0.07 | 0.46±0.07 | 0.44±0.07 |
| | | 0.5 | 0.38±0.15 | **0.67**±0.25 | 0.23±0.26 | 0.35±0.22 | 0.04±0.12 | 0.22±0.12 | 0.64±0.18 | 0.42±0.06 | 0.49±0.07 |
| | | 1.0 | 0.56±0.20 | 0.66±0.27 | 0.14±0.25 | 0.42±0.19 | 0.05±0.16 | 0.14±0.15 | **0.69**±0.11 | 0.34±0.23 | 0.49±0.11 |
| | 0.5 | 0.0 | 0.27±0.11 | **0.84**±0.14 | 0.45±0.22 | 0.06±0.14 | 0.01±0.08 | 0.26±0.11 | 0.48±0.07 | 0.44±0.06 | 0.44±0.07 |
| | | 0.1 | 0.27±0.11 | **0.79**±0.18 | 0.44±0.23 | 0.08±0.17 | 0.01±0.07 | 0.26±0.11 | 0.48±0.07 | 0.46±0.06 | 0.44±0.07 |
| | | 0.5 | 0.39±0.15 | **0.79**±0.17 | 0.37±0.29 | 0.32±0.21 | 0.01±0.06 | 0.24±0.14 | 0.57±0.11 | 0.53±0.11 | 0.46±0.08 |
| | | 1.0 | 0.57±0.19 | **0.82**±0.18 | 0.26±0.30 | 0.36±0.20 | 0.01±0.07 | 0.15±0.15 | 0.71±0.13 | 0.45±0.13 | 0.48±0.08 |
| | 1.0 | 0.0 | 0.27±0.11 | **0.80**±0.15 | 0.53±0.21 | 0.07±0.16 | 0.02±0.09 | 0.25±0.11 | 0.46±0.10 | 0.44±0.08 | 0.43±0.09 |
| | | 0.1 | 0.28±0.11 | **0.81**±0.17 | 0.53±0.21 | 0.09±0.20 | 0.02±0.12 | 0.25±0.12 | 0.46±0.10 | 0.44±0.08 | 0.43±0.09 |
| | | 0.5 | 0.38±0.16 | **0.76**±0.18 | 0.48±0.30 | 0.24±0.21 | 0.02±0.11 | 0.25±0.14 | 0.54±0.12 | 0.48±0.08 | 0.46±0.08 |
| | | 1.0 | 0.53±0.18 | **0.78**±0.17 | 0.39±0.32 | 0.28±0.21 | 0.03±0.14 | 0.21±0.15 | 0.68±0.13 | 0.53±0.10 | 0.48±0.08 |

*Table 4.* TNR Score - $N = 5$, $\sigma_g^a = 0.5$

| Dataset | C | $\sigma_b$ | Granger | PLACy | PCMCI | RCV-VarLiNGAM | DYNOTEARS | PCMCI$_\Omega$ | BCGeweke | DTF | GewekeNP |
|---|---|---|---|---|---|---|---|---|---|---|---|
| $O\Pi(\sigma_m^\beta = 0)$ | 0.2 | 0.0 | **0.96**±0.04 | 0.96±0.04 | **0.99**±0.02 | 0.72±0.06 | 0.98±0.04 | 0.78±0.09 | 0.89±0.05 | 0.93±0.05 | 0.63±0.11 |
| | | 0.1 | 0.65±0.14 | 0.96±0.04 | **0.98**±0.03 | 0.73±0.07 | 0.88±0.09 | 0.78±0.09 | 0.55±0.20 | 0.79±0.08 | 0.60±0.14 |
| | | 0.5 | 0.59±0.12 | 0.95±0.05 | **0.96**±0.04 | 0.84±0.07 | 0.53±0.16 | 0.78±0.08 | 0.35±0.09 | 0.45±0.09 | 0.60±0.11 |
| | | 1.0 | 0.72±0.11 | 0.94±0.05 | **0.99**±0.02 | 0.87±0.06 | 0.46±0.14 | 0.77±0.09 | 0.52±0.09 | 0.31±0.05 | 0.62±0.08 |
| | 0.5 | 0.0 | 0.94±0.05 | 0.95±0.04 | **0.99**±0.02 | 0.74±0.07 | 0.95±0.05 | 0.77±0.11 | 0.82±0.05 | 0.85±0.09 | 0.46±0.13 |
| | | 0.1 | 0.63±0.14 | 0.95±0.05 | **0.98**±0.03 | 0.78±0.07 | 0.98±0.05 | 0.75±0.11 | 0.46±0.09 | 0.77±0.12 | 0.48±0.10 |
| | | 0.5 | 0.55±0.13 | 0.95±0.05 | 0.95±0.05 | 0.85±0.07 | 0.94±0.10 | 0.74±0.11 | 0.36±0.09 | 0.42±0.07 | 0.47±0.10 |
| | | 1.0 | 0.66±0.12 | 0.93±0.06 | **0.98**±0.03 | 0.89±0.07 | 0.93±0.09 | 0.75±0.11 | 0.38±0.09 | 0.34±0.08 | 0.47±0.12 |
| | 1.0 | 0.0 | **0.92**±0.08 | 0.91±0.07 | **0.99**±0.02 | 0.75±0.08 | 0.90±0.08 | 0.76±0.12 | 0.81±0.09 | 0.68±0.13 | 0.49±0.14 |
| | | 0.1 | 0.63±0.17 | 0.91±0.07 | **0.98**±0.04 | 0.79±0.07 | 0.94±0.05 | 0.76±0.11 | 0.41±0.12 | 0.61±0.20 | 0.48±0.14 |
| | | 0.5 | 0.51±0.14 | **0.91**±0.07 | 0.94±0.05 | 0.87±0.07 | 0.92±0.11 | 0.73±0.13 | 0.32±0.07 | 0.36±0.11 | 0.48±0.15 |
| | | 1.0 | 0.61±0.15 | **0.90**±0.08 | 0.97±0.04 | 0.92±0.06 | 0.91±0.10 | 0.73±0.13 | 0.40±0.10 | 0.29±0.06 | 0.51±0.14 |
| $\underline{O}\Pi(\sigma_m^\beta = 0)$ | 0.2 | 0.0 | **0.96**±0.04 | 0.93±0.05 | 0.99±0.02 | 0.72±0.06 | **1.00**±0.00 | 0.78±0.09 | 0.89±0.07 | 0.26±0.02 | 0.26±0.02 |
| | | 0.1 | 0.45±0.10 | 0.86±0.08 | 0.97±0.03 | 0.73±0.06 | **1.00**±0.00 | 0.78±0.09 | 0.33±0.05 | 0.26±0.02 | 0.26±0.02 |
| | | 0.5 | 0.26±0.04 | 0.92±0.06 | 0.95±0.05 | 0.82±0.06 | **0.99**±0.03 | 0.77±0.08 | 0.36±0.05 | 0.26±0.02 | 0.38±0.14 |
| | | 1.0 | 0.52±0.12 | 0.91±0.06 | **0.98**±0.03 | 0.85±0.05 | 0.96±0.05 | 0.78±0.09 | 0.48±0.10 | 0.26±0.02 | 0.55±0.13 |
| | 0.5 | 0.0 | **0.94**±0.06 | 0.92±0.05 | 1.00±0.01 | 0.75±0.08 | **1.00**±0.00 | 0.76±0.11 | 0.85±0.06 | 0.26±0.02 | 0.26±0.02 |
| | | 0.1 | 0.44±0.10 | 0.85±0.08 | 0.95±0.05 | 0.78±0.07 | **1.00**±0.00 | 0.75±0.11 | 0.36±0.05 | 0.26±0.02 | 0.26±0.02 |
| | | 0.5 | 0.26±0.05 | **0.91**±0.06 | 0.94±0.05 | 0.84±0.07 | **0.99**±0.03 | 0.76±0.10 | 0.35±0.05 | 0.26±0.02 | 0.34±0.11 |
| | | 1.0 | 0.50±0.12 | 0.90±0.07 | **0.97**±0.03 | 0.87±0.07 | 0.95±0.05 | 0.76±0.11 | 0.41±0.13 | 0.26±0.02 | **0.45**±0.10 |
| | 1.0 | 0.0 | **0.92**±0.07 | 0.88±0.09 | 1.00±0.02 | 0.77±0.09 | **1.00**±0.00 | 0.76±0.11 | 0.81±0.09 | 0.26±0.02 | 0.26±0.02 |
| | | 0.1 | 0.43±0.12 | 0.80±0.10 | 0.96±0.04 | 0.78±0.08 | **1.00**±0.00 | 0.75±0.12 | 0.37±0.09 | 0.26±0.02 | 0.26±0.02 |
| | | 0.5 | 0.26±0.05 | **0.88**±0.08 | 0.93±0.06 | 0.87±0.08 | **0.98**±0.04 | 0.73±0.11 | 0.30±0.05 | 0.26±0.02 | 0.33±0.11 |
| | | 1.0 | 0.47±0.12 | **0.85**±0.10 | 0.97±0.03 | 0.91±0.07 | 0.93±0.06 | 0.73±0.13 | 0.43±0.10 | 0.26±0.02 | 0.45±0.13 |
| $O\Pi(\sigma_m^\beta > 0)$ | 0.2 | 0.0 | **0.91**±0.06 | 0.96±0.04 | **0.99**±0.02 | 0.73±0.06 | 0.72±0.09 | 0.77±0.09 | 0.91±0.06 | 0.91±0.06 | **0.63**±0.12 |
| | | 0.1 | 0.89±0.07 | 0.95±0.05 | **0.99**±0.03 | 0.73±0.07 | 0.77±0.13 | 0.76±0.10 | 0.86±0.08 | 0.86±0.08 | 0.57±0.08 |
| | | 0.5 | 0.88±0.07 | **0.96**±0.06 | **0.98**±0.03 | 0.74±0.07 | 0.49±0.19 | 0.76±0.11 | 0.81±0.08 | 0.87±0.09 | 0.63±0.10 |
| | | 1.0 | 0.88±0.08 | **0.96**±0.04 | **0.98**±0.03 | 0.74±0.07 | 0.40±0.15 | 0.75±0.10 | 0.84±0.07 | 0.90±0.07 | 0.57±0.11 |
| | 0.5 | 0.0 | 0.88±0.09 | 0.94±0.05 | **0.99**±0.02 | 0.75±0.07 | 0.95±0.05 | 0.75±0.10 | 0.86±0.07 | 0.75±0.18 | 0.46±0.12 |
| | | 0.1 | 0.86±0.08 | 0.95±0.06 | **0.98**±0.03 | 0.76±0.08 | 0.96±0.05 | 0.73±0.12 | 0.87±0.09 | 0.82±0.12 | 0.47±0.12 |
| | | 0.5 | 0.84±0.11 | **0.94**±0.06 | **0.98**±0.03 | 0.76±0.09 | 0.81±0.15 | 0.74±0.12 | 0.84±0.15 | 0.80±0.08 | 0.51±0.18 |
| | | 1.0 | 0.84±0.10 | **0.94**±0.06 | **0.98**±0.03 | 0.78±0.08 | 0.75±0.16 | 0.72±0.12 | 0.90±0.06 | 0.81±0.14 | 0.50±0.18 |
| | 1.0 | 0.0 | 0.85±0.13 | 0.91±0.09 | **0.98**±0.03 | 0.76±0.07 | 0.91±0.08 | 0.75±0.11 | 0.80±0.14 | 0.69±0.15 | 0.49±0.17 |
| | | 0.1 | 0.82±0.13 | 0.91±0.08 | **0.98**±0.03 | 0.77±0.09 | 0.93±0.07 | 0.73±0.12 | 0.78±0.16 | 0.68±0.15 | 0.48±0.15 |
| | | 0.5 | 0.83±0.13 | **0.91**±0.08 | **0.98**±0.03 | 0.78±0.08 | 0.81±0.14 | 0.72±0.14 | 0.77±0.14 | 0.71±0.16 | 0.46±0.17 |
| | | 1.0 | 0.81±0.15 | **0.90**±0.08 | **0.98**±0.03 | 0.79±0.08 | 0.75±0.14 | 0.73±0.12 | 0.75±0.11 | 0.66±0.14 | 0.51±0.16 |
| $\underline{O}\Pi(\sigma_m^\beta > 0)$ | 0.2 | 0.0 | 0.28±0.08 | 0.96±0.04 | 0.84±0.07 | 0.97±0.06 | **0.97**±0.05 | 0.28±0.08 | 0.37±0.08 | 0.36±0.07 | 0.26±0.02 |
| | | 0.1 | 0.30±0.10 | 0.96±0.05 | 0.86±0.07 | 0.96±0.06 | **0.97**±0.05 | 0.30±0.11 | 0.36±0.07 | 0.40±0.08 | 0.26±0.02 |
| | | 0.5 | 0.56±0.15 | 0.96±0.04 | 0.94±0.04 | 0.81±0.09 | 0.96±0.05 | 0.53±0.13 | 0.62±0.19 | 0.63±0.17 | 0.34±0.08 |
| | | 1.0 | 0.79±0.11 | 0.96±0.04 | **0.98**±0.03 | 0.76±0.08 | 0.95±0.05 | 0.70±0.12 | 0.78±0.10 | 0.81±0.09 | 0.54±0.16 |
| | 0.5 | 0.0 | 0.28±0.08 | 0.95±0.06 | 0.84±0.07 | **0.97**±0.05 | 0.96±0.05 | 0.28±0.08 | 0.37±0.07 | 0.32±0.06 | 0.26±0.02 |
| | | 0.1 | 0.28±0.09 | 0.94±0.05 | 0.86±0.07 | 0.96±0.06 | **0.96**±0.05 | 0.29±0.09 | 0.37±0.10 | 0.35±0.06 | 0.27±0.03 |
| | | 0.5 | 0.56±0.18 | 0.94±0.06 | 0.95±0.05 | 0.83±0.09 | **0.96**±0.05 | 0.53±0.15 | 0.55±0.15 | 0.51±0.18 | 0.34±0.09 |
| | | 1.0 | 0.80±0.11 | 0.95±0.05 | **0.98**±0.03 | 0.79±0.08 | 0.93±0.06 | 0.70±0.13 | 0.76±0.14 | 0.68±0.17 | 0.41±0.10 |
| | 1.0 | 0.0 | 0.27±0.08 | 0.92±0.08 | 0.84±0.08 | **0.97**±0.05 | 0.96±0.06 | 0.28±0.08 | 0.34±0.08 | 0.29±0.05 | 0.26±0.02 |
| | | 0.1 | 0.29±0.11 | 0.93±0.07 | 0.86±0.07 | **0.96**±0.06 | 0.96±0.06 | 0.29±0.08 | 0.36±0.10 | 0.29±0.06 | 0.26±0.02 |
| | | 0.5 | 0.52±0.18 | 0.92±0.07 | 0.95±0.05 | 0.83±0.08 | 0.95±0.06 | 0.51±0.15 | 0.52±0.15 | 0.42±0.10 | 0.34±0.11 |
| | | 1.0 | 0.74±0.15 | 0.91±0.09 | **0.97**±0.03 | 0.80±0.08 | 0.92±0.07 | 0.69±0.12 | 0.73±0.12 | 0.54±0.15 | 0.40±0.13 |

Table 5. F1 Score - $N = 5$, $\sigma_g^a = 1.0$

| Dataset | C | $\sigma_b$ | Granger | PLACy | PCMCI | RCV-VarLiNGAM | DYNOTEARS | PCMCI$_\Omega$ | BCGeweke | DTF | GewekeNP |
|---|---|---|---|---|---|---|---|---|---|---|---|
| $O \cap (\sigma_m^g = 0)$ | 0.2 | 0.0 | **0.85**±0.20 | 0.13±0.21 | 0.08±0.19 | 0.44±0.17 | 0.05±0.16 | 0.06±0.12 | 0.82±0.10 | 0.45±0.13 | 0.60±0.06 |
| | | 0.1 | 0.63±0.21 | 0.14±0.21 | 0.06±0.14 | 0.43±0.17 | 0.15±0.21 | 0.04±0.11 | **0.68**±0.16 | 0.18±0.20 | 0.58±0.06 |
| | | 0.5 | 0.43±0.17 | 0.13±0.21 | 0.22±0.28 | 0.36±0.19 | 0.23±0.15 | 0.10±0.13 | 0.47±0.07 | 0.40±0.13 | **0.60**±0.05 |
| | | 1.0 | 0.39±0.14 | 0.15±0.24 | 0.24±0.26 | 0.36±0.21 | 0.23±0.13 | 0.10±0.14 | 0.48±0.08 | 0.37±0.12 | **0.59**±0.06 |
| | 0.5 | 0.0 | 0.82±0.19 | **0.83**±0.16 | 0.08±0.19 | 0.44±0.18 | 0.02±0.12 | 0.08±0.12 | 0.76±0.08 | 0.75±0.09 | 0.52±0.06 |
| | | 0.1 | 0.62±0.22 | **0.81**±0.19 | 0.07±0.17 | 0.43±0.16 | 0.02±0.09 | 0.08±0.12 | 0.64±0.08 | 0.42±0.13 | 0.51±0.07 |
| | | 0.5 | 0.43±0.16 | **0.67**±0.28 | 0.34±0.27 | 0.37±0.19 | 0.12±0.20 | 0.11±0.15 | 0.47±0.08 | 0.44±0.16 | 0.53±0.07 |
| | | 1.0 | 0.37±0.13 | **0.60**±0.25 | 0.32±0.27 | 0.33±0.19 | 0.16±0.22 | 0.18±0.16 | 0.46±0.07 | 0.47±0.09 | 0.51±0.08 |
| | 1.0 | 0.0 | 0.78±0.20 | **0.78**±0.15 | 0.19±0.27 | 0.40±0.17 | 0.02±0.09 | 0.08±0.13 | 0.73±0.09 | 0.67±0.14 | 0.52±0.08 |
| | | 0.1 | 0.61±0.21 | **0.77**±0.18 | 0.15±0.25 | 0.39±0.17 | 0.04±0.13 | 0.09±0.13 | 0.61±0.10 | 0.56±0.08 | 0.51±0.07 |
| | | 0.5 | 0.39±0.16 | **0.78**±0.16 | 0.38±0.29 | 0.27±0.22 | 0.12±0.18 | 0.15±0.16 | 0.44±0.09 | 0.46±0.09 | 0.50±0.07 |
| | | 1.0 | 0.35±0.12 | **0.73**±0.19 | 0.37±0.27 | 0.22±0.20 | 0.14±0.22 | 0.20±0.16 | 0.44±0.09 | 0.46±0.08 | 0.52±0.08 |
| $\underline{O} \cap (\sigma_m^g = 0)$ | 0.2 | 0.0 | **0.85**±0.19 | 0.21±0.24 | 0.06±0.18 | 0.44±0.17 | 0.00±0.00 | 0.06±0.12 | **0.83**±0.11 | 0.44±0.07 | 0.44±0.07 |
| | | 0.1 | 0.47±0.19 | 0.34±0.24 | 0.04±0.13 | 0.44±0.17 | 0.00±0.00 | 0.06±0.12 | **0.57**±0.11 | 0.43±0.07 | 0.44±0.07 |
| | | 0.5 | 0.27±0.11 | 0.22±0.24 | 0.31±0.27 | 0.38±0.19 | 0.02±0.12 | 0.09±0.13 | 0.45±0.08 | 0.44±0.08 | **0.50**±0.09 |
| | | 1.0 | 0.28±0.11 | 0.19±0.24 | 0.24±0.26 | 0.38±0.20 | 0.05±0.16 | 0.09±0.13 | 0.46±0.09 | 0.44±0.07 | **0.58**±0.08 |
| | 0.5 | 0.0 | **0.81**±0.19 | 0.79±0.18 | 0.10±0.19 | 0.43±0.17 | 0.00±0.00 | 0.08±0.12 | 0.78±0.10 | 0.44±0.07 | 0.44±0.07 |
| | | 0.1 | 0.45±0.18 | **0.71**±0.18 | 0.06±0.15 | 0.43±0.17 | 0.00±0.00 | 0.07±0.12 | 0.52±0.06 | 0.44±0.08 | 0.44±0.07 |
| | | 0.5 | 0.26±0.11 | **0.70**±0.22 | 0.39±0.25 | 0.32±0.20 | 0.00±0.04 | 0.09±0.15 | 0.45±0.06 | 0.44±0.07 | 0.46±0.07 |
| | | 1.0 | 0.28±0.11 | **0.56**±0.25 | 0.31±0.26 | 0.30±0.22 | 0.02±0.10 | 0.16±0.16 | 0.46±0.08 | 0.44±0.07 | 0.50±0.05 |
| | 1.0 | 0.0 | **0.78**±0.19 | 0.74±0.18 | 0.18±0.25 | 0.37±0.17 | 0.00±0.00 | 0.08±0.13 | 0.75±0.09 | 0.43±0.09 | 0.43±0.09 |
| | | 0.1 | 0.46±0.19 | **0.68**±0.17 | 0.12±0.21 | 0.37±0.17 | 0.00±0.00 | 0.08±0.13 | 0.53±0.10 | 0.43±0.09 | 0.43±0.09 |
| | | 0.5 | 0.27±0.11 | **0.71**±0.16 | 0.45±0.27 | 0.29±0.20 | 0.01±0.05 | 0.11±0.13 | 0.43±0.09 | 0.42±0.09 | 0.46±0.08 |
| | | 1.0 | 0.28±0.11 | **0.66**±0.17 | 0.42±0.30 | 0.25±0.20 | 0.02±0.10 | 0.23±0.17 | 0.45±0.08 | 0.43±0.09 | 0.51±0.07 |
| $O \cap (\sigma_m^g > 0)$ | 0.2 | 0.0 | 0.65±0.22 | 0.68±0.25 | 0.25±0.31 | 0.41±0.19 | 0.20±0.16 | 0.16±0.16 | **0.84**±0.09 | 0.56±0.11 | 0.51±0.06 |
| | | 0.1 | 0.64±0.22 | 0.70±0.25 | 0.25±0.30 | 0.39±0.18 | 0.18±0.16 | 0.15±0.16 | **0.83**±0.07 | 0.48±0.11 | 0.51±0.07 |
| | | 0.5 | 0.59±0.21 | 0.71±0.24 | 0.23±0.28 | 0.37±0.18 | 0.24±0.12 | 0.17±0.17 | **0.76**±0.12 | 0.54±0.17 | 0.50±0.07 |
| | | 1.0 | 0.61±0.19 | **0.79**±0.22 | 0.21±0.21 | 0.38±0.18 | 0.26±0.11 | 0.15±0.16 | 0.74±0.12 | 0.42±0.20 | 0.50±0.11 |
| | 0.5 | 0.0 | 0.61±0.19 | **0.82**±0.17 | 0.41±0.34 | 0.35±0.19 | 0.09±0.17 | 0.20±0.16 | 0.76±0.08 | 0.68±0.10 | 0.49±0.07 |
| | | 0.1 | 0.64±0.19 | **0.76**±0.18 | 0.42±0.32 | 0.34±0.19 | 0.09±0.17 | 0.23±0.17 | 0.75±0.12 | 0.69±0.10 | 0.53±0.04 |
| | | 0.5 | 0.61±0.20 | **0.80**±0.16 | 0.43±0.32 | 0.32±0.21 | 0.20±0.16 | 0.22±0.16 | 0.71±0.10 | 0.67±0.13 | 0.48±0.09 |
| | | 1.0 | 0.61±0.21 | **0.80**±0.16 | 0.37±0.32 | 0.33±0.22 | 0.21±0.14 | 0.22±0.16 | 0.73±0.11 | 0.50±0.14 | 0.50±0.05 |
| | 1.0 | 0.0 | 0.57±0.18 | **0.78**±0.18 | 0.62±0.28 | 0.32±0.20 | 0.11±0.17 | 0.22±0.16 | 0.70±0.13 | 0.60±0.12 | 0.51±0.08 |
| | | 0.1 | 0.57±0.18 | **0.76**±0.18 | 0.61±0.29 | 0.28±0.21 | 0.13±0.19 | 0.21±0.17 | 0.67±0.14 | 0.57±0.12 | 0.52±0.07 |
| | | 0.5 | 0.56±0.19 | **0.76**±0.17 | 0.55±0.32 | 0.25±0.21 | 0.23±0.17 | 0.24±0.18 | 0.71±0.12 | 0.61±0.13 | 0.50±0.08 |
| | | 1.0 | 0.58±0.19 | **0.77**±0.19 | 0.60±0.29 | 0.28±0.21 | 0.23±0.16 | 0.25±0.19 | 0.67±0.16 | 0.59±0.13 | 0.50±0.10 |
| $\underline{O} \cap (\sigma_m^g > 0)$ | 0.2 | 0.0 | 0.29±0.12 | **0.74**±0.21 | 0.37±0.24 | 0.12±0.21 | 0.02±0.08 | 0.25±0.12 | 0.49±0.08 | 0.43±0.05 | 0.44±0.08 |
| | | 0.1 | 0.30±0.12 | **0.74**±0.21 | 0.36±0.24 | 0.12±0.20 | 0.03±0.13 | 0.26±0.12 | 0.50±0.09 | 0.46±0.04 | 0.44±0.07 |
| | | 0.5 | 0.43±0.19 | **0.77**±0.20 | 0.31±0.26 | 0.28±0.22 | 0.02±0.07 | 0.19±0.14 | 0.64±0.13 | 0.44±0.17 | 0.49±0.09 |
| | | 1.0 | 0.54±0.22 | **0.77**±0.23 | 0.25±0.30 | 0.35±0.18 | 0.02±0.10 | 0.17±0.15 | **0.79**±0.14 | 0.52±0.15 | 0.48±0.07 |
| | 0.5 | 0.0 | 0.29±0.12 | **0.82**±0.16 | 0.46±0.20 | 0.07±0.16 | 0.01±0.06 | 0.26±0.11 | 0.49±0.07 | 0.48±0.08 | 0.44±0.07 |
| | | 0.1 | 0.31±0.12 | **0.80**±0.17 | 0.46±0.21 | 0.11±0.20 | 0.01±0.07 | 0.26±0.12 | 0.50±0.08 | 0.47±0.09 | 0.45±0.07 |
| | | 0.5 | 0.42±0.16 | **0.80**±0.18 | 0.47±0.28 | 0.21±0.20 | 0.00±0.03 | 0.22±0.14 | 0.61±0.08 | 0.57±0.10 | 0.48±0.08 |
| | | 1.0 | 0.53±0.19 | **0.80**±0.18 | 0.44±0.33 | 0.29±0.22 | 0.01±0.05 | 0.22±0.15 | 0.69±0.15 | 0.56±0.13 | 0.49±0.10 |
| | 1.0 | 0.0 | 0.29±0.11 | **0.78**±0.17 | 0.51±0.21 | 0.07±0.15 | 0.01±0.05 | 0.27±0.12 | 0.48±0.09 | 0.46±0.08 | 0.44±0.08 |
| | | 0.1 | 0.30±0.13 | **0.77**±0.17 | 0.52±0.22 | 0.09±0.17 | 0.02±0.11 | 0.26±0.13 | 0.48±0.10 | 0.47±0.08 | 0.44±0.08 |
| | | 0.5 | 0.43±0.17 | **0.77**±0.17 | 0.58±0.26 | 0.22±0.22 | 0.00±0.03 | 0.25±0.15 | 0.55±0.10 | 0.56±0.11 | 0.46±0.09 |
| | | 1.0 | 0.52±0.18 | **0.78**±0.17 | 0.60±0.29 | 0.22±0.20 | 0.01±0.05 | 0.25±0.18 | 0.66±0.12 | 0.57±0.10 | 0.50±0.08 |

*Table 6.* TNR Score - $N = 5$, $\sigma_g^a = 1.0$

| Dataset | C | $\sigma_b$ | Granger | PLACy | PCMCI | RCV-VarLiNGAM | DYNOTEARS | PCMCI$_\Omega$ | BCGeweke | DTF | GewekeNP |
|---|---|---|---|---|---|---|---|---|---|---|---|
| $O \cap (\sigma_m^\beta = 0)$ | 0.2 | 0.0 | **0.96**±0.04 | 0.96±0.04 | **0.99**±0.02 | 0.72±0.06 | 0.94±0.08 | 0.78±0.09 | 0.86±0.08 | 0.93±0.05 | 0.60±0.11 |
| | | 0.1 | **0.84**±0.09 | 0.96±0.04 | **0.99**±0.03 | 0.72±0.06 | 0.82±0.16 | 0.79±0.08 | 0.71±0.17 | 0.89±0.05 | 0.58±0.09 |
| | | 0.5 | 0.63±0.13 | **0.96**±0.04 | 0.95±0.05 | 0.77±0.08 | 0.52±0.18 | 0.77±0.09 | 0.35±0.07 | 0.61±0.11 | **0.60**±0.13 |
| | | 1.0 | 0.58±0.14 | **0.96**±0.04 | 0.96±0.04 | 0.82±0.07 | 0.49±0.15 | 0.78±0.09 | 0.38±0.07 | 0.35±0.09 | **0.61**±0.08 |
| | 0.5 | 0.0 | 0.94±0.05 | **0.95**±0.04 | **0.99**±0.02 | 0.74±0.07 | 0.96±0.05 | 0.76±0.11 | 0.82±0.06 | 0.87±0.09 | 0.46±0.13 |
| | | 0.1 | 0.83±0.12 | **0.96**±0.04 | **0.99**±0.02 | 0.76±0.07 | 0.97±0.06 | 0.75±0.12 | 0.68±0.08 | 0.80±0.08 | 0.45±0.12 |
| | | 0.5 | 0.63±0.15 | **0.95**±0.04 | 0.95±0.04 | 0.83±0.06 | 0.89±0.13 | 0.76±0.10 | 0.35±0.11 | 0.61±0.12 | 0.49±0.11 |
| | | 1.0 | 0.54±0.12 | **0.93**±0.05 | 0.94±0.05 | 0.84±0.06 | 0.91±0.10 | 0.75±0.10 | 0.32±0.05 | 0.40±0.11 | 0.44±0.15 |
| | 1.0 | 0.0 | **0.92**±0.08 | 0.91±0.07 | **0.99**±0.02 | 0.75±0.08 | 0.91±0.08 | 0.76±0.11 | 0.80±0.09 | 0.70±0.14 | 0.48±0.14 |
| | | 0.1 | 0.81±0.13 | **0.92**±0.07 | **0.99**±0.02 | 0.76±0.08 | 0.94±0.06 | 0.76±0.12 | 0.63±0.12 | 0.70±0.13 | 0.47±0.13 |
| | | 0.5 | 0.57±0.16 | **0.92**±0.07 | 0.95±0.04 | 0.85±0.08 | 0.90±0.12 | 0.75±0.11 | 0.31±0.07 | 0.45±0.13 | 0.45±0.11 |
| | | 1.0 | 0.50±0.15 | **0.90**±0.08 | 0.93±0.05 | 0.87±0.08 | 0.89±0.15 | 0.73±0.12 | 0.29±0.06 | 0.35±0.09 | 0.48±0.15 |
| $\underline{O} \cup (\sigma_m^\beta = 0)$ | 0.2 | 0.0 | **0.96**±0.04 | 0.95±0.05 | 1.00±0.02 | 0.72±0.06 | **1.00**±0.00 | 0.78±0.09 | 0.89±0.07 | 0.26±0.02 | 0.26±0.02 |
| | | 0.1 | 0.68±0.13 | 0.89±0.07 | 0.98±0.03 | 0.73±0.06 | **1.00**±0.00 | 0.78±0.09 | **0.57**±0.09 | 0.27±0.02 | 0.26±0.02 |
| | | 0.5 | 0.26±0.04 | 0.94±0.06 | 0.93±0.05 | 0.77±0.07 | **0.98**±0.03 | 0.77±0.09 | 0.30±0.05 | 0.28±0.04 | 0.26±0.02 |
| | | 1.0 | 0.31±0.07 | 0.94±0.05 | 0.95±0.05 | 0.81±0.07 | **0.96**±0.04 | 0.77±0.08 | 0.33±0.07 | 0.26±0.02 | **0.58**±0.14 |
| | 0.5 | 0.0 | **0.94**±0.06 | 0.93±0.05 | 1.00±0.01 | 0.75±0.07 | **1.00**±0.00 | 0.77±0.10 | 0.85±0.06 | 0.26±0.02 | 0.26±0.02 |
| | | 0.1 | 0.67±0.13 | 0.90±0.06 | 0.98±0.02 | 0.76±0.08 | **1.00**±0.00 | 0.76±0.10 | 0.47±0.11 | 0.27±0.02 | 0.26±0.02 |
| | | 0.5 | 0.30±0.04 | **0.92**±0.07 | 0.94±0.05 | 0.81±0.06 | **0.99**±0.03 | 0.76±0.09 | 0.30±0.03 | 0.28±0.03 | 0.33±0.08 |
| | | 1.0 | 0.30±0.07 | **0.92**±0.06 | 0.95±0.04 | 0.83±0.07 | 0.95±0.05 | 0.76±0.10 | 0.32±0.07 | 0.27±0.03 | 0.41±0.12 |
| | 1.0 | 0.0 | **0.92**±0.07 | 0.89±0.09 | 1.00±0.01 | 0.77±0.08 | **1.00**±0.00 | 0.76±0.11 | 0.81±0.09 | 0.26±0.02 | 0.26±0.02 |
| | | 0.1 | 0.67±0.14 | 0.86±0.09 | 0.98±0.03 | 0.77±0.08 | **1.00**±0.00 | 0.75±0.13 | 0.51±0.11 | 0.26±0.02 | 0.26±0.02 |
| | | 0.5 | 0.26±0.05 | **0.88**±0.09 | 0.93±0.05 | 0.83±0.08 | **0.98**±0.04 | 0.75±0.11 | 0.28±0.04 | 0.26±0.02 | 0.34±0.09 |
| | | 1.0 | 0.30±0.06 | **0.88**±0.09 | 0.94±0.05 | 0.87±0.08 | 0.93±0.06 | 0.74±0.12 | 0.32±0.05 | 0.26±0.02 | 0.46±0.15 |
| $O \cap (\sigma_m^\beta > 0)$ | 0.2 | 0.0 | 0.87±0.09 | **0.96**±0.04 | **0.98**±0.03 | 0.75±0.07 | 0.64±0.15 | 0.76±0.10 | 0.92±0.06 | 0.92±0.09 | 0.61±0.14 |
| | | 0.1 | 0.87±0.09 | **0.96**±0.04 | **0.98**±0.03 | 0.76±0.08 | 0.66±0.17 | 0.76±0.10 | 0.90±0.06 | 0.91±0.04 | 0.64±0.12 |
| | | 0.5 | 0.86±0.10 | **0.96**±0.05 | **0.98**±0.04 | 0.77±0.07 | 0.44±0.15 | 0.76±0.11 | 0.84±0.13 | 0.89±0.08 | 0.67±0.10 |
| | | 1.0 | 0.82±0.12 | **0.94**±0.05 | **0.98**±0.03 | 0.77±0.08 | 0.35±0.13 | 0.76±0.11 | 0.86±0.08 | 0.93±0.06 | 0.64±0.11 |
| | 0.5 | 0.0 | 0.82±0.12 | **0.92**±0.07 | **0.98**±0.03 | 0.77±0.08 | 0.88±0.10 | 0.74±0.12 | 0.80±0.13 | 0.81±0.12 | 0.43±0.15 |
| | | 0.1 | 0.83±0.13 | **0.94**±0.06 | 0.97±0.04 | 0.77±0.08 | 0.87±0.10 | 0.74±0.12 | 0.77±0.18 | 0.79±0.14 | 0.57±0.17 |
| | | 0.5 | 0.82±0.14 | **0.93**±0.07 | 0.97±0.04 | 0.79±0.07 | 0.69±0.15 | 0.74±0.11 | 0.78±0.10 | 0.79±0.09 | 0.40±0.11 |
| | | 1.0 | 0.78±0.15 | 0.91±0.08 | 0.97±0.04 | 0.79±0.09 | 0.65±0.16 | 0.74±0.12 | 0.81±0.08 | 0.77±0.12 | 0.52±0.20 |
| | 1.0 | 0.0 | 0.78±0.15 | **0.91**±0.08 | **0.97**±0.04 | 0.79±0.09 | 0.84±0.11 | 0.73±0.12 | 0.75±0.17 | 0.66±0.18 | 0.45±0.15 |
| | | 0.1 | 0.77±0.16 | 0.90±0.09 | 0.96±0.06 | 0.80±0.08 | 0.83±0.12 | 0.73±0.12 | 0.72±0.16 | 0.68±0.18 | 0.51±0.17 |
| | | 0.5 | 0.77±0.16 | **0.91**±0.09 | 0.97±0.04 | 0.81±0.08 | 0.69±0.14 | 0.74±0.12 | 0.77±0.13 | 0.73±0.15 | 0.53±0.13 |
| | | 1.0 | 0.79±0.15 | 0.91±0.08 | **0.97**±0.04 | 0.82±0.09 | 0.63±0.15 | 0.74±0.13 | 0.70±0.20 | 0.67±0.20 | 0.48±0.15 |
| $\underline{O} \cup (\sigma_m^\beta > 0)$ | 0.2 | 0.0 | 0.35±0.14 | **0.96**±0.05 | 0.82±0.10 | 0.95±0.07 | **0.97**±0.05 | 0.37±0.14 | 0.39±0.11 | 0.41±0.13 | 0.30±0.05 |
| | | 0.1 | 0.38±0.15 | **0.95**±0.04 | 0.84±0.08 | 0.95±0.06 | **0.97**±0.05 | 0.41±0.16 | 0.43±0.12 | 0.47±0.12 | 0.31±0.06 |
| | | 0.5 | 0.65±0.17 | **0.95**±0.05 | 0.94±0.06 | 0.85±0.09 | **0.96**±0.05 | 0.62±0.16 | 0.67±0.15 | 0.75±0.14 | 0.46±0.16 |
| | | 1.0 | 0.80±0.12 | **0.96**±0.04 | **0.97**±0.04 | 0.79±0.08 | 0.94±0.06 | 0.71±0.12 | 0.85±0.10 | 0.83±0.12 | 0.54±0.19 |
| | 0.5 | 0.0 | 0.35±0.14 | **0.94**±0.06 | 0.82±0.10 | 0.96±0.06 | **0.97**±0.05 | 0.36±0.14 | 0.39±0.10 | 0.40±0.10 | 0.28±0.04 |
| | | 0.1 | 0.39±0.16 | **0.93**±0.06 | 0.83±0.09 | 0.95±0.07 | **0.97**±0.05 | 0.38±0.14 | 0.42±0.11 | 0.40±0.10 | 0.30±0.06 |
| | | 0.5 | 0.60±0.19 | **0.93**±0.06 | 0.94±0.05 | 0.86±0.08 | **0.96**±0.05 | 0.61±0.17 | 0.62±0.11 | 0.64±0.16 | 0.39±0.12 |
| | | 1.0 | 0.75±0.16 | **0.93**±0.07 | **0.96**±0.04 | 0.82±0.08 | 0.95±0.06 | 0.70±0.13 | 0.73±0.13 | 0.73±0.18 | 0.42±0.16 |
| | 1.0 | 0.0 | 0.35±0.15 | **0.91**±0.09 | 0.81±0.10 | 0.97±0.06 | **0.97**±0.05 | 0.36±0.14 | 0.39±0.12 | 0.36±0.10 | 0.31±0.09 |
| | | 0.1 | 0.37±0.17 | **0.91**±0.08 | 0.83±0.10 | 0.95±0.06 | **0.97**±0.05 | 0.39±0.15 | 0.40±0.13 | 0.38±0.12 | 0.30±0.09 |
| | | 0.5 | 0.60±0.20 | 0.91±0.08 | **0.93**±0.06 | 0.87±0.08 | **0.96**±0.06 | 0.60±0.16 | 0.53±0.15 | 0.58±0.18 | 0.36±0.11 |
| | | 1.0 | 0.73±0.17 | 0.91±0.08 | **0.97**±0.04 | 0.84±0.08 | 0.93±0.06 | 0.69±0.14 | 0.70±0.14 | 0.61±0.16 | 0.44±0.15 |

*Table 7.* F1 Score - $N = 10$, $\sigma_g^a = 0.5$

| Dataset | C | $\sigma_b$ | Granger | PLACy | PCMCI | RCV-VarLiNGAM | DYNOTEARS | PCMCI$\Omega$ | BCGeweke | DTF | GewekeNP |
|---|---|---|---|---|---|---|---|---|---|---|---|
| $O\cap(\sigma_m^\beta=0)$ | 0.2 | 0.0 | **0.87**±0.07 | 0.17±0.12 | 0.05±0.09 | 0.46±0.08 | 0.02±0.05 | 0.07±0.07 | 0.78±0.02 | 0.42±0.14 | 0.54±0.02 |
| | | 0.1 | 0.46±0.09 | 0.22±0.13 | 0.09±0.10 | 0.45±0.08 | 0.12±0.09 | 0.08±0.06 | 0.50±0.06 | 0.33±0.05 | **0.55**±0.03 |
| | | 0.5 | 0.41±0.07 | 0.16±0.11 | 0.24±0.16 | 0.40±0.09 | 0.23±0.08 | 0.10±0.07 | 0.43±0.04 | 0.36±0.05 | **0.54**±0.01 |
| | | 1.0 | 0.51±0.09 | 0.13±0.10 | 0.11±0.10 | 0.41±0.09 | 0.21±0.09 | 0.12±0.06 | 0.50±0.05 | 0.39±0.06 | **0.55**±0.02 |
| | 0.5 | 0.0 | 0.67±0.08 | **0.80**±0.07 | 0.16±0.11 | 0.43±0.08 | 0.01±0.02 | 0.10±0.06 | 0.62±0.05 | 0.53±0.02 | 0.43±0.03 |
| | | 0.1 | 0.41±0.07 | **0.76**±0.07 | 0.12±0.11 | 0.41±0.09 | 0.04±0.07 | 0.10±0.06 | 0.47±0.05 | 0.39±0.08 | 0.43±0.02 |
| | | 0.5 | 0.36±0.06 | **0.63**±0.06 | 0.29±0.13 | 0.36±0.09 | 0.07±0.10 | 0.14±0.06 | 0.42±0.03 | 0.39±0.04 | 0.43±0.02 |
| | | 1.0 | 0.41±0.06 | 0.44±0.12 | 0.19±0.12 | 0.36±0.11 | 0.08±0.10 | 0.20±0.06 | **0.46**±0.02 | 0.41±0.04 | 0.44±0.02 |
| | 1.0 | 0.0 | **0.59**±0.09 | **0.59**±0.09 | 0.38±0.18 | 0.38±0.08 | 0.00±0.02 | 0.12±0.06 | 0.58±0.06 | 0.46±0.04 | 0.41±0.03 |
| | | 0.1 | 0.40±0.08 | **0.59**±0.09 | 0.21±0.14 | 0.38±0.08 | 0.04±0.07 | 0.12±0.06 | 0.44±0.03 | 0.41±0.04 | 0.42±0.03 |
| | | 0.5 | 0.33±0.06 | **0.58**±0.06 | 0.32±0.14 | 0.31±0.10 | 0.12±0.11 | 0.19±0.07 | 0.41±0.03 | 0.40±0.03 | 0.41±0.04 |
| | | 1.0 | 0.36±0.06 | **0.53**±0.09 | 0.23±0.14 | 0.28±0.10 | 0.13±0.12 | 0.24±0.07 | 0.42±0.03 | 0.39±0.04 | 0.41±0.03 |
| $\underline{O}\cap(\sigma_m^\beta=0)$ | 0.2 | 0.0 | **0.87**±0.07 | 0.26±0.12 | 0.06±0.09 | 0.46±0.09 | 0.00±0.00 | 0.07±0.06 | 0.77±0.06 | 0.40±0.03 | 0.40±0.03 |
| | | 0.1 | 0.34±0.06 | 0.35±0.10 | 0.12±0.11 | **0.46**±0.08 | 0.00±0.00 | 0.07±0.06 | 0.45±0.03 | 0.40±0.03 | 0.40±0.03 |
| | | 0.5 | 0.28±0.05 | 0.20±0.11 | 0.25±0.14 | 0.41±0.10 | 0.01±0.04 | 0.11±0.07 | 0.42±0.04 | 0.40±0.03 | **0.43**±0.03 |
| | | 1.0 | 0.37±0.05 | 0.19±0.11 | 0.15±0.11 | 0.40±0.09 | 0.04±0.06 | 0.12±0.06 | **0.50**±0.02 | 0.40±0.03 | 0.47±0.01 |
| | 0.5 | 0.0 | 0.67±0.07 | **0.73**±0.07 | 0.17±0.13 | 0.44±0.08 | 0.00±0.00 | 0.10±0.06 | 0.61±0.07 | 0.40±0.03 | 0.40±0.03 |
| | | 0.1 | 0.33±0.05 | **0.57**±0.07 | 0.15±0.12 | 0.42±0.09 | 0.00±0.00 | 0.11±0.06 | 0.44±0.04 | 0.40±0.03 | 0.40±0.03 |
| | | 0.5 | 0.28±0.05 | **0.57**±0.05 | 0.28±0.12 | 0.35±0.10 | 0.00±0.01 | 0.15±0.07 | 0.42±0.03 | 0.40±0.03 | 0.41±0.03 |
| | | 1.0 | 0.34±0.04 | 0.45±0.10 | 0.18±0.13 | 0.33±0.09 | 0.02±0.05 | 0.21±0.07 | **0.46**±0.03 | 0.40±0.03 | 0.43±0.02 |
| | 1.0 | 0.0 | **0.59**±0.09 | 0.54±0.08 | 0.38±0.19 | 0.37±0.08 | 0.00±0.00 | 0.13±0.06 | 0.58±0.06 | 0.39±0.03 | 0.39±0.03 |
| | | 0.1 | 0.32±0.06 | **0.47**±0.07 | 0.22±0.13 | 0.36±0.08 | 0.00±0.00 | 0.13±0.07 | 0.42±0.03 | 0.39±0.03 | 0.39±0.03 |
| | | 0.5 | 0.28±0.05 | **0.53**±0.08 | 0.29±0.14 | 0.32±0.11 | 0.00±0.02 | 0.18±0.06 | 0.41±0.03 | 0.39±0.03 | 0.40±0.03 |
| | | 1.0 | 0.32±0.05 | **0.48**±0.07 | 0.21±0.12 | 0.30±0.11 | 0.01±0.03 | 0.24±0.06 | 0.42±0.03 | 0.39±0.03 | 0.41±0.03 |
| $O\cap(\sigma_m^\beta>0)$ | 0.2 | 0.0 | 0.74±0.09 | 0.55±0.14 | 0.13±0.13 | 0.44±0.07 | 0.15±0.09 | 0.11±0.06 | **0.75**±0.07 | 0.40±0.11 | 0.51±0.05 |
| | | 0.1 | 0.68±0.10 | 0.63±0.13 | 0.15±0.11 | 0.43±0.07 | 0.14±0.08 | 0.13±0.07 | **0.74**±0.08 | 0.38±0.09 | 0.47±0.05 |
| | | 0.5 | 0.66±0.09 | 0.68±0.11 | 0.14±0.12 | 0.44±0.08 | 0.20±0.07 | 0.14±0.06 | **0.68**±0.05 | 0.38±0.09 | 0.49±0.04 |
| | | 1.0 | 0.66±0.09 | 0.65±0.12 | 0.11±0.13 | 0.43±0.07 | 0.22±0.07 | 0.12±0.07 | **0.72**±0.07 | 0.40±0.15 | 0.43±0.03 |
| | 0.5 | 0.0 | 0.60±0.09 | **0.72**±0.09 | 0.31±0.19 | 0.42±0.07 | 0.03±0.05 | 0.16±0.07 | 0.62±0.07 | 0.52±0.05 | 0.44±0.03 |
| | | 0.1 | 0.57±0.08 | **0.69**±0.09 | 0.30±0.15 | 0.40±0.08 | 0.02±0.05 | 0.17±0.07 | 0.66±0.06 | 0.48±0.02 | 0.44±0.04 |
| | | 0.5 | 0.56±0.08 | **0.69**±0.08 | 0.28±0.16 | 0.39±0.09 | 0.08±0.09 | 0.17±0.07 | 0.64±0.08 | 0.54±0.03 | 0.46±0.02 |
| | | 1.0 | 0.57±0.10 | **0.68**±0.08 | 0.28±0.16 | 0.39±0.10 | 0.08±0.08 | 0.18±0.07 | 0.61±0.04 | 0.48±0.05 | 0.45±0.01 |
| | 1.0 | 0.0 | 0.53±0.10 | **0.58**±0.09 | 0.54±0.17 | 0.34±0.09 | 0.05±0.06 | 0.18±0.08 | 0.57±0.05 | 0.47±0.05 | 0.42±0.03 |
| | | 0.1 | 0.51±0.10 | **0.58**±0.08 | 0.51±0.16 | 0.35±0.10 | 0.03±0.05 | 0.18±0.07 | 0.57±0.07 | 0.48±0.05 | 0.42±0.04 |
| | | 0.5 | 0.48±0.10 | **0.58**±0.09 | 0.51±0.16 | 0.32±0.10 | 0.07±0.08 | 0.21±0.07 | 0.53±0.06 | 0.45±0.04 | 0.42±0.03 |
| | | 1.0 | 0.49±0.08 | **0.58**±0.08 | 0.47±0.16 | 0.31±0.11 | 0.09±0.09 | 0.21±0.07 | 0.55±0.05 | 0.46±0.06 | 0.41±0.04 |
| $\underline{O}\cap(\sigma_m^\beta>0)$ | 0.2 | 0.0 | 0.28±0.05 | **0.59**±0.14 | 0.34±0.12 | 0.12±0.13 | 0.03±0.06 | 0.26±0.05 | 0.43±0.03 | 0.42±0.03 | 0.40±0.04 |
| | | 0.1 | 0.28±0.05 | **0.67**±0.12 | 0.33±0.13 | 0.14±0.13 | 0.02±0.06 | 0.26±0.05 | 0.45±0.06 | 0.44±0.03 | 0.40±0.03 |
| | | 0.5 | 0.42±0.09 | **0.68**±0.11 | 0.19±0.12 | 0.39±0.09 | 0.03±0.06 | 0.20±0.07 | 0.58±0.08 | 0.46±0.03 | 0.44±0.03 |
| | | 1.0 | 0.60±0.11 | **0.66**±0.07 | 0.12±0.12 | 0.44±0.08 | 0.05±0.07 | 0.16±0.07 | **0.66**±0.07 | 0.45±0.04 | 0.48±0.03 |
| | 0.5 | 0.0 | 0.28±0.05 | **0.70**±0.09 | 0.48±0.13 | 0.12±0.13 | 0.01±0.04 | 0.27±0.05 | 0.42±0.04 | 0.42±0.03 | 0.40±0.03 |
| | | 0.1 | 0.28±0.05 | **0.69**±0.09 | 0.47±0.13 | 0.14±0.13 | 0.01±0.03 | 0.27±0.05 | 0.44±0.03 | 0.42±0.03 | 0.40±0.03 |
| | | 0.5 | 0.39±0.08 | **0.70**±0.09 | 0.40±0.16 | 0.34±0.10 | 0.01±0.03 | 0.22±0.06 | 0.56±0.05 | 0.43±0.03 | 0.42±0.02 |
| | | 1.0 | 0.53±0.08 | **0.69**±0.10 | 0.27±0.17 | 0.38±0.10 | 0.01±0.03 | 0.18±0.07 | 0.60±0.02 | 0.48±0.05 | 0.44±0.01 |
| | 1.0 | 0.0 | 0.28±0.05 | **0.59**±0.09 | 0.56±0.10 | 0.08±0.10 | 0.01±0.03 | 0.26±0.05 | 0.41±0.03 | 0.40±0.03 | 0.39±0.03 |
| | | 0.1 | 0.28±0.05 | **0.59**±0.09 | 0.56±0.09 | 0.11±0.10 | 0.01±0.03 | 0.27±0.05 | 0.41±0.03 | 0.40±0.03 | 0.39±0.03 |
| | | 0.5 | 0.36±0.07 | **0.60**±0.10 | 0.54±0.13 | 0.27±0.10 | 0.00±0.02 | 0.24±0.06 | 0.48±0.06 | 0.42±0.03 | 0.40±0.03 |
| | | 1.0 | 0.47±0.08 | **0.58**±0.09 | 0.47±0.15 | 0.29±0.09 | 0.01±0.02 | 0.21±0.06 | 0.54±0.06 | 0.43±0.04 | 0.41±0.04 |

*Table 8.* TNR Score - $N = 10$, $\sigma_g^a = 0.5$

| Dataset | C | $\sigma_b$ | Granger | PLACy | PCMCI | RCV-VarLiNGAM | DYNOTEARS | PCMCI$_{\Omega_2}$ | BCGeweke | DTF | GewekeNP |
|---|---|---|---|---|---|---|---|---|---|---|---|
| $O \cap (\sigma_m^\beta = 0)$ | 0.2 | 0.0 | **0.95**±0.02 | 0.95±0.02 | **1.00**±0.01 | 0.69±0.04 | 0.97±0.02 | 0.72±0.06 | 0.83±0.04 | 0.83±0.06 | 0.50±0.07 |
| | | 0.1 | 0.61±0.09 | 0.95±0.03 | 0.98±0.01 | 0.69±0.04 | 0.85±0.09 | 0.73±0.06 | 0.41±0.12 | 0.74±0.09 | **0.53**±0.10 |
| | | 0.5 | 0.54±0.08 | 0.95±0.02 | 0.97±0.02 | 0.77±0.04 | 0.65±0.12 | 0.72±0.06 | 0.22±0.05 | 0.41±0.08 | **0.52**±0.08 |
| | | 1.0 | **0.68**±0.08 | 0.93±0.03 | **0.99**±0.01 | 0.80±0.04 | 0.64±0.12 | 0.72±0.06 | 0.41±0.06 | 0.21±0.04 | 0.52±0.10 |
| | 0.5 | 0.0 | 0.83±0.08 | **0.93**±0.04 | **0.99**±0.01 | 0.75±0.05 | 0.93±0.04 | 0.61±0.10 | 0.63±0.11 | 0.54±0.08 | 0.23±0.07 |
| | | 0.1 | 0.53±0.12 | **0.93**±0.04 | 0.98±0.02 | 0.78±0.05 | 0.95±0.05 | 0.62±0.10 | 0.32±0.17 | 0.56±0.12 | 0.23±0.07 |
| | | 0.5 | 0.42±0.11 | **0.93**±0.02 | 0.95±0.02 | 0.82±0.04 | **0.96**±0.07 | 0.63±0.09 | 0.18±0.06 | 0.23±0.07 | 0.22±0.06 |
| | | 1.0 | 0.52±0.10 | 0.90±0.04 | **0.98**±0.02 | 0.84±0.04 | 0.97±0.05 | 0.61±0.09 | 0.30±0.08 | 0.19±0.04 | 0.24±0.09 |
| | 1.0 | 0.0 | 0.76±0.11 | **0.77**±0.11 | **0.98**±0.02 | 0.77±0.05 | 0.88±0.05 | 0.61±0.10 | 0.59±0.09 | 0.34±0.12 | 0.20±0.07 |
| | | 0.1 | 0.50±0.15 | **0.78**±0.11 | 0.97±0.02 | 0.80±0.05 | 0.89±0.05 | 0.58±0.11 | 0.28±0.09 | 0.29±0.09 | 0.21±0.08 |
| | | 0.5 | 0.34±0.12 | **0.79**±0.12 | 0.94±0.03 | 0.85±0.05 | 0.93±0.10 | 0.58±0.10 | 0.18±0.04 | 0.19±0.07 | 0.19±0.07 |
| | | 1.0 | 0.42±0.13 | **0.78**±0.12 | 0.97±0.02 | 0.89±0.05 | 0.94±0.07 | 0.55±0.11 | 0.21±0.06 | 0.15±0.03 | 0.20±0.07 |
| $\underline{O} \cap (\sigma_m^\beta = 0)$ | 0.2 | 0.0 | **0.95**±0.03 | 0.92±0.03 | **1.00**±0.00 | 0.69±0.04 | **1.00**±0.00 | 0.73±0.06 | 0.83±0.06 | 0.13±0.00 | 0.13±0.00 |
| | | 0.1 | 0.37±0.07 | 0.83±0.04 | 0.98±0.02 | 0.69±0.04 | **1.00**±0.00 | 0.73±0.06 | 0.29±0.09 | 0.13±0.00 | 0.13±0.00 |
| | | 0.5 | 0.15±0.03 | 0.90±0.04 | 0.97±0.02 | 0.76±0.04 | **0.99**±0.01 | 0.72±0.05 | 0.19±0.02 | 0.13±0.00 | 0.23±0.07 |
| | | 1.0 | 0.45±0.08 | 0.90±0.03 | **0.99**±0.01 | 0.79±0.04 | 0.95±0.02 | 0.72±0.05 | 0.40±0.08 | 0.13±0.00 | 0.35±0.11 |
| | 0.5 | 0.0 | 0.83±0.08 | **0.88**±0.05 | 1.00±0.01 | 0.76±0.05 | **1.00**±0.00 | 0.61±0.10 | 0.62±0.11 | 0.13±0.00 | 0.13±0.00 |
| | | 0.1 | 0.32±0.08 | **0.78**±0.06 | 0.96±0.02 | 0.77±0.05 | **1.00**±0.00 | 0.62±0.10 | 0.25±0.07 | 0.13±0.00 | 0.13±0.00 |
| | | 0.5 | 0.15±0.03 | **0.87**±0.05 | 0.95±0.02 | 0.82±0.04 | **0.99**±0.02 | 0.64±0.10 | 0.20±0.04 | 0.13±0.00 | 0.16±0.05 |
| | | 1.0 | 0.35±0.09 | **0.86**±0.05 | 0.98±0.02 | 0.85±0.04 | 0.94±0.03 | 0.61±0.10 | 0.31±0.08 | 0.13±0.00 | 0.23±0.08 |
| | 1.0 | 0.0 | **0.76**±0.11 | 0.73±0.11 | 0.98±0.02 | 0.79±0.06 | **1.00**±0.00 | 0.61±0.10 | 0.58±0.09 | 0.13±0.00 | 0.13±0.00 |
| | | 0.1 | 0.31±0.09 | 0.64±0.11 | 0.96±0.03 | 0.80±0.05 | **1.00**±0.00 | 0.58±0.11 | 0.23±0.07 | 0.13±0.00 | 0.13±0.00 |
| | | 0.5 | 0.16±0.11 | **0.74**±0.11 | 0.94±0.03 | 0.84±0.05 | **0.99**±0.02 | 0.57±0.11 | 0.18±0.03 | 0.13±0.00 | 0.14±0.02 |
| | | 1.0 | 0.31±0.09 | 0.71±0.13 | **0.97**±0.02 | 0.89±0.05 | 0.93±0.04 | 0.56±0.11 | 0.22±0.06 | 0.13±0.00 | 0.19±0.07 |
| $O \cap (\sigma_m^\beta > 0)$ | 0.2 | 0.0 | 0.88±0.05 | **0.95**±0.02 | **0.99**±0.01 | 0.71±0.04 | 0.77±0.06 | 0.70±0.07 | 0.83±0.05 | 0.82±0.09 | 0.49±0.12 |
| | | 0.1 | 0.85±0.06 | **0.95**±0.02 | **0.99**±0.01 | 0.72±0.05 | 0.79±0.09 | 0.68±0.08 | 0.83±0.05 | 0.80±0.10 | 0.48±0.11 |
| | | 0.5 | 0.83±0.06 | **0.95**±0.03 | 0.98±0.01 | 0.72±0.04 | 0.61±0.17 | 0.68±0.07 | 0.76±0.03 | 0.84±0.06 | 0.42±0.15 |
| | | 1.0 | 0.84±0.08 | **0.95**±0.03 | **0.99**±0.01 | 0.72±0.04 | 0.58±0.16 | 0.68±0.07 | 0.82±0.05 | 0.87±0.05 | 0.38±0.14 |
| | 0.5 | 0.0 | 0.77±0.11 | **0.88**±0.06 | **0.99**±0.01 | 0.75±0.05 | 0.92±0.04 | 0.61±0.10 | 0.65±0.09 | 0.60±0.13 | 0.30±0.11 |
| | | 0.1 | 0.75±0.09 | **0.87**±0.07 | 0.98±0.02 | 0.76±0.06 | 0.94±0.05 | 0.60±0.11 | 0.69±0.09 | 0.62±0.16 | 0.29±0.11 |
| | | 0.5 | 0.72±0.11 | **0.87**±0.06 | 0.98±0.01 | 0.77±0.05 | 0.85±0.10 | 0.60±0.10 | 0.66±0.12 | 0.63±0.14 | 0.30±0.12 |
| | | 1.0 | 0.73±0.13 | **0.86**±0.07 | 0.98±0.02 | 0.77±0.05 | 0.84±0.12 | 0.59±0.10 | 0.61±0.12 | 0.63±0.16 | 0.28±0.11 |
| | 1.0 | 0.0 | 0.68±0.15 | 0.77±0.11 | **0.96**±0.04 | 0.79±0.05 | 0.87±0.06 | 0.60±0.10 | 0.57±0.10 | 0.39±0.12 | 0.22±0.08 |
| | | 0.1 | 0.65±0.15 | 0.77±0.10 | **0.96**±0.04 | 0.80±0.05 | 0.88±0.06 | 0.59±0.11 | 0.57±0.11 | 0.43±0.13 | 0.22±0.09 |
| | | 0.5 | 0.62±0.17 | 0.77±0.11 | **0.96**±0.03 | 0.81±0.05 | 0.83±0.11 | 0.57±0.10 | 0.50±0.12 | 0.36±0.13 | 0.21±0.08 |
| | | 1.0 | 0.64±0.14 | **0.77**±0.10 | 0.96±0.04 | 0.82±0.05 | 0.80±0.14 | 0.57±0.11 | 0.54±0.10 | 0.39±0.15 | 0.22±0.06 |
| $\underline{O} \cap (\sigma_m^\beta > 0)$ | 0.2 | 0.0 | 0.17±0.07 | 0.94±0.03 | 0.88±0.04 | 0.96±0.04 | **0.97**±0.03 | 0.18±0.06 | 0.24±0.06 | 0.25±0.07 | 0.13±0.01 |
| | | 0.1 | 0.18±0.07 | **0.95**±0.03 | 0.90±0.04 | 0.95±0.05 | 0.97±0.03 | 0.20±0.07 | 0.30±0.12 | 0.29±0.10 | 0.14±0.02 |
| | | 0.5 | 0.55±0.12 | **0.95**±0.03 | 0.96±0.02 | 0.78±0.06 | 0.96±0.03 | 0.49±0.10 | 0.63±0.08 | 0.56±0.09 | 0.34±0.11 |
| | | 1.0 | 0.79±0.08 | **0.95**±0.02 | 0.98±0.02 | 0.73±0.05 | 0.94±0.04 | 0.65±0.08 | 0.73±0.03 | 0.69±0.09 | 0.43±0.11 |
| | 0.5 | 0.0 | 0.16±0.07 | **0.88**±0.06 | 0.88±0.04 | 0.96±0.04 | **0.98**±0.03 | 0.16±0.05 | 0.21±0.05 | 0.18±0.03 | 0.13±0.00 |
| | | 0.1 | 0.17±0.07 | **0.87**±0.07 | 0.90±0.04 | 0.95±0.05 | **0.98**±0.03 | 0.18±0.06 | 0.26±0.06 | 0.18±0.03 | 0.14±0.02 |
| | | 0.5 | 0.46±0.14 | **0.87**±0.07 | 0.97±0.02 | 0.81±0.05 | 0.96±0.04 | 0.44±0.11 | 0.54±0.07 | 0.35±0.06 | 0.21±0.05 |
| | | 1.0 | 0.70±0.10 | **0.87**±0.07 | 0.98±0.01 | 0.78±0.05 | 0.93±0.04 | 0.57±0.11 | 0.61±0.08 | 0.48±0.13 | 0.26±0.08 |
| | 1.0 | 0.0 | 0.15±0.07 | 0.78±0.10 | 0.86±0.05 | 0.97±0.04 | **0.98**±0.03 | 0.16±0.05 | 0.19±0.04 | 0.15±0.02 | 0.13±0.00 |
| | | 0.1 | 0.16±0.08 | 0.78±0.11 | 0.87±0.05 | 0.96±0.04 | **0.98**±0.03 | 0.19±0.07 | 0.20±0.05 | 0.14±0.01 | 0.13±0.00 |
| | | 0.5 | 0.39±0.14 | 0.78±0.11 | **0.94**±0.04 | 0.85±0.05 | 0.96±0.04 | 0.41±0.12 | 0.39±0.11 | 0.22±0.05 | 0.16±0.04 |
| | | 1.0 | 0.61±0.14 | **0.77**±0.10 | 0.96±0.03 | 0.82±0.05 | 0.92±0.05 | 0.54±0.11 | 0.51±0.11 | 0.29±0.06 | 0.18±0.07 |

Table 9. F1 Score - $N = 10$, $\sigma_g^a = 1.0$

| Dataset | C | $\sigma_b$ | Granger | PLACy | PCMCI | RCV-VarLiNGAM | DYNOTEARS | PCMCI$_\Omega$ | BCGeweke | DTF | GewekeNP |
|---|---|---|---|---|---|---|---|---|---|---|---|
| $O \cap (\sigma_m^\beta = 0)$ | 0.2 | 0.0 | **0.88**±0.07 | 0.16±0.11 | 0.06±0.09 | 0.46±0.08 | 0.08±0.09 | 0.07±0.06 | 0.78±0.02 | 0.43±0.14 | 0.55±0.01 |
| | | 0.1 | **0.65**±0.10 | 0.17±0.11 | 0.06±0.08 | 0.46±0.08 | 0.13±0.09 | 0.07±0.06 | 0.62±0.05 | 0.33±0.09 | 0.55±0.01 |
| | | 0.5 | 0.46±0.09 | 0.17±0.11 | 0.20±0.13 | 0.44±0.08 | 0.21±0.09 | 0.09±0.06 | 0.46±0.04 | 0.36±0.05 | **0.56**±0.02 |
| | | 1.0 | 0.41±0.07 | 0.16±0.12 | 0.24±0.12 | 0.41±0.09 | 0.23±0.09 | 0.11±0.06 | 0.44±0.03 | 0.39±0.06 | **0.55**±0.01 |
| | 0.5 | 0.0 | 0.67±0.08 | **0.80**±0.07 | 0.16±0.14 | 0.43±0.08 | 0.03±0.06 | 0.10±0.06 | 0.62±0.06 | 0.53±0.03 | 0.43±0.03 |
| | | 0.1 | 0.55±0.08 | **0.79**±0.07 | 0.10±0.11 | 0.43±0.09 | 0.04±0.07 | 0.10±0.06 | 0.57±0.06 | 0.49±0.05 | 0.44±0.02 |
| | | 0.5 | 0.39±0.07 | **0.73**±0.07 | 0.28±0.13 | 0.38±0.09 | 0.07±0.09 | 0.12±0.06 | 0.44±0.02 | 0.37±0.06 | 0.44±0.02 |
| | | 1.0 | 0.36±0.05 | **0.60**±0.11 | 0.27±0.13 | 0.36±0.11 | 0.07±0.10 | 0.14±0.06 | 0.44±0.03 | 0.41±0.04 | 0.43±0.03 |
| | 1.0 | 0.0 | 0.59±0.09 | 0.59±0.09 | 0.38±0.19 | 0.38±0.09 | 0.03±0.07 | 0.12±0.06 | **0.59**±0.06 | 0.47±0.04 | 0.41±0.03 |
| | | 0.1 | 0.53±0.08 | **0.60**±0.09 | 0.24±0.16 | 0.38±0.08 | 0.05±0.08 | 0.13±0.06 | 0.52±0.04 | 0.44±0.05 | 0.41±0.03 |
| | | 0.5 | 0.37±0.06 | **0.59**±0.09 | 0.31±0.13 | 0.31±0.09 | 0.13±0.13 | 0.14±0.07 | 0.40±0.03 | 0.39±0.07 | 0.41±0.03 |
| | | 1.0 | 0.34±0.05 | **0.58**±0.09 | 0.30±0.12 | 0.32±0.10 | 0.14±0.11 | 0.19±0.07 | 0.40±0.03 | 0.40±0.03 | 0.41±0.03 |
| $O \subseteq (\sigma_m^\beta = 0)$ | 0.2 | 0.0 | **0.87**±0.07 | 0.24±0.13 | 0.05±0.08 | 0.45±0.09 | 0.00±0.00 | 0.07±0.06 | 0.75±0.05 | 0.41±0.04 | 0.40±0.04 |
| | | 0.1 | 0.49±0.09 | 0.35±0.11 | 0.07±0.09 | 0.45±0.08 | 0.00±0.00 | 0.08±0.06 | **0.53**±0.02 | 0.41±0.03 | 0.41±0.04 |
| | | 0.5 | 0.28±0.05 | 0.25±0.12 | 0.28±0.12 | **0.44**±0.08 | 0.01±0.03 | 0.09±0.06 | 0.43±0.03 | 0.41±0.04 | 0.43±0.02 |
| | | 1.0 | 0.29±0.05 | 0.20±0.10 | 0.25±0.13 | 0.39±0.09 | 0.04±0.07 | 0.10±0.06 | 0.44±0.03 | 0.41±0.04 | **0.50**±0.01 |
| | 0.5 | 0.0 | 0.67±0.07 | **0.75**±0.07 | 0.17±0.13 | 0.43±0.08 | 0.00±0.00 | 0.10±0.06 | 0.58±0.02 | 0.41±0.04 | 0.40±0.04 |
| | | 0.1 | 0.43±0.09 | **0.65**±0.06 | 0.10±0.12 | 0.43±0.09 | 0.00±0.00 | 0.10±0.06 | 0.52±0.02 | 0.41±0.04 | 0.41±0.04 |
| | | 0.5 | 0.28±0.05 | **0.69**±0.08 | 0.29±0.12 | 0.38±0.10 | 0.00±0.00 | 0.11±0.07 | 0.42±0.04 | 0.40±0.04 | 0.42±0.03 |
| | | 1.0 | 0.29±0.05 | **0.59**±0.09 | 0.29±0.14 | 0.34±0.11 | 0.02±0.04 | 0.14±0.06 | 0.44±0.03 | 0.41±0.04 | 0.44±0.02 |
| | 1.0 | 0.0 | 0.59±0.09 | 0.56±0.08 | 0.38±0.19 | 0.37±0.08 | 0.00±0.00 | 0.13±0.06 | 0.58±0.06 | 0.39±0.03 | 0.39±0.03 |
| | | 0.1 | 0.42±0.08 | **0.51**±0.07 | 0.23±0.17 | 0.36±0.08 | 0.00±0.00 | 0.14±0.06 | 0.47±0.05 | 0.39±0.03 | 0.39±0.03 |
| | | 0.5 | 0.28±0.05 | **0.56**±0.08 | 0.30±0.14 | 0.32±0.10 | 0.01±0.03 | 0.13±0.07 | 0.40±0.03 | 0.39±0.03 | 0.40±0.03 |
| | | 1.0 | 0.29±0.05 | **0.54**±0.08 | 0.31±0.12 | 0.29±0.11 | 0.01±0.03 | 0.19±0.06 | 0.41±0.03 | 0.39±0.03 | 0.41±0.04 |
| $O \cap (\sigma_m^\beta > 0)$ | 0.2 | 0.0 | 0.64±0.11 | 0.69±0.11 | 0.26±0.16 | 0.43±0.08 | 0.17±0.09 | 0.15±0.08 | 0.66±0.08 | 0.48±0.09 | 0.46±0.03 |
| | | 0.1 | 0.61±0.11 | 0.70±0.10 | 0.32±0.15 | 0.42±0.08 | 0.18±0.09 | 0.16±0.07 | 0.60±0.08 | 0.47±0.08 | 0.49±0.01 |
| | | 0.5 | 0.61±0.10 | **0.73**±0.10 | 0.26±0.15 | 0.41±0.08 | 0.23±0.07 | 0.17±0.06 | 0.65±0.07 | 0.43±0.02 | 0.44±0.03 |
| | | 1.0 | 0.60±0.10 | **0.74**±0.09 | 0.28±0.17 | 0.41±0.08 | 0.23±0.07 | 0.18±0.08 | 0.67±0.04 | 0.53±0.05 | 0.45±0.06 |
| | 0.5 | 0.0 | 0.54±0.10 | **0.69**±0.09 | 0.49±0.17 | 0.39±0.08 | 0.07±0.09 | 0.19±0.08 | 0.57±0.04 | 0.51±0.08 | 0.43±0.03 |
| | | 0.1 | 0.54±0.09 | **0.68**±0.08 | 0.47±0.16 | 0.38±0.09 | 0.09±0.09 | 0.20±0.07 | 0.57±0.06 | 0.44±0.06 | 0.43±0.03 |
| | | 0.5 | 0.51±0.10 | **0.66**±0.08 | 0.47±0.15 | 0.34±0.09 | 0.15±0.10 | 0.20±0.08 | 0.61±0.05 | 0.52±0.09 | 0.43±0.03 |
| | | 1.0 | 0.51±0.11 | **0.67**±0.09 | 0.47±0.14 | 0.35±0.09 | 0.19±0.09 | 0.20±0.08 | 0.57±0.05 | 0.50±0.07 | 0.44±0.03 |
| | 1.0 | 0.0 | 0.47±0.09 | 0.58±0.08 | **0.63**±0.14 | 0.31±0.09 | 0.07±0.07 | 0.20±0.07 | 0.53±0.05 | 0.46±0.06 | 0.41±0.03 |
| | | 0.1 | 0.46±0.09 | 0.58±0.09 | **0.63**±0.12 | 0.28±0.10 | 0.07±0.08 | 0.22±0.07 | 0.54±0.06 | 0.46±0.06 | 0.41±0.04 |
| | | 0.5 | 0.46±0.08 | 0.58±0.10 | **0.61**±0.12 | 0.27±0.11 | 0.15±0.09 | 0.23±0.06 | 0.52±0.05 | 0.47±0.07 | 0.41±0.03 |
| | | 1.0 | 0.44±0.08 | 0.58±0.10 | **0.61**±0.13 | 0.28±0.10 | 0.15±0.09 | 0.21±0.07 | 0.49±0.08 | 0.44±0.07 | 0.42±0.03 |
| $O \subseteq (\sigma_m^\beta > 0)$ | 0.2 | 0.0 | 0.30±0.06 | **0.71**±0.10 | 0.40±0.12 | 0.18±0.13 | 0.02±0.05 | 0.26±0.06 | 0.45±0.08 | 0.46±0.05 | 0.41±0.05 |
| | | 0.1 | 0.32±0.06 | **0.71**±0.09 | 0.38±0.11 | 0.21±0.12 | 0.02±0.05 | 0.25±0.06 | 0.45±0.06 | 0.47±0.05 | 0.41±0.04 |
| | | 0.5 | 0.46±0.10 | **0.74**±0.10 | 0.34±0.14 | 0.36±0.10 | 0.03±0.05 | 0.21±0.07 | 0.60±0.09 | 0.50±0.06 | 0.44±0.05 |
| | | 1.0 | 0.56±0.12 | **0.75**±0.10 | 0.27±0.15 | 0.39±0.09 | 0.04±0.06 | 0.18±0.08 | 0.56±0.04 | 0.46±0.03 | 0.45±0.03 |
| | 0.5 | 0.0 | 0.30±0.06 | **0.68**±0.09 | 0.51±0.10 | 0.14±0.12 | 0.01±0.03 | 0.26±0.05 | 0.45±0.07 | 0.45±0.04 | 0.41±0.04 |
| | | 0.1 | 0.31±0.06 | **0.68**±0.09 | 0.51±0.12 | 0.15±0.12 | 0.01±0.03 | 0.25±0.05 | 0.48±0.07 | 0.45±0.03 | 0.41±0.05 |
| | | 0.5 | 0.42±0.09 | **0.67**±0.09 | 0.50±0.13 | 0.30±0.11 | 0.01±0.02 | 0.24±0.07 | 0.54±0.06 | 0.47±0.01 | 0.43±0.03 |
| | | 1.0 | 0.49±0.10 | **0.66**±0.09 | 0.48±0.16 | 0.32±0.10 | 0.02±0.04 | 0.22±0.06 | 0.57±0.04 | 0.44±0.07 | 0.43±0.03 |
| | 1.0 | 0.0 | 0.30±0.05 | **0.59**±0.09 | 0.56±0.09 | 0.11±0.10 | 0.00±0.02 | 0.26±0.05 | 0.42±0.04 | 0.41±0.04 | 0.39±0.04 |
| | | 0.1 | 0.31±0.06 | 0.58±0.10 | **0.58**±0.10 | 0.12±0.11 | 0.00±0.01 | 0.25±0.05 | 0.43±0.05 | 0.41±0.04 | 0.39±0.04 |
| | | 0.5 | 0.38±0.08 | 0.59±0.08 | **0.61**±0.12 | 0.21±0.12 | 0.00±0.02 | 0.23±0.06 | 0.49±0.06 | 0.43±0.04 | 0.40±0.03 |
| | | 1.0 | 0.43±0.08 | 0.58±0.09 | **0.62**±0.11 | 0.24±0.10 | 0.01±0.03 | 0.23±0.07 | 0.52±0.04 | 0.46±0.04 | 0.41±0.04 |

*Table 10. TNR Score - $N = 10$, $\sigma_g^a = 1.0$*

| Dataset | C | $\sigma_b$ | Granger | PLACy | PCMCI | RCV-VarLiNGAM | DYNOTEARS | PCMCI$_\Omega$ | BCGeweke | DTF | GewekeNP |
|---|---|---|---|---|---|---|---|---|---|---|---|
| $O \cap (\sigma_m^\beta = 0)$ | 0.2 | 0.0 | **0.95**±0.02 | 0.95±0.02 | **0.99**±0.01 | 0.69±0.04 | 0.92±0.07 | 0.72±0.06 | 0.83±0.05 | 0.86±0.04 | 0.53±0.09 |
| | | 0.1 | **0.82**±0.06 | 0.96±0.02 | **0.99**±0.01 | 0.69±0.04 | 0.82±0.12 | 0.73±0.06 | 0.62±0.09 | 0.84±0.06 | 0.50±0.08 |
| | | 0.5 | 0.61±0.09 | 0.95±0.02 | **0.96**±0.02 | 0.70±0.05 | 0.63±0.15 | 0.72±0.05 | 0.29±0.03 | 0.56±0.03 | **0.53**±0.09 |
| | | 1.0 | 0.53±0.09 | 0.94±0.02 | **0.97**±0.02 | 0.78±0.04 | 0.63±0.15 | 0.71±0.06 | 0.24±0.03 | 0.36±0.12 | **0.51**±0.09 |
| | 0.5 | 0.0 | 0.83±0.08 | **0.93**±0.04 | **0.99**±0.01 | 0.75±0.05 | 0.94±0.04 | 0.61±0.10 | 0.62±0.12 | 0.55±0.10 | 0.23±0.08 |
| | | 0.1 | 0.72±0.09 | **0.93**±0.04 | **0.99**±0.01 | 0.76±0.05 | 0.95±0.04 | 0.61±0.11 | 0.52±0.15 | 0.65±0.10 | 0.22±0.08 |
| | | 0.5 | 0.49±0.12 | **0.93**±0.03 | 0.95±0.02 | 0.81±0.04 | **0.96**±0.06 | 0.63±0.09 | 0.23±0.05 | 0.43±0.08 | 0.24±0.11 |
| | | 1.0 | 0.42±0.10 | **0.92**±0.04 | 0.96±0.02 | 0.81±0.04 | **0.97**±0.06 | 0.62±0.09 | 0.23±0.06 | 0.24±0.05 | 0.20±0.08 |
| | 1.0 | 0.0 | 0.76±0.11 | **0.77**±0.11 | 0.98±0.02 | 0.77±0.05 | 0.87±0.06 | 0.61±0.11 | 0.60±0.10 | 0.37±0.11 | 0.20±0.07 |
| | | 0.1 | 0.70±0.10 | **0.78**±0.11 | 0.98±0.02 | 0.78±0.05 | 0.89±0.06 | 0.58±0.11 | 0.46±0.11 | 0.38±0.11 | 0.20±0.08 |
| | | 0.5 | 0.42±0.14 | **0.79**±0.11 | 0.95±0.03 | 0.83±0.05 | 0.92±0.09 | 0.59±0.12 | 0.15±0.04 | 0.23±0.06 | 0.19±0.05 |
| | | 1.0 | 0.36±0.11 | **0.79**±0.11 | 0.94±0.03 | 0.84±0.05 | 0.92±0.09 | 0.57±0.11 | 0.16±0.03 | 0.21±0.07 | 0.19±0.07 |
| $\bar{O} \cap (\sigma_m^\beta = 0)$ | 0.2 | 0.0 | **0.95**±0.03 | 0.93±0.03 | 1.00±0.01 | 0.69±0.04 | **1.00**±0.00 | 0.73±0.06 | 0.81±0.04 | 0.13±0.01 | 0.13±0.00 |
| | | 0.1 | 0.65±0.08 | 0.89±0.03 | 0.99±0.01 | 0.69±0.04 | **1.00**±0.00 | 0.73±0.06 | **0.46**±0.09 | 0.13±0.00 | 0.13±0.00 |
| | | 0.5 | 0.15±0.02 | 0.92±0.03 | 0.96±0.02 | 0.71±0.04 | **0.99**±0.01 | 0.72±0.06 | 0.20±0.02 | 0.14±0.01 | 0.19±0.08 |
| | | 1.0 | 0.21±0.04 | 0.92±0.03 | **0.97**±0.02 | 0.77±0.04 | 0.95±0.02 | 0.72±0.05 | 0.22±0.03 | 0.14±0.00 | 0.40±0.11 |
| | 0.5 | 0.0 | 0.83±0.08 | **0.89**±0.05 | 1.00±0.01 | 0.76±0.05 | **1.00**±0.00 | 0.60±0.10 | 0.58±0.06 | 0.13±0.00 | 0.13±0.00 |
| | | 0.1 | 0.55±0.12 | **0.83**±0.06 | 0.98±0.02 | 0.76±0.05 | **1.00**±0.00 | 0.61±0.10 | 0.42±0.14 | 0.13±0.00 | 0.13±0.00 |
| | | 0.5 | 0.15±0.03 | **0.90**±0.05 | 0.94±0.02 | 0.81±0.05 | **0.99**±0.01 | 0.63±0.10 | 0.16±0.03 | 0.13±0.01 | 0.16±0.03 |
| | | 1.0 | 0.19±0.05 | **0.89**±0.04 | **0.95**±0.02 | 0.82±0.04 | 0.94±0.03 | 0.63±0.10 | 0.22±0.07 | 0.14±0.01 | 0.22±0.08 |
| | 1.0 | 0.0 | **0.76**±0.11 | 0.74±0.11 | 0.98±0.02 | 0.79±0.06 | **1.00**±0.00 | 0.61±0.10 | 0.58±0.09 | 0.13±0.00 | 0.13±0.00 |
| | | 0.1 | 0.53±0.13 | 0.69±0.11 | 0.98±0.02 | 0.79±0.05 | **1.00**±0.00 | 0.57±0.12 | 0.36±0.09 | 0.13±0.00 | 0.13±0.00 |
| | | 0.5 | 0.15±0.02 | **0.75**±0.11 | 0.94±0.02 | 0.83±0.05 | **0.99**±0.00 | 0.60±0.11 | 0.16±0.02 | 0.13±0.00 | 0.15±0.02 |
| | | 1.0 | 0.19±0.05 | **0.75**±0.12 | 0.94±0.02 | 0.85±0.05 | 0.93±0.04 | 0.57±0.12 | 0.18±0.05 | 0.13±0.00 | 0.19±0.08 |
| $O \cap (\sigma_m^\beta < 0)$ | 0.2 | 0.0 | 0.83±0.08 | **0.94**±0.03 | **0.98**±0.02 | 0.74±0.05 | 0.71±0.12 | 0.70±0.08 | 0.76±0.09 | 0.75±0.18 | 0.45±0.06 |
| | | 0.1 | 0.82±0.09 | **0.94**±0.04 | **0.98**±0.02 | 0.73±0.05 | 0.71±0.12 | 0.69±0.07 | 0.67±0.13 | 0.70±0.15 | 0.42±0.07 |
| | | 0.5 | 0.81±0.09 | **0.93**±0.04 | **0.98**±0.02 | 0.74±0.04 | 0.53±0.15 | 0.69±0.07 | 0.72±0.10 | 0.70±0.15 | 0.50±0.12 |
| | | 1.0 | 0.81±0.09 | **0.93**±0.04 | **0.97**±0.02 | 0.74±0.04 | 0.50±0.16 | 0.68±0.08 | 0.76±0.09 | 0.79±0.17 | 0.47±0.10 |
| | 0.5 | 0.0 | 0.71±0.13 | **0.86**±0.07 | **0.97**±0.03 | 0.78±0.05 | 0.88±0.08 | 0.63±0.10 | 0.60±0.05 | 0.52±0.17 | 0.28±0.10 |
| | | 0.1 | 0.71±0.13 | **0.86**±0.07 | **0.97**±0.03 | 0.78±0.05 | 0.87±0.08 | 0.63±0.09 | 0.58±0.09 | 0.53±0.12 | 0.26±0.08 |
| | | 0.5 | 0.67±0.13 | **0.84**±0.07 | **0.96**±0.04 | 0.80±0.05 | 0.75±0.12 | 0.61±0.10 | 0.63±0.08 | 0.54±0.22 | 0.30±0.13 |
| | | 1.0 | 0.67±0.16 | **0.84**±0.08 | **0.96**±0.04 | 0.81±0.06 | 0.68±0.16 | 0.61±0.10 | 0.60±0.08 | 0.52±0.15 | 0.28±0.11 |
| | 1.0 | 0.0 | 0.61±0.15 | 0.77±0.11 | **0.94**±0.06 | 0.82±0.05 | 0.83±0.08 | 0.62±0.10 | 0.51±0.09 | 0.39±0.12 | 0.23±0.08 |
| | | 0.1 | 0.60±0.16 | 0.77±0.10 | **0.94**±0.04 | 0.82±0.05 | 0.82±0.08 | 0.60±0.10 | 0.54±0.09 | 0.38±0.12 | 0.24±0.10 |
| | | 0.5 | 0.60±0.15 | 0.77±0.10 | **0.94**±0.05 | 0.84±0.05 | 0.73±0.13 | 0.60±0.09 | 0.50±0.14 | 0.45±0.16 | 0.25±0.11 |
| | | 1.0 | 0.58±0.16 | 0.77±0.10 | **0.94**±0.05 | 0.84±0.05 | 0.71±0.12 | 0.60±0.10 | 0.43±0.16 | 0.38±0.17 | 0.25±0.08 |
| $\bar{O} \cap (\sigma_m^\beta < 0)$ | 0.2 | 0.0 | 0.28±0.13 | **0.93**±0.04 | 0.88±0.04 | 0.93±0.05 | **0.97**±0.04 | 0.31±0.11 | 0.35±0.16 | 0.34±0.18 | 0.20±0.09 |
| | | 0.1 | 0.33±0.14 | **0.93**±0.04 | 0.89±0.04 | 0.92±0.05 | **0.97**±0.04 | 0.34±0.12 | 0.36±0.15 | 0.41±0.16 | 0.25±0.12 |
| | | 0.5 | 0.64±0.13 | 0.93±0.04 | **0.96**±0.02 | 0.80±0.05 | 0.95±0.04 | 0.57±0.12 | 0.67±0.13 | 0.60±0.13 | 0.38±0.04 |
| | | 1.0 | 0.76±0.11 | 0.94±0.04 | **0.97**±0.02 | 0.77±0.05 | 0.94±0.04 | 0.66±0.09 | 0.67±0.06 | 0.72±0.15 | 0.38±0.13 |
| | 0.5 | 0.0 | 0.25±0.13 | 0.85±0.07 | 0.87±0.05 | 0.95±0.04 | **0.97**±0.03 | 0.29±0.11 | 0.28±0.15 | 0.28±0.11 | 0.17±0.05 |
| | | 0.1 | 0.27±0.14 | 0.85±0.08 | 0.88±0.05 | 0.93±0.04 | **0.97**±0.03 | 0.31±0.11 | 0.33±0.16 | 0.30±0.13 | 0.16±0.04 |
| | | 0.5 | 0.53±0.16 | 0.84±0.09 | **0.94**±0.04 | 0.84±0.05 | **0.95**±0.04 | 0.52±0.11 | 0.55±0.10 | 0.42±0.07 | 0.29±0.10 |
| | | 1.0 | 0.65±0.16 | 0.84±0.08 | **0.96**±0.03 | 0.82±0.06 | 0.92±0.05 | 0.58±0.10 | 0.57±0.10 | 0.54±0.19 | 0.28±0.08 |
| | 1.0 | 0.0 | 0.23±0.13 | 0.77±0.10 | 0.84±0.05 | 0.96±0.03 | **0.97**±0.04 | 0.28±0.10 | 0.24±0.08 | 0.19±0.07 | 0.15±0.04 |
| | | 0.1 | 0.26±0.15 | 0.77±0.10 | 0.86±0.05 | 0.95±0.04 | **0.97**±0.04 | 0.31±0.10 | 0.28±0.13 | 0.20±0.09 | 0.15±0.04 |
| | | 0.5 | 0.46±0.17 | 0.78±0.09 | 0.92±0.05 | 0.87±0.05 | **0.94**±0.05 | 0.51±0.12 | 0.42±0.11 | 0.29±0.09 | 0.18±0.05 |
| | | 1.0 | 0.55±0.16 | 0.77±0.10 | **0.93**±0.05 | 0.85±0.05 | 0.91±0.05 | 0.58±0.11 | 0.47±0.11 | 0.35±0.10 | 0.19±0.07 |

### E.3. Hyper-Parameters

Table 11 shows the list of the hyper-parameters that we set for each causal discovery algorithm. Wherever possible, we adopted the hyper-parameters specified in the original papers, adjusting them only in cases of extremely slow computation or lack of algorithmic convergence.

*Table 11.* Hyper-parameters of the causal discovery algorithms.

| Algorithm | Hyper-Parameter | Value |
|---|---|---|
| **Granger** | `maxlag`
$p$-value | 10
0.05 |
| **PLaCy** | window length ($l$)
stride ($s$)
$p$-value | 50
1
0.05 |
| **PCMCI** | $\tau_{max}$
Conditional Independence Test
`PC`$_\alpha$ | 10
Partial Correlation
0.05 |
| **CCM-Filtering** | Filter Size
Stride
$\tau$
$L$ | 5
1
10
`range(25, L, 250)` |
| **RCV-VarLiNGAM** | $k$
Sequence Length
$\tau_c$
$\tau_v$ | 7
300
0.7
0.4 |
| **Rhino** | Noise Distribution
`init_rho`
`init_alpha` | Gaussian
30
0.2 |
| **DYNOTEARS** | `p`
`max_iter` | 10
100 |
| **PCMCI$_\Omega$** | $\tau_{max}$ | 10 |
| **BCGeweke** | $n_{freqs}$
`Frequency band` | 128
`Three equal band in the spectrum "low", "mid", "high"` |
| **DTF** | $n_{freqs}$
$n_{perm}$ | 128
100 |
| **GewekeNP** | $n_{freqs}$
$n_{perm}$ | 128
100 |

### E.4. PCMCI in Frequency

We observed that performing causal discovery on the $(\mathbf{a}, \boldsymbol{\lambda})$ parameters leads to significant performance improvements for other CD algorithms. For instance, regarding the **PCMCI** algorithm, Table 12 shows that instead of applying the algorithm directly to the time-domain data, running it on the spectral parameters results in higher F1 scores with a small reduction in TNR in some cases. These results suggest that our preprocessing approach can be beneficial also when applied to other causal discovery paradigms, paving the way for future works in this direction.

*Table 12.* Performance Improvement - $N = 5$, $\sigma_g^a = 1.0$

| | | Granger | | PCMCI | |
|---|---|---|---|---|---|
| | | **F1** | **TNR** | **F1** | **TNR** |
| | **C** | | | | |
| $\mathrm{OU}(\sigma_g^m = 0)$ | 0.2 | -0.44±0.32 | +0.21±0.19 | +0.02±0.32 | -0.03±0.07 |
| | 0.5 | +0.16±0.26 | +0.21±0.20 | +0.38±0.44 | -0.03±0.07 |
| | 1.0 | +0.23±0.26 | +0.21±0.21 | +0.46±0.34 | -0.03±0.08 |
| $\widehat{\mathrm{OU}}(\sigma_g^m = 0)$ | 0.2 | -0.22±0.35 | +0.38±0.30 | +0.13±0.39 | -0.05±0.10 |
| | 0.5 | +0.24±0.27 | +0.38±0.29 | +0.37±0.38 | -0.06±0.09 |
| | 1.0 | +0.25±0.26 | +0.34±0.29 | +0.42±0.34 | -0.05±0.09 |
| $\mathrm{OU}(\sigma_g^m > 0)$ | 0.2 | +0.09±0.28 | +0.09±0.10 | +0.38±0.37 | -0.01±0.05 |
| | 0.5 | +0.18±0.20 | +0.11±0.12 | +0.43±0.35 | -0.01±0.05 |
| | 1.0 | +0.20±0.20 | +0.13±0.14 | +0.27±0.31 | -0.00±0.05 |
| $\widehat{\mathrm{OU}}(\sigma_g^m > 0)$ | 0.2 | +0.36±0.25 | +0.41±0.24 | +0.35±0.35 | +0.08±0.10 |
| | 0.5 | +0.41±0.21 | +0.41±0.23 | +0.39±0.27 | +0.08±0.10 |
| | 1.0 | +0.39±0.22 | +0.40±0.24 | +0.31±0.25 | +0.08±0.11 |

### E.5. Non linear system analysis

Some additional experiments have been conducted in cases where the underlying process wasn't linear in order to test the stability of this algorithm in more challenging scenarios. The equation that was used to generate the process in these experiments is Equation (9)

$$x(t + \Delta t) = x(t) + \frac{\Delta t}{\tau_c} \big( \mu - x(t) \big)^2 + \big( \sigma_b \epsilon_b(t) + \sigma_g^a \epsilon_g^a(t) + \sigma_g^m \epsilon_g^m(t) \cdot x(t) \big) \sqrt{\Delta t}, \tag{9}$$

Results of this experiments are reported in figure Figure 5

*Figure 5.* Non linear process. $N = 5$, $C = 1.0$, $s_g = 1.0$, $s_b = 1.0$.

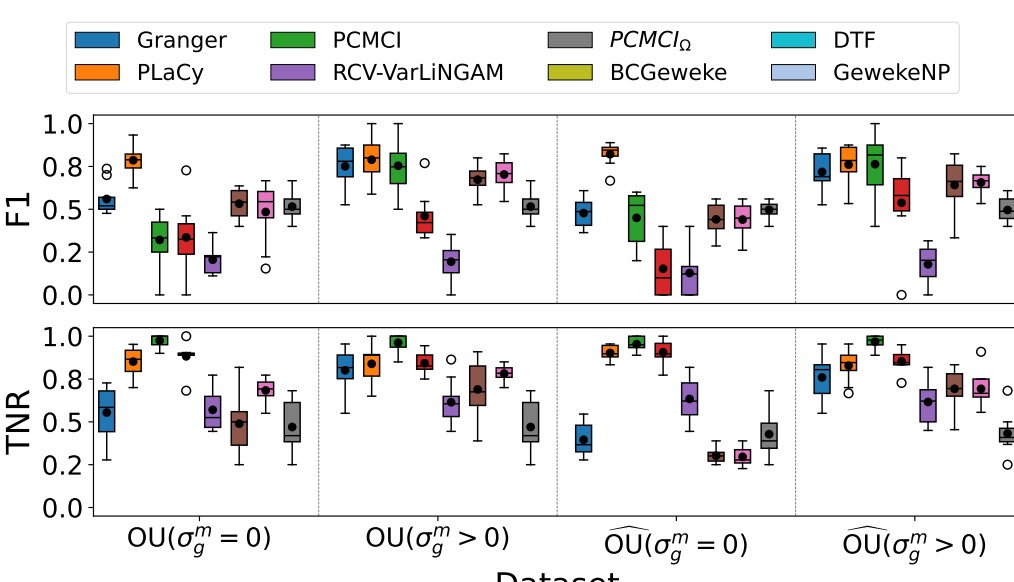

### E.6. Negative Control Experiments

We performed additional numerical experiments and evaluated the Petersen test (Petersen, 2025) on the datasets of Table 1 and Table 2, as well as on the datasets of Table 1 with $N$ increased from 5 to 10. Across all settings, the negative-control baseline yields consistently lower F1 and TNR scores, confirming the effectiveness of our approach. We also remark that the $p$-values returned by the tests were always statistically significant. Results can be found in Table 13

Additionally, we note that the Petersen test can vary simply due to the number of edges reconstructed by each algorithm. To account for this effect, we performed an alternative evaluation in which the Petersen test is computed using the true number of edges in the underlying causal graph, rather than the number predicted by each method. This alternative evaluation does not depend on the specific algorithm under consideration and therefore yields a single value for each graph size. The corresponding results are reported below in Table 14 and again demonstrate that our approach is substantially more robust than a random guess.

### E.7. Acyclicity Assumption

Finally, while the synthetic graphs used for our controlled experiments were generated as DAGs, we do not necessarily assume acyclicity of the underlying graphs, nor do we impose this constraint on the recovered graphs. This design choice is intentional, as it allows the method to remain as general and broadly applicable as possible. However, we performed additional experiments to evaluate the behavior of PLaCy when acyclicity is explicitly enforced.

Specifically, we iteratively detect cycles in the recovered graph and remove, at each step, the single edge with the highest

*Table 13.* Comparison of PLaCy with the Petersen test across synthetic and real-world datasets. Results are reported as mean $\pm$ standard deviation over multiple runs with $\sigma_g = 0.5$: $C = 0.5$ $\sigma_b = 0.5$.

| | Dataset / Condition | F1 (PLaCy) | TNR (PLaCy) | F1 (Petersen test) | TNR (Petersen test) |
|---|---|---|---|---|---|
| $N = 5$ | $OU(\sigma_g^m = 0)$ | $0.58 \pm 0.27$ | $0.93 \pm 0.06$ | $0.11 \pm 0.17$ | $0.83 \pm 0.11$ |
| | $\widehat{OU}(\sigma_g^m = 0)$ | $0.57 \pm 0.25$ | $0.89 \pm 0.08$ | $0.15 \pm 0.19$ | $0.79 \pm 0.10$ |
| | $OU(\sigma_g^m = 0.5)$ | $0.77 \pm 0.20$ | $0.92 \pm 0.08$ | $0.18 \pm 0.22$ | $0.79 \pm 0.11$ |
| | $\widehat{OU}(\sigma_g^m = 0.5)$ | $0.78 \pm 0.20$ | $0.92 \pm 0.08$ | $0.16 \pm 0.19$ | $0.79 \pm 0.11$ |
| $N = 10$ | $OU(\sigma_g^m = 0.0)$ | $0.63 \pm 0.10$ | $0.92 \pm 0.04$ | $0.15 \pm 0.10$ | $0.82 \pm 0.06$ |
| | $\widehat{OU}(\sigma_g^m = 0.0)$ | $0.57 \pm 0.09$ | $0.86 \pm 0.06$ | $0.17 \pm 0.09$ | $0.76 \pm 0.07$ |
| | $OU(\sigma_g^m = 0.5)$ | $0.69 \pm 0.08$ | $0.86 \pm 0.07$ | $0.19 \pm 0.09$ | $0.74 \pm 0.08$ |
| | $\widehat{OU}(\sigma_g^m = 0.5)$ | $0.70 \pm 0.09$ | $0.86 \pm 0.08$ | $0.19 \pm 0.09$ | $0.73 \pm 0.09$ |
| | Rivers | $0.51 \pm 0.10$ | $0.68 \pm 0.16$ | $0.26 \pm 0.08$ | $0.61 \pm 0.15$ |
| | AirQuality | $0.45 \pm 0.04$ | $0.65 \pm 0.07$ | $0.32 \pm 0.03$ | $0.59 \pm 0.08$ |

*Table 14.* Modified Petersen test using the true number of edges.

| Dataset | Mean F1 | Mean TNR |
|---|---|---|
| $N = 5$ | $0.15 \pm 0.20$ | $0.85 \pm 0.08$ |
| $N = 10$ | $0.16 \pm 0.10$ | $0.84 \pm 0.04$ |
| Rivers | $0.16 \pm 0.15$ | $0.83 \pm 0.03$ |
| AirQuality | $0.24 \pm 0.02$ | $0.74 \pm 0.01$ |

$p$-value (i.e., the lowest statistical significance), until no cycles remain. The results of this procedure are reported in Table 15.

*Table 15.* Comparison between acyclic PLaCy and standard PLaCy for $N = 5$; $\sigma_g = 0.5$; $C = 0.5$; $\sigma_b = 0.5$ .

| Dataset / Condition | Acyclic PLaCy (F1) | Acyclic PLaCy (TNR) | PLaCy (F1) | PLaCy (TNR) |
|---|---|---|---|---|
| $OU(\sigma_g^m = 0)$ | $0.65 \pm 0.19$ | $0.96 \pm 0.06$ | $0.58 \pm 0.27$ | $0.93 \pm 0.06$ |
| $\widehat{OU}(\sigma_g^m = 0)$ | $0.60 \pm 0.20$ | $0.90 \pm 0.06$ | $0.57 \pm 0.25$ | $0.89 \pm 0.08$ |
| $OU(\sigma_g^m = 0.5)$ | $0.90 \pm 0.10$ | $0.96 \pm 0.04$ | $0.77 \pm 0.20$ | $0.92 \pm 0.08$ |
| $\widehat{OU}(\sigma_g^m = 0.5)$ | $0.91 \pm 0.09$ | $0.94 \pm 0.06$ | $0.78 \pm 0.20$ | $0.92 \pm 0.08$ |

These results show that imposing acyclicity does not deteriorate the reconstructive performance of our method and, in several cases, even leads to a slight improvement

### E.8. Benjamini–Hochberg correction

Causal discovery procedures involve multiple hypothesis tests, which can increase the risk of false positive edge detections, especially in high-dimensional settings. In the main experiments, we do not apply standard FDR correction (Benjamini & Hochberg, 1995), to ensure comparability with the other methods and because the analysis focuses on individual link performance.

As a robustness check, we repeated the synthetic benchmark of Figure 2 using Benjamini–Hochberg false discovery rate correction on the $p$-values produced by PLaCy, standard Granger causality, and PCMCI. Tables 16 and 17 report the resulting F1 and TNR scores under the same setting used in the main text ($C = 0.5$, $s_b = 0.5$, $n_{\text{vars}} = 5$, $s_g = 1.0$).

These results further show that the advantage of the proposed algorithm remains stable and statistically relevant under multiple-testing control. Therefore, when causal discovery is evaluated at the graph level rather than on individual edges, FDR correction is recommended to control for spurious detections arising from multiple simultaneous tests.

*Table 16.* F1 score with Benjamini–Hochberg false discovery rate correction.

| Dataset | Granger | PLaCy | PCMCI |
|---------|---------|-------|-------|
| $OU(\sigma_g^m = 0)$ | $0.576 \pm 0.083$ | $\mathbf{0.754 \pm 0.110}$ | $0.291 \pm 0.185$ |
| $\overline{OU}(\sigma_g^m = 0)$ | $0.560 \pm 0.088$ | $\mathbf{0.776 \pm 0.150}$ | $0.400 \pm 0.313$ |
| $OU(\sigma_g^m > 0)$ | $0.710 \pm 0.143$ | $\mathbf{0.855 \pm 0.120}$ | $0.459 \pm 0.337$ |
| $\overline{OU}(\sigma_g^m > 0)$ | $0.581 \pm 0.122$ | $\mathbf{0.853 \pm 0.061}$ | $0.674 \pm 0.259$ |

*Table 17.* TNR score with Benjamini–Hochberg false discovery rate correction.

| Dataset | Granger | PLaCy | PCMCI |
|---------|---------|-------|-------|
| $OU(\sigma_g^m = 0)$ | $0.569 \pm 0.116$ | $\mathbf{0.984 \pm 0.035}$ | $0.979 \pm 0.034$ |
| $\overline{OU}(\sigma_g^m = 0)$ | $0.558 \pm 0.046$ | $0.931 \pm 0.064$ | $\mathbf{0.974 \pm 0.026}$ |
| $OU(\sigma_g^m > 0)$ | $0.739 \pm 0.159$ | $0.947 \pm 0.054$ | $\mathbf{0.990 \pm 0.029}$ |
| $\overline{OU}(\sigma_g^m > 0)$ | $0.568 \pm 0.178$ | $0.934 \pm 0.043$ | $\mathbf{0.973 \pm 0.051}$ |

## E.9. Complexity analysis

Let $N$ be the number of time series (nodes), $L$ the length of each time series, $l$ the window length used for the spectral analysis, and $s$ the stride of the sliding window.

**Spectral preprocessing.** Each time series is divided into overlapping windows of length $l$ and stride $s$. The number of windows generated from a single time series is therefore

$$W = \left\lfloor \frac{L - l}{s} \right\rfloor + 1 = O\left(\frac{L}{s}\right).$$

In each window, the algorithm performs: (i) an FFT of size $l$ with cost $O(l \log l)$, (ii) a log–log transformation of amplitudes and frequencies ($O(l)$), and (iii) a linear regression to estimate the spectral coefficients $(a, \lambda)$ ($O(l)$). The FFT dominates the per-window computation, so the cost per window is

$$O(l \log l).$$

Hence, the cost for a single time series is

$$O\left(\frac{L}{s}\right) \cdot O(l \log l) = O\left(\frac{L}{s} l \log l\right),$$

and for all $N$ time series:

$$O\left(N \cdot \frac{L}{s} l \log l\right).$$

Since the stride $s$ is typically very small (often $s = 1$), we may simplify this as

$$O(NL \, l \log l).$$

After the spectral preprocessing, the method performs pairwise Granger causality tests across all ordered pairs of the $N$ variables, resulting in $N(N - 1) = O(N^2)$ tests. A single Granger test on two time series of length $L$ (with fixed lag order) requires solving a small linear regression and therefore costs $O(L)$. The overall cost of this step is thus

$$O(N^2 L).$$

Combining both contributions, the total computational complexity of the method is

$$O(NL \, l \log l) \; + \; O(N^2 L).$$

**E.10. Execution Times**

The experiments were conducted on a high-performance computing cluster comprising 50 DELL EMC PowerEdge R7425 servers, each equipped with dual AMD EPYC 7301 processors (32 cores total per node at 2.2GHz). Among the nodes, 19 are enhanced with NVIDIA Quadro RTX 6000 GPUs (24GB VRAM). These GPUs were used only for running **Rhino**, as this method requires the training of deep neural networks. As such, for $N = 5$, the execution of **Rhino** took $\sim 4$ hours.

*Table 18.* Elapsed time for a single experiment (seconds).

| Method | N = 5 | N = 10 |
|---|---|---|
| Granger | 0.16±0.01 | 0.71±0.00 |
| PLaCy | 5.80±0.01 | 11.94±0.03 |
| PCMCI | 1.05±0.00 | 4.86±0.03 |
| CCM-Filtering | 426.35±1.57 | 1924.20±19.32 |
| RCV-VarLiNGAM | 1.98±0.10 | 7.03±0.27 |
| DYNOTEARS | 0.00±0.00 | 0.00±0.00 |
| PCMCI$\omega$ | 7.06±0.04 | 44.42±0.13 |
| BCGeweke | 2.45±0.10 | 4.63±0.26 |
| DTF | 14.19±1.33 | 93.10±2.01 |
| GewekeNP | 3.34±0.03 | 9.69±0.30 |

**E.11. Time-dependent mean reversion and endogenous spectral changes**

To assess whether non-stationary mean-reversion dynamics can alter the spectral properties of the process and potentially affect causal inference, we performed additional experiments in which the OU mean-reversion parameter was made explicitly time-dependent. We considered two forms of temporal modulation: a sinusoidal variation and an exponentially decaying mean-reversion coefficient. Since these settings fall outside the strict theoretical assumptions of PLaCy, they provide robustness tests under endogenous spectral changes.

In these experiments, we used the post-causality setting, where the OU dynamics are first simulated without causal interactions and the causal effects are then added post hoc as lagged perturbations. For each variable $i = 1, \ldots, N$, the non-causal time-dependent OU process is first generated as

$$x_i(t + \Delta t) = x_i(t) - \Delta t \, a(t) \frac{\text{sign}(x_i(t))|x_i(t)|^\gamma}{\tau_c} + \sqrt{\Delta t} \, \eta_i(t),$$

where $\gamma$ is the decay exponent, fixed to $\gamma = 1$ in our experiments, $\tau_c$ is the characteristic relaxation time, and $\eta_i(t)$ is the noise term.

For the sinusoidal modulation experiment, we used

$$a(t) = \sin(t),$$

whereas for the exponentially decaying modulation experiment we used

$$a(t) = \exp(-\lambda t),$$

with decay rate $\lambda = 0.1$. After simulating the non-causal trajectories, causal effects are added post hoc. For each directed edge $i \to j$, with $A_{ij} = 1$, the target variable is updated as

$$x_j(t) \leftarrow x_j(t) + c \, A_{ij} \, \text{sign}(x_i(t - \ell))|x_i(t - \ell)|^\beta, \qquad t \geq \ell,$$

where $c$ is the causal strength, $\ell$ is the causal lag, and $\beta$ is the causal exponent.

Overall, PLaCy remains stable under explicitly time-varying mean-reversion dynamics. In particular, it consistently improves TNR across all analyzed scenarios, while achieving comparable or superior F1 scores in the presence of modulated mean

*Table 19.* Results for sinusoidal time-dependent mean-reversion experiments with $N = 5$ and $\sigma_g = 1.0$. Best values between Granger and PLaCy are shown in bold.

| Dataset | F1 Granger | F1 PLaCy | TNR Granger | TNR PLaCy |
|---|---|---|---|---|
| $\mathrm{OU}(\sigma_g^m = 0)$ | $\mathbf{0.54 \pm 0.09}$ | $0.49 \pm 0.11$ | $0.47 \pm 0.18$ | $\mathbf{0.53 \pm 0.22}$ |
| $\overline{\mathrm{OU}}(\sigma_g^m = 0)$ | $\mathbf{0.51 \pm 0.10}$ | $0.50 \pm 0.11$ | $0.44 \pm 0.15$ | $\mathbf{0.54 \pm 0.19}$ |
| $\mathrm{OU}(\sigma_g^m > 0)$ | $0.67 \pm 0.12$ | $\mathbf{0.79 \pm 0.12}$ | $0.73 \pm 0.15$ | $\mathbf{0.85 \pm 0.09}$ |
| $\overline{\mathrm{OU}}(\sigma_g^m > 0)$ | $0.68 \pm 0.13$ | $\mathbf{0.72 \pm 0.15}$ | $0.70 \pm 0.19$ | $\mathbf{0.81 \pm 0.13}$ |

*Table 20.* Results for exponentially decaying time-dependent mean-reversion experiments with $N = 5$ and $\sigma_g = 1.0$. Best values between Granger and PLaCy are shown in bold.

| Dataset | F1 Granger | F1 PLaCy | TNR Granger | TNR PLaCy |
|---|---|---|---|---|
| $\mathrm{OU}(\sigma_g^m = 0)$ | $\mathbf{0.56 \pm 0.09}$ | $0.54 \pm 0.11$ | $0.55 \pm 0.12$ | $\mathbf{0.69 \pm 0.13}$ |
| $\overline{\mathrm{OU}}(\sigma_g^m = 0)$ | $\mathbf{0.57 \pm 0.13}$ | $0.54 \pm 0.11$ | $0.55 \pm 0.17$ | $\mathbf{0.66 \pm 0.18}$ |
| $\mathrm{OU}(\sigma_g^m > 0)$ | $0.61 \pm 0.15$ | $\mathbf{0.77 \pm 0.11}$ | $0.68 \pm 0.16$ | $\mathbf{0.86 \pm 0.09}$ |
| $\overline{\mathrm{OU}}(\sigma_g^m > 0)$ | $0.61 \pm 0.16$ | $\mathbf{0.81 \pm 0.14}$ | $0.61 \pm 0.21$ | $\mathbf{0.86 \pm 0.11}$ |

reversion. These results suggest that PLaCy does not systematically misinterpret endogenous temporal variation in the mean-reversion coefficient as causal structure. Instead, the method appears robust to local spectral changes induced by non-stationary dynamics, supporting its applicability beyond idealized stationary regimes.

