# OpenReview forum: "Robust Causal Discovery in Real-World Time Series with Power-Laws"
_ICML.cc/2026/Conference — ICML 2026 spotlight_

### Official Review · Reviewer_L9j6 · 2026-03-05

**Soundness:** 4
**Presentation:** 4
**Significance:** 4
**Originality:** 4
**Overall Recommendation:** 5
**Confidence:** 4

**Summary:**

This paper proposes a novel causal discovery framework PLaCy , which recovers causal relationships between variables from random time series. PLaCy observes that many real-world time series exhibit power-law spectral structure characteristics. It proposes extracting spectral parameters (amplitude $a$ and spectral exponent  $\lambda$) in the frequency domain and utilizing these spectral feature sequences for causal analysis. A sliding window approach is used to perform spectral analysis on the time series, fitting a power-law spectral model within each window to obtain a time-varying sequence of spectral parameters. Then, the Granger causality test is applied to these feature sequences to reconstruct the causal graph. Theoretically, the property of preserving the original causal graph structure under spectral feature transformation is analyzed, and the method's performance is verified through experiments using synthetic and real data. Experiments show that this method achieves good results in the presence of non-stationary noise, multiplicative noise, and complex dynamic structures.

**Compliance With Llm Reviewing Policy:**

Affirmed.

**Final Justification:**

After reading the rebuttal, my concerns have been addressed and I maintain my score.

**Key Questions For Authors:**

- How does PLACy perform when the time series exhibits non-power-law characteristics within a window, such as in cases of poor power-law fit?

- Stability analysis of the causal graph relative to window length $(l)$ and step size $(s)$ and its impact on experimental results F1 and TNR.

-  In nonlinear experiments, how do nonlinear characteristics affect the analysis of causal relationships?

**Limitations:**

Although this work has thoroughly discussed the issues of slowly changing spectra and short time series, and mentioned the assumptions about power-law characteristics, the current method is mainly based on the Granger causal framework, which may be biased in the presence of potential confounding variables. Experiments on sensitivity analysis of some time windows and step size parameters still need further testing and discussion.

**Strengths And Weaknesses:**

**Strengths：**
- The core idea of ​​shifting causal discovery from the time domain to the evolutionary spectral parameter domain is innovative and highly attractive. Real-world complex systems (e.g., climate, financial, and neuroscience data) often possess scale-free structures, making causal analysis based on spectral structure highly relevant.

- The design is concise and interpretable, avoiding the interpretability issues of black-box models from spectral fitting to causal graph construction.

Theorem 3.1 and Appendix B prove that the causal graph is invariant under the proposed spectral transformation for linear processes, providing a rigorous theoretical foundation.

- PLaCy exhibits excellent robustness in scenarios where traditional methods fail, particularly when dealing with multiplicative noise and non-equilibrium initializations (such as phase transitions). PLaCy achieves a high true negative rate (TNR), avoiding spurious correlations.

- This work validates the framework's excellent performance in handling missing data and heterogeneous dynamic scenarios through experiments on numerous real-world datasets with known real-world structures.

**Weaknesses:**
- While this work shows the prevalence of power-law properties in time series, the performance of this method depends on the effectiveness of the power-law fit within each window.

- Theoretical guarantees primarily address the derivation of linear causal mechanisms. Although experiments show good performance under certain nonlinear settings (Appendix E.5), the effectiveness of spectral parameters in capturing the nonlinear characteristics of time series requires further discussion.

- PLaCy introduces window length and step size analysis for causality and proposes an adaptive selection process based on p-values; however, the impact of these hyperparameters on the stability of the causal graph is not adequately analyzed.

---

> ### Author Rebuttal · Authors · 2026-03-31
>
> We sincerely thank the Reviewer for their thorough assessment, and we are pleased that they deeply understood and appreciated the core contributions of our work.
> Below, we address each of the raised questions and suggestions in turn.
>
> **P1. Performance for non power-law time series**
> > How does PLACy perform when the time series exhibits non-power-law characteristics within a window, such as in cases of poor power-law fit?
>
> In general, we do expect the performance of PLaCy to be positively correlated with the quality of the power-law fit (this is why we also expect the method to work well for real-world, scale-free, systems).
> Obviously, as the Reviewer pointed out, not all windows of a scale-free time series will strictly exhibit power-law behaviour (as we also observe in our experiments), which may introduce some noise.
> To mitigate issues arising from poor window fits, we employ a $p$-value–based criterion for selecting the best window size, such that each window captures as well as possible a power-law distribution.
> We observe robustness for deviations from power-law distributions, such as in the case of the Rivers dataset. However, we explicitly suggest performing a preliminary spectral inspection in the Limitations section and in case of strongly non-power-law time series other methods might be more suitable.
>
> **P2. Stability of the causal graph**
> > Stability analysis of the causal graph relative to window length (l) and step size (s) and its impact on experimental results F1 and TNR.
>
> We agree that a stability analysis with respect to window length and step size is quite relevant. We have analysed some of these aspects in the Appendix, and the results are presented in Figure 3 of Appendix D.1. We are willing to perform further ablation studies if the Reviewer might consider them useful.
>
> **P3. Nonlinear characteristics**
> > In nonlinear experiments, how do nonlinear characteristics affect the analysis of causal relationships?
>
> We have performed some experiments checking these aspects. These are reported in Appendix E.5.  We find that the algorithm appears to be extraordinarily robust to nonlinearity, thus confirming that PLaCy performs well also behyond the theoretical guarantees. Furthermore, we are carrying out additional experiments with time-varying mean-reversion terms. Preliminary results are reported in our reply to the second Reviewer.
>
> ---
>
> We thank the reviewer again for their feedback and questions. We hope our responses answer their questions, and we would be happy to provide any further clarification.

---

> > ### Author Rebuttal · Reviewer_L9j6 · 2026-04-02
> >
> > Thank you for your explanation. I do not have further questions

---

> > > ### Author Response · Authors · 2026-04-07
> > >
> > > We thank the Reviewer for their very positive evaluation of our work. We are pleased that our rebuttal resolved all the raised points.

---

### Official Review · Reviewer_dtUt · 2026-03-08

**Soundness:** 2
**Presentation:** 2
**Significance:** 2
**Originality:** 2
**Overall Recommendation:** 5
**Confidence:** 4

**Summary:**

The paper introduces PLaCy, a framework designed for causal discovery in complex, real-world time series that follow power-law frequency distributions. Traditional algorithms often struggle with non-stationarity and noise, leading to the detection of false causal links. To address this, PLaCy transforms raw data into spectral features to isolate causal signals from external interference. The authors provide a theoretical proof that this spectral transformation preserves the original causal structure while enhancing robustness. Extensive experiments on synthetic and real-world datasets, such as river levels and air quality, demonstrate that PLaCy outperforms existing methods. By focusing on the evolution of spectral properties, the algorithm offers a new approach to identifying cause-and-effect relationships in stochastic systems.

**Compliance With Llm Reviewing Policy:**

Affirmed.

**Final Justification:**

The authors addressed my concerns and conducted follow-up experiments that further established the utility of placy. I am happy to raise my score to Accept.

**Key Questions For Authors:**

The key questions are written mostly based on the weaknesses listed above. The questions are also written below for sake of completeness.

*Choice of the power-law model* - Why is the power-law model chosen to characterize the spectral density? How robust is the method when the spectrum deviates from a power-law form?

*Interpretation of the spectral exponent* - The method relies heavily on the dynamics of the exponent $\lambda$. Could the authors provide more intuition on how causal interactions are encoded in $\lambda$?

*Window selection procedure* - The paper states that the window length is chosen as the smallest one for which the Wald test for the spectral fit achieves $p<0.05$. Could the authors clarify this procedure? In particular, if no causal structure is present, searching for a window satisfying this criterion may produce spurious results. Also, do the authors conduct the tests for every window? If yes, it makes the practical implementation of the method computationally expensive.

*Multiple testing correction* - Since many pairwise Granger tests are conducted, are the resulting p-values adjusted to control the false discovery rate or any other multiple testing criterion?

*Causal discovery model in the feature space* - What is the precise causal model assumed for the transformed feature series $(a,\lambda)$?

*Statement in Theorem 3.1* - In Theorem 3.1, it is assumed that `$x$ has power-law spectra with common frequency dependence across all frequencies $f$'. The meaning of this assumption is not clear.

*Simulation design* - Are the proposed simulation settings sufficiently challenging to stress-test the method? In particular, do they reflect realistic scenarios encountered in applications such as neuroscience or climate science?

*Real-world dataset ground truth* - For the AirQuality dataset, how exactly are sensor distances used to construct the ground-truth causal graph? More details on the construction of the causal matrix would help interpret the results.

*Clarification on proof arguments* - The statement $x_j = \alpha\ x_i$ in the proof of Theorem B.2 is either incomplete or incorrect. Setting $\alpha=0$ makes the series $x_j$ a zero valued time-series. The statement needs clarification.

**Limitations:**

Yes.

**Strengths And Weaknesses:**

**Strengths**:

*Interesting idea combining spectral analysis and causal discovery* - The manuscript proposes PLaCy, a framework that performs causal discovery using time-varying spectral parameters of power-law processes, rather than the raw time series. This manuscript seeks to present a notable context in which causal inference can be improved by exploiting structural properties of real-world time series.

*Motivation rooted in empirical properties of real data* - The paper highlights that many real-world systems exhibit power-law spectra and scale-free behavior, and attempts to leverage this property to improve robustness of causal discovery methods.

*Conceptually simple pipeline* - The steps in the proposed approach, namely, segmenting the time series, estimating the power-law parameters $(a,\lambda)$, and applying Granger causality on these features - is intuitive and relatively straightforward to implement.

*Theoretical discussion of invariance* - The authors attempt to provide theoretical justification by arguing that the causal graph is preserved under the proposed spectral feature mapping. This adds conceptual support to the methodology.

*Empirical comparison with several baselines* - The experimental section includes comparisons with multiple established causal discovery methods, including PCMCI, DYNOTEARS, VarLiNGAM variants, and spectral methods, suggesting the authors attempted a reasonably broad evaluation.

*Evaluation on both synthetic and real datasets* - Experiments are conducted on simulated data as well as real-world datasets (e.g., hydrological and air quality data), which strengthens the empirical narrative.




**Weaknesses** :

*Justification for the power-law model is insufficient* - The method assumes that the spectral amplitude follows a power-law decay $A(f)=e^a f^{-\lambda}$. However, it is not clear why this specific model should be appropriate for all considered time series, nor how robust the method is when this assumption is violated.

*Unclear role of the spectral parameters in causal discovery* - The paper claims that causal signals are encoded in the dynamics of $\lambda$, but does not provide a clear explanation of how causal relationships manifest in the evolution of this parameter.

*Methodological choices are insufficiently justified* - For example, the window length selection procedure relies on selecting the shortest window whose spectral fit passes a Wald test at
$p<0.05$. This procedure may introduce bias and potentially lead to spurious detections, particularly when no true causal structure is present.

*Multiple testing issues are not addressed* - The method applies numerous Granger tests across variable pairs, yet it is unclear whether the resulting p-values are adjusted to control the false discovery rate.

*Experimental setup may not be sufficiently challenging* - The synthetic simulations appear relatively structured (e.g., OU processes with injected causal links), and it is unclear whether these scenarios adequately reflect the complexity of real-world causal systems.

*Incomplete reporting of metrics* - While TNR is discussed extensively in the text, it is not reported in Table 1, which limits the ability to evaluate the trade-off between detecting causal links and avoiding false positives.

*Ground-truth construction for real datasets is unclear* - For the AirQuality dataset, the ground-truth causal graph is derived from sensor distances, but the procedure used to construct the full causal matrix is not clearly described.

*Proofs of the theorems* - the proofs, particularly Theorem B.1 lacks formal mathematical arguments, and mostly based on textual narrative. While showing the preservation of linear dependence under spectral transformations, the authors consider only scale transformation in the proof of Theorem B.2.


Apart from the major issues detailed above, there are a couple of minor issues listed below:

*Several unclear or undefined notations* - Some notations appear without proper definition (e.g., variables in the spectral fitting description), which makes parts of the methodology difficult to follow.

*Some claims lack citations* - Certain statements (e.g., regarding power-law behavior or spectral properties) would benefit from additional references.

---

> ### Author Rebuttal · Authors · 2026-03-31
>
> We thank the Reviewer for their detailed comment. In general, most of the concerns raised stem from minor misunderstandings. We address each point systematically below to clarify these issues and improve the manuscript.
>
> **P1. Justification for the power-law model**
>
> Section 2.3 is dedicated entirely to establishing the empirical and theoretical basis for our power-law assumption.
> The assumption is based on decades of evidence across different scientific fields.
> Furthermore, while our main objective is to propose a method specifically designed for power-law distributed time series (ubiquitously found in real-world systems), we do address the robustness of the method even when this core assumption is violated.
>
> **P2. Role of spectral parameters in causal discovery**
>
> We agree that the role of the spectral parameters can be further clarified.
> Under the power-law assumption, the spectral amplitude is modelled as $A(f) = a f^{-\lambda}$.
> This representation compresses the spectral behaviour of the time series into two parameters: the intercept $a$ and the slope $\lambda$.
>
> Causal interactions affect the autocorrelation properties of the time series and, since the spectral representation is directly related to the autocovariance structure, such changes induce systematic variations in $A(f)$, and therefore in the parameters $(a, \lambda)$.
> The general connection between causality and spectral representations is well established in the literature. In particular, Geweke's formulation of causality shows how causal interactions can be characteried in the frequency domain, linking predictive relationships to spectral properties of the process [Geweke, 1982].
>
> We will clarify this connection more explicitly in the manuscript.
>
> **P3. Window length selection procedure**
>
> A Wald test is used to test the quality of the power-law fit to the spectrum within each window and to select the window length accordingly.
> Our intuition is that selecting a window where the spectral fit is valid cannot really inflate false positives in the subsequent causality test.
> This intuition is corroborated by the experiments reported in Appendix D.1, where we empirically investigate the influence of the window length on the performance of PLaCy.
> We find that the window length is not so impactful as long as it allows to fit reasonably well a power-law behaviour.
>
> **P4. Multiple testing correction**
>
> As correctly pointed out, in the current implementation we use a fixed significance threshold ($p$ < 0.05) for the causality tests and do not apply an explicit multiple testing correction.
> Introducing corrections such as Bonferroni or Benjamini-Hochberg would definitely be possible, and we expect this to further increase the reported TNR. We will highlight this possibility in the revised manuscript. Furthermore, we are willing to expand the existing ablation studies in the Appendix with new tests, if the Reviewer deems this necessary for the camera-ready version.
>
> **P5. Experimental setup**
>
> We agree that synthetic OU simulations represent a structured and controlled setting.
> This is our intention: to provide a controlled environment where the underlying causal structure is known, allowing us to systematically stress-test the method and analyse its behaviour under different conditions.
> Importantly, we do not restrict our evaluation to _simple_ OU dynamics. We also consider very challenging settings, including nonlinear dependencies, non-Gaussian noise, and multiplicative noise.
> In these regimes, we observe that several state-of-the-art methods fail in at least part of the scenarios, while PLaCy maintains strong performance.
> In addition, we evaluate the method on real-world datasets, and still observe consistent performance improvements.
>
>
> **P6. Theorem clarifications**
>
> Th. B.1: The presentation is largely textual since it relies on classical results available in the literature (e.g., [van der Vaart, 1998]). In the revision, we will include the missing explicit citations for clarity and completeness.
>
> Th. B.2: We thank the Reviewer this observation. We will explicitly state that the results work only for $\alpha  \neq 0$. Furthermore, the proof focused on scale transformations only because the constant shift only impacts the spectra at zero-frequencies that are discarded during the analysis. We will also clarify this in the revised manuscript.
>
> Th. 3.1: We will revise the statement "$x$ has power-law spectra with common frequency dependence across all frequencies $f$" to clarify that all components need to share the same spectral exponent within each window.
>
> **P7. Clarifications**
>
> _Some claims lack citations_
>
> We are expanding the references and are happy to add any particular reference the Reviewer finds important.
>
> _Reporting_
>
> The TNR values are currently reported in Appendix E.2. Following the Reviewer's suggestion, we will add them to Table 1.
>
> _AirQuality_
>
> We use the provided ground-truth for consistency with the literature.

---

> > ### Author Rebuttal · Reviewer_dtUt · 2026-04-02
> >
> > I thank the authors for their detailed and thoughtful responses. Most of my earlier questions have been addressed satisfactorily. However, I still have some reservations regarding the statistical soundness of the proposed algorithm, which affect my overall assessment.
> >
> > In particular, the method involves multiple hypothesis testing at two stages:
> >
> > (a) Window size selection (MINOR).
> > The window size is chosen from a set of candidate values using a p-value-based criterion. The authors note that, in practice, the downstream analysis is not highly sensitive to this choice. While this may hold when candidates are spaced linearly, alternative choices (e.g., logarithmic or exponential grids) could potentially lead to different outcomes. Moreover, as the size of the candidate set increases, the role of multiple testing adjustments becomes more important. Although the lack of p-value correction may have limited practical impact in some cases, incorporating such adjustments would make the procedure more statistically principled and potentially more robust, particularly when underlying assumptions (e.g., power-law behavior) are not fully satisfied.
> >
> > (b) Granger causality testing (MAJOR).
> > A more significant concern arises in the Granger causality step, where multiple hypotheses are again tested without any correction for multiplicity. In the absence of false discovery rate (FDR) control, there is a risk of identifying spurious edges, especially in high-dimensional settings with many nodes. This raises concerns about the fairness of comparisons with baseline methods that may incorporate such controls.
> >
> > Overall, while the method shows promising empirical performance in both simulations and real data applications, I believe that addressing these statistical considerations would substantially strengthen the rigor and reliability of the approach. In its current form, I find it difficult to fully assess the methodological soundness of the proposed framework, and therefore I will maintain my current evaluation.

---

> > > ### Author Response · Authors · 2026-04-03
> > >
> > > We thank the Reviewer for carefully reading our rebuttal and for their feedback.
> > > We are sorry that, given the space constraints, we could not exhaustively comment on all the raised points. We address the Reviewer's two remaining concerns below.
> > >
> > > ---
> > >
> > > **P1. (Minor) Window size selection**
> > >
> > > The objective of our window selection procedure is not to identify all statistically significant window sizes, but rather to select a single reasonable window size using a non-parametric criterion. Our strong empirical results demonstrate the suitability of our approach.
> > >
> > > Furthermore, we would like to highlight that an example of non-linear windowing strategy is shown in Figure 3 of Appendix D1, where we consider window lengths of 10, 20, 30, 50, 100, 200, and 500, which approximate a logarithmic/exponential spacing rather than a uniformly spaced linear grid.
> > >
> > > The figure shows that even strong variations in the window size do not lead to any significant drop in the algorithm's performance.
> > >
> > > In any case, we acknowledge the Reviewer's minor concern, and we will mention the possibility of using a multiple testing adjustment for the window selection procedure in the final version of the paper.
> > >
> > >
> > > **P2. Granger causality testing**
> > >
> > >
> > > We sincerely thank the reviewer for raising the important issue of multiple testing control in causal discovery. We agree that this is a relevant methodological aspect, and we appreciate the opportunity to clarify and extend our evaluation.
> > >
> > > In our original experiments, we adopted an evaluation protocol consistent with prior work in the literature, where methods are assessed without enforcing a global multiple-testing correction, in order to focus on individual link detection performance and maintain comparability across approaches. For example, in the original PCMCI study, the authors note that false discovery rate control was not applied for this reason.
> > >
> > > That said, we fully acknowledge the reviewer’s point that incorporating multiple-testing correction provides an additional and valuable perspective on the robustness of the results.
> > >
> > > Following this suggestion, we have conducted additional experiments using Benjamini–Hochberg false discovery rate (FDR) correction, applied consistently across PLACy, Granger, and PCMCI. The results, corresponding to the same experimental setting as Figure 2, are reported below.
> > >
> > >
> > > ---
> > >
> > > _F1 Score (C = 0.5, sb = 0.5, n_vars = 5, s_g = 1.0)_
> > >
> > > | Dataset | Granger | PLACy | PCMCI |
> > > |--------|--------|--------|--------|
> > > | $OU(\sigma_g^m = 0)$ | 0.576 ± 0.083 | **0.754 ± 0.110** | 0.291 ± 0.185 |
> > > | $\widehat{OU}(\sigma_g^m = 0)$ | 0.560 ± 0.088 | **0.776 ± 0.150** | 0.400 ± 0.313 |
> > > | $OU(\sigma_g^m > 0)$ | 0.710 ± 0.143 | **0.855 ± 0.120** | 0.459 ± 0.337 |
> > > | $\widehat{OU}(\sigma_g^m > 0)$ | 0.581 ± 0.122 | **0.853 ± 0.061** | 0.674 ± 0.259 |
> > >
> > > ---
> > >
> > > _TNR Score (C = 0.5, sb = 0.5, n_vars = 5, s_g = 1.0)_
> > >
> > > | Dataset | Granger | PLACy | PCMCI |
> > > |--------|--------|--------|--------|
> > > | $OU(\sigma_g^m = 0)$ | 0.569 ± 0.116 | **0.984 ± 0.035** | 0.979 ± 0.034 |
> > > | $\widehat{OU}(\sigma_g^m = 0)$ | 0.558 ± 0.046 | 0.931 ± 0.064 | **0.974 ± 0.026** |
> > > | $OU(\sigma_g^m > 0)$ | 0.739 ± 0.159 | 0.947 ± 0.054 | **0.990 ± 0.029** |
> > > | $\widehat{OU}(\sigma_g^m > 0)$ | 0.568 ± 0.178 | 0.934 ± 0.043 | **0.973 ± 0.051** |
> > >
> > > ---
> > >
> > > Notably, the relative ranking of the methods remains unchanged after applying the FDR correction, and the overall conclusions are consistent with the original setting.
> > >
> > > The high TNR values remain stable compared to the setting without FDR correction, suggesting that PLaCy is inherently robust to false positives.
> > >
> > > We will include these additional numerical tests in the Appendix of our work and mention them in the main text.
> > >
> > > Furthermore, as in the PCMCI paper, we will highlight in the final manuscript that Benjamini–Hochberg false discovery rate correction will be available as an option for the user in our open-source code, but is not enforced in the main results in order to ensure fair comparison with existing methods and consistency with the state-of-the-art evaluation protocols.

---

### Official Review · Reviewer_7GYz · 2026-03-12

**Soundness:** 4
**Presentation:** 3
**Significance:** 3
**Originality:** 3
**Overall Recommendation:** 5
**Confidence:** 4

**Summary:**

The article introduces PLaCy (Power-Law Causal Discovery) for causality detection in stochastic time series, which utilizes the power-law frequency spectra ubiquitous in real-world systems (e.g., finance, climate, neuroscience). Unlike conventional methods, which require stationarity and single-scale dynamics, PLaCy segments data into overlapping windows to extract local spectral amplitudes and exponents using discrete Fourier transform. These extracted parameters form new time series that represent structural changes and are then analyzed using standard multivariate Granger causality tests. Empirically, synthetic (Ornstein-Uhlenbeck processes) and real data sets (rivers, air quality) are examined. PLaCy outperforms the current baselines in terms of robustness against noise, non-stationarity, and multiplicative effects. The method is particularly good (compared to competing models) under multiplicative noise and non-equilibrium initialization.

**Compliance With Llm Reviewing Policy:**

Affirmed.

**Final Justification:**

The authors addressed my concerns and ran a follow-up experiment that further strenghtened the soundess od the study. I am happy to raise my score to Accept.

**Key Questions For Authors:**

1) What is the impact of keeping the time window (50 points), but reducing the mean-reversal timescale? Does the generalisation claim made on the Rivers dataset, where precipitation lacks clear power-law behavior, works in the synthetic OU testcase too? Does this success rely on the target variable (river level) acting as a power-law medium (even if the cause/precipitation does not)?
2) Follow-up on 1), how well does PLaCy work when having a time dependent mean-reversal timescale? Would PLaCy interpret these purely endogenous as causal signals?
3) Have the authors considered using second-order structure functions to validate the power-law assumption within the fitting window? Also, higher-order structure functions could reveal multifractal behavior. Are there any such signals in any dataset used? Could be also an interesting test for implementations of questions 1) and 2).
4) To what extend can we believe the causal graph of the AirQuality example? What is the basis for assuming causation based on spatial proximity? This is minor, because it seems to be a standard benchmark dataset introduced in an ICLR 2024 paper, but may be still worth commenting on more explicitly (the authors already mention it).

**Limitations:**

The authors already explain the limitations with respect to slowly varying spectra and short time series. Depending on the outcome of the above questions, limitations regarding the con straints for mean-reversal timescale for OU and implications for realistic datasets may be added.

**Strengths And Weaknesses:**

*Strengths*
- Effectively handles multiplicative noise and non-equilibrium dynamics by filtering out transient fluctuations that mimic causal signals in the raw time domain. It does so in an original way by combining known components: DFT, power-law fitting, Granger causality
- The study is well-organized with solid data comparisons that clearly visualize performance gaps between models in solving complex scientific queries. Figure 1 and Algorithm 1 are great and very helpful for a quick conceptual overview.
- The base test with the Ornstein-Uhlenbeck (OU) is very informative and generally well described and motivated.
- Provides a proof sketch for the invariance of linear causal graphs under spectral feature mapping, validating the domain shift from raw signals to spectral parameters.
- Reuses established Granger causality testing frameworks on transformed features, ensuring statistical validity while improving reliability.
While I have not read the appendix in detail, the tests seem to strengthen the soundness.


*Weaknesses*
- The method relies heavily on the presence of power-law spectral distributions. Performance may degrade in systems where this structural regularity is absent or slowly varying (though the authors note some generalization capability).
- To this end, the Ornstein-Uhlenbeck (OU) process with mean-reversal timescale of order of the time window used for the analysis creates a stable lambda = 2 background power-law slope (this is the high-frequency limit of the OU process reducing the Lorentzian spectrum to a simple power-law one). It deserves more explicit discussion of having an uncoupled baseline that is homogeneous across all series and temporally stable, such as if it easier isolates the source variation. Having used the baseline only in such a favrouable regime, while empirical claims extend to nonlinear settings (Appendix E.5), limits the generality of the soundness claims.
- Real-world ground truth (Rivers, AirQuality) is often derived from expert knowledge or distance-based heuristics (AirQuality), not necessarily true causal mechanisms.
- Minor: in Fig. 1, please add ‘log’ notation to the labels in (2) and y-label in (3), which is otherwise a very illustrative Figure!

---

> ### Author Rebuttal · Authors · 2026-03-31
>
> We thank the Reviewer for their careful reading of the paper and for the constructive suggestions. We address each in turn below.
>
> **P1. Power law spectra assumption**
>
> PLaCy is deliberately designed to exploit local power-law structure in the spectrum, and its performance naturally depends on the quality of this approximation.
> Rather than aiming for complete generality, our goal is to improve causal discovery in a broad and practically relevant class of systems where scale-free behavior is present (finance, climate, and so on, as detailed in Section 2.3).
> Importantly, our experiments suggest that PLaCy can provide benefits even when the assumption is only partially satisfied (such as in the Rivers dataset).
> We do not claim robustness in fully non-power-law settings, and we explicitly recommend time-domain alternatives in such cases (Section 6, Appendix D.2).
>
> **P2. Other parametrisations for the OU processes**
>
> We agree, the OU process can induce a relatively stable and homogeneous spectral background with $\lambda \approx 2$ in the high frequency limit (mean-reversal term going to zero) with additive Gaussian noise.
> However, we note that this is "simple" setting is not the regime where PLaCy performs best.
> In fact, in more complex settings such as those with non-Gaussian or multiplicative noise, the spectral exponent becomes more variable. Especially in these scenarios PLaCy outperforms existing approaches.
> We agree that the high-frequency limit deserves an explicit discussion and we will add it to the revised manuscript.
>
> **P3. Minor presentation issue in Figure 1**
>
> We thank the reviewer for catching this. Both labels will be corrected in the camera-ready version.
>
> **P4. Time-dependent mean-reversion and endogenous spectral changes**
>
> _Lower mean-reversion time_
>
> Reducing the mean-reversion timescale in the OU process, while keeping a fixed window size, effectively shortens the correlation length within each window, which is qualitatively similar to increasing the noise level. Empirically (Tables 3 and 5), this does not significantly impact performance, indicating robustness to such changes. Regarding the Rivers dataset, we emphasise that precipitation does not need to exhibit clear power-law behaviour itself. Rather, rainfall events perturb the scaling structure of the observed signal (river level), modifying the local power-law fit within each window.
>
> _Time-dependent mean reversion_
>
> In part, our non-equilibrium experiments mimic the kind of time-dependent behaviour that the Reviewer is suggesting to explore.
> Indeed, initialising the processes far from equilibrium (at value 100 rather than 1), creates a strong transient phase where the effective mean-reversion dynamics are time-varying.
> In any case, following the Reviewer's suggestion, we are performing additional numerical tests with explicitly time-dependent mean-reversion parameters modelled either through a sinusoidal function or through an exponential decay.
> Preliminary results are reported in the tables below.
>
> ---
>
> **Results — N = 5, σ_g = 1.0**
>
> |Dataset|Sin F1 Granger|Sin F1 PLACy|Sin TNR Granger|Sin TNR PLACy|Exp F1 Granger|Exp F1 PLACy|Exp TNR Granger|Exp TNR PLACy|
> |--|--|--|--|--|--|--|--|--|
> |$\text{OU}(\sigma_g^m=0)$|**0.54 ± 0.09**|0.49 ± 0.11|0.47 ± 0.18|**0.53 ± 0.22**|**0.56 ± 0.09**|0.54 ± 0.11|0.55 ± 0.12|**0.69 ± 0.13**|
> |$\widehat{\text{OU}}(\sigma_g^m=0)$|**0.51 ± 0.10**|0.50 ± 0.11|0.44 ± 0.15|**0.54 ± 0.19**|**0.57 ± 0.13**|0.54 ± 0.11|0.55 ± 0.17|**0.66 ± 0.18**|
> |$\text{OU}(\sigma_g^m>0)$|0.67 ± 0.12|**0.79 ± 0.12**|0.73 ± 0.15|**0.85 ± 0.09**|0.61 ± 0.15|**0.77 ± 0.11**|0.68 ± 0.16|**0.86 ± 0.09**|
> |$\widehat{\text{OU}}(\sigma_g^m>0)$|0.68 ± 0.13|**0.72 ± 0.15**|0.70 ± 0.19|**0.81 ± 0.13**|0.61 ± 0.16|**0.81 ± 0.14**|0.61 ± 0.21|**0.86 ± 0.11**|
>
> ---
>
> Although these experiments fall outside the theoretical guarantees of PLACy, the results show equivalent or superior performance in terms of F1 and consistently improved TNR across all analyzed scenarios.
> We hence find that PLaCy does not appear to misinterpret the endogenous time-varying mean-reversion as causal signal since the TNR remains high across all settings.
> We will include a more comprehensive experimental analysis in the camera-ready version of the paper.
>
> **P5. Second-order structure functions**
>
> We thank the Reviewer for this insightful suggestion. We agree that might be interesting to look at higher-order structure functions, to potentially probe multifractal behaviour.
> This would be an interesting direction for future work but we believe it would not fit well within the scope of the current work (which already encompasses dozens of different experimental settings).
>
> **P6. AirQuality dataset**
>
> We agree that the ground-truth values can be approximations. Nevertheless, as the Reviewer also mentioned, for consistency with the existing literature we adopt these standard benchmarks. We will add a brief comment along these lines in the revised manuscript.

---

> > ### Author Rebuttal · Reviewer_7GYz · 2026-04-04
> >
> > The authors addressed my concerns and ran the suggested time-dependent τ_c experiments that turns out to further support PLaCy.

---

> > > ### Author Response · Authors · 2026-04-07
> > >
> > > We thank the Reviewer for their careful and thoughtful review. We are happy that the additional experiments resolved all the raised points.

---

### Official Review · Reviewer_L9M6 · 2026-03-15

**Soundness:** 3
**Presentation:** 3
**Significance:** 3
**Originality:** 3
**Overall Recommendation:** 5
**Confidence:** 2

**Summary:**

This paper deals with causal discovery for time series.
Particularly, it uses an encoding with the frequency spectra.
While the literature is often highly sensitive to the noise, in this paper they adapte a power law to be robust.

**Compliance With Llm Reviewing Policy:**

Affirmed.

**Final Justification:**

The rebuttal has answered to my questions.
I increased my score.

**Key Questions For Authors:**

- I am always troubeld with Granger causality, which is not classical causality (in the sense of Pearl for example). Would the same analysis work wiht calssical causality?
- I am convinced that if there is a frequency dependence, this type of analysis may overpass the potential problems with discrete time points in classical time series caual discovery. However, an ablation study would be itneresting to illustrate / comment on that.

**Limitations:**

- about the metrics in the experimental section, F1 score and TNR are not particularly designed for causality. Maybe you can add the random guessing (see UAI 2025, Petersen), particularly for the rreeal dataset.
- In Eq. 1, it seems that everything relates on L, which should be large enough. In practice, does it change the performance? IS there any criterion to select it?
- what are the assumptions in thm 3.1?

**Strengths And Weaknesses:**

- this is interesting to combine frequency spectra and power-law
- I am always troubeld with Granger causality, which is not classical causality (in the sense of Pearl for example). Would the same analysis work wiht calssical causality?
- I am convinced that if there is a frequency dependence, this type of analysis may overpass the potential problems with discrete time points in classical time series caual discovery. However, an ablation study would be itneresting to illustrate / comment on that.

---

> ### Author Rebuttal · Authors · 2026-03-31
>
> We thank the Reviewer for their positive feedback, we address each question and suggestion below.
>
>
> **P1. Granger causality vs. classical causality**
>
> > I am always troubeld with Granger causality, which is not classical causality (in the sense of Pearl for example). Would the same analysis work wiht calssical causality?
>
> We agree with the Reviewer, and we like the idea of applying our approach in other causality frameworks. Some interesting results are provided in Appendix E.4 (Table 12) where we apply our approach to the PCMCI algorithm. We observe consistent improvements, indicating that the benefits of power-law causal discovery are not necessarily tied to Granger-based estimators.
>
> **P2. Ablation study for frequency-dependent analysis**
>
> > I am convinced that if there is a frequency dependence, this type of analysis may overpass the potential problems with discrete time points in classical time series casual discovery. However, an ablation study would be interesting to illustrate / comment on that.
>
> In Appendix E.4 (Table 12), we present an ablation study comparing Granger causality and PCMCI applied to both discrete time series and their spectral transformation. In both cases, we observe consistent improvements, indicating that the performance gains stem from the frequency-domain representation rather than from any specific causal discovery algorithm. More broadly, this suggests that the proposed spectral transformation may benefit a wider class of causal discovery methods. In the final version of the paper, we will highlight this result and direct readers to the Appendix.
>
>
> **P3. Metrics and baseline**
>
> > About the metrics in the experimental section, F1 score and TNR are not particularly designed for causality. Maybe you can add the random guessing (see UAI 2025, Petersen), particularly for the real dataset.
>
> We agree that F1 and TNR provide only a partial view of causal discovery performance, as they are generic graph-recovery metrics. To address this, we report additional evaluation metrics in Appendix E.6 (Tables 13–14), including the metric proposed in [UAI 2025, Petersen], as well as a more strict variant introduced in this work. These metrics provide a more informative baseline by accounting for random guessing. As shown in the Appendix, PLaCy consistently outperforms competing methods under both metrics.
>
> **P4. Dependence on L**
>
> > In Eq. 1, it seems that everything relates on L, which should be large enough. In practice, does it change the performance? Is there any criterion to select it?
>
> We thank the Reviewer for bringing this point to our attention. The choice of the sliding window length involves a trade-off between statistical reliability and temporal resolution. Short windows may lead to unstable estimates due to limited data, while long windows may smooth out temporal variations and attenuate causal signals. To address this, we introduce a data-driven selection procedure based on the quality of the power-law fit in each window. Specifically, for a set of candidate window lengths, we evaluate the statistical significance of the linear fit in log–log space using a Wald test on the slope parameter.
> We then select the smallest window length for which the fitted slope is statistically significant ($p$-value < 0.05), ensuring that the local power-law approximation is reliable while preserving temporal resolution.
> In Appendix D.1 (Figure 3) we report a systematic study showing that the window length and stride only slightly affect the performance of our approach.
>
>
> **P5. Assumptions for Theorem 3.1**
>
> > What are the assumptions in thm 3.1?
>
> We thank the Reviewer for this question. The theoretical results of Theorem 3.1 rely on the following assumptions:
> (i) the power spectral density is power-law distributed within each analysis window
> (ii) the time series is weakly stationary
> (iii) the causal relationships are all linear
> (iv) the noise is only additive
> Furthermore, as in standard time-series causal discovery, interpreting the recovered graph causally additionally relies on assumptions such as the identifiability of the causal graph.
> These assumptions ensure that the spectral transformation preserves the relevant dependency structure. We will revise the manuscript to state them more explicitly.
>
>
> ---
>
> We thank again the Reviewer for their comments and suggestions.
> We hope we were able to clarify their doubts, and we would be happy to provide any further clarification.

---

> > ### Author Rebuttal · Reviewer_L9M6 · 2026-04-03
> >
> > thanks for the response. I maintain my score, being ok with the paper.

---

> > > ### Author Response · Authors · 2026-04-07
> > >
> > > We thank the Reviewer for their constructive and encouraging assessment. We are pleased that our rebuttal clarified all the raised points .

---

### Decision · Program_Chairs · 2026-04-30

**Decision:**

Accept (spotlight)

**Comment:**

The reviewers agreed that this paper tackles an important problem in time-series causal discovery and proposes a novel, practically relevant approach. They found the idea of performing causal discovery on time-varying power-law spectral features to be interesting and well motivated, and they viewed the empirical evaluation as strong, especially in noisy and non-stationary settings where the method appears to outperform several baselines. All four reviewers were supportive in their final assessments, and several explicitly stated that the rebuttal and follow-up experiments addressed their concerns.

Overall, I recommend acceptance. During discussion, reviewers raised questions about the scope of the power-law assumption, the clarity of the theoretical assumptions and guarantees, the choice of window size and stride, the construction of real-world ground truth, and statistical issues related to multiple testing. I have read the paper and author responses carefully, and I find that these concerns were adequately addressed through clarification and additional experiments, including follow-up analyses on time-varying dynamics and FDR correction. The paper makes a meaningful contribution: the core idea is original, the method is technically solid within its stated assumptions, and the empirical evidence is strong enough to support publication. For the final version, I encourage the authors to further clarify the regime of applicability, make assumptions and benchmark limitations more explicit, and briefly discuss multiple-testing issues in the main paper.